# Diversity as a Reward: Fine-Tuning LLMs on a Mixture of Domain-Undetermined Data

**Zhenqing Ling**[1*], **Daoyuan Chen**[2*], **Liuyi Yao**[2], **Qianli Shen**[2], **Yaliang Li**[2], **Ying Shen**[1,3†]

[1]Sun Yat-sen University, [2]Alibaba Group, [3]FSIETP
lingzhq@mail2.sysu.edu.cn, sheny76@mail.sysu.edu.cn
{daoyuanchen.cdy, yly287738, shenqianli.sql, yaliang.li}@alibaba-inc.com

## Abstract

Fine-tuning large language models (LLMs) using diverse datasets is crucial for enhancing their overall performance across various domains. In practical scenarios, existing methods based on modeling the mixture proportions of data composition often struggle with data whose domain labels are missing, imprecise or non-normalized, while methods based on data selection usually encounter difficulties in balancing multi-domain performance. To address these challenges, in this work, we investigate the role of data diversity in enhancing the overall abilities of LLMs by empirically constructing contrastive data pools and theoretically deriving explanations. Building upon the insights gained, we propose a new method that gives the LLM a dual identity: an output model to cognitively probe and select data based on diversity reward, as well as an input model to be tuned with the selected data. Extensive experiments show that the proposed method notably boosts performance across domain-undetermined data and a series of foundational downstream tasks when applied to various advanced LLMs. We release our code and hope this study can shed light on the understanding of data diversity and advance feedback-driven data-model co-design for LLMs.

## 1 Introduction

The rapid advancement of large language models (LLMs) — exemplified by open-source series such as LLaMA [18], Qwen [53], and DeepSeek [31], as well as closed-source models like GPT [2] — has transformed artificial intelligence by significantly enhancing capabilities in core areas such as common sense reasoning, mathematics, and code generation. Fine-tuning these models further improves their usability by aligning their behaviors with specific instructions and human preferences [43, 39].

To cultivate comprehensive capabilities in LLMs, prior studies have explored preferable trade-offs between quality, quantity, and diversity of their training data [30, 7, 38, 59]. For example, methods focusing on data selection [29, 48, 44] and mixture [21] demonstrate promising capabilities to enhance model performance, particularly through semantic diversity [34, 33].

However, real-world applications frequently encounter unlabeled data and difficulties in domain labeling [22], posing challenges for data mixture methodologies that take the domain tags as a prior, as well as the data selection approaches, which often prioritize quality over diversity, especially with data sourced from quite different domains.

---

*Equal contribution.
†Corresponding author.
[3]FSIETP: Guangdong Provincial Key Laboratory of Fire Science and Intelligent Emergency Technology.

To leverage the best of both worlds, in this work, we dive into the role of semantic diversity of LLM data. Our investigation begins with a systematic analysis of 40k instruction pairs across four capability domains. Through controlled experiments with contrastive data pools, we uncover two insights: (1) Optimal diversity thresholds for model performance vary significantly across architectures and task domains; (2) Current centroid-based diversity metrics [34] fail to capture the dynamic interaction between inter-domain separation and intra-domain variance when labels are unavailable.

These findings motivate our theoretical analysis, which establishes that traditional data mixing strategies assuming known domain weights [54] cannot achieve Pareto-optimal performance in label-free settings. Our analysis further reveals that when domain labels are unavailable, an effective data selection strategy should prioritize samples that enhance sample-level diversity. This strategy involves balancing the goal of enhancing sample-level diversity to approximate the ideal importance weights, with the need to preserve the base model's latent feature geometry.

We operationalize these insights through DAAR, a self-supervised framework that learns **d**iversity **as a r**eward signal through three technical innovations: (1) Automatic synthesis of model-aware domain centroids via iterative embedding-space generation, (2) A lightweight MLP probe trained to predict semantic entropy using only the LLM's frozen embeddings, and (3) Closed-loop fine-tuning where the model's own diversity estimates guide subsequent data selection.

Extensive evaluations across 7 benchmarks and 3 model families reveal that DAAR achieves new state-of-the-art (SOTA) average performance, consistently outperforming 9 baseline methods on high-difficulty tasks while maintaining computational efficiency. Crucially, our method requires no domain labels or external models - the LLM self-supervises its diversity estimation through geometric constraints in its native embedding space. Our core contributions for LLM data optimization are:

- Formal characterization of diversity's dual role in LLM fine-tuning through controlled experiments with contrastive data pools.

- A new method with label-free diversity reward mechanism using embedding-space entropy prediction, theoretically grounded in importance sampling.

- Extensive empirical validations showing consistent gains across model families and tasks, with notable improvements in mathematical reasoning (+27%) and coding (+7.4%), whereas other baselines struggle in such challenging scenarios. Our code is released at https://github.com/modelscope/data-juicer/tree/DaaR to foster more data-centric research for LLMs.

## 2  Preliminaries

### 2.1  Related Works

**Data Selection**  Fine-tuning is a pivotal training paradigm for enhancing LLMs' domain-specific capabilities. Research has shown that a small set of instruction pairs can enable LLMs to follow major instructions effectively [62, 52]. This fine-tuning can be achieved through rule-based methods, which focus on attributes such as Error L2-Norm [37] and token length [40]. More recently, model-based heuristics have been explored, including methods based on instruction-following difficulty [29], GPT scoring [10, 46], data model selection [19], and influence scores derived from loss [48].

**Diversity & Data Mixing**  A fundamental principle for LLMs is being able to handle diverse human requests, underscoring data diversity as essential for effective pre-training [32, 12, 51, 20] and fine-tuning [55, 17]. Data diversity encompasses aspects such as data deduplication [1], the coverage scope of tags [34], model-based diversity evaluation [33, 58], and scaling properties [42]. Typically, data mixing approaches [3, 54, 21, 32] focus on adjusting the proportional weights of different domains to enhance model capabilities.

**Our Position**  This work relates to data mixing and selection fields through its focus on diversity quantification, but advances both of them in three key aspects: First, we address practical challenges in real-world applications where unlabeled datasets hinder existing methods' ability. Second, our framework distinguishes by integrating reward-driven learning with semantic entropy-based selection criteria. Third, we critically examine the under-explored applicability of current selection paradigms in multi-domain contexts, revealing critical limitations in modeling diverse distribution patterns.

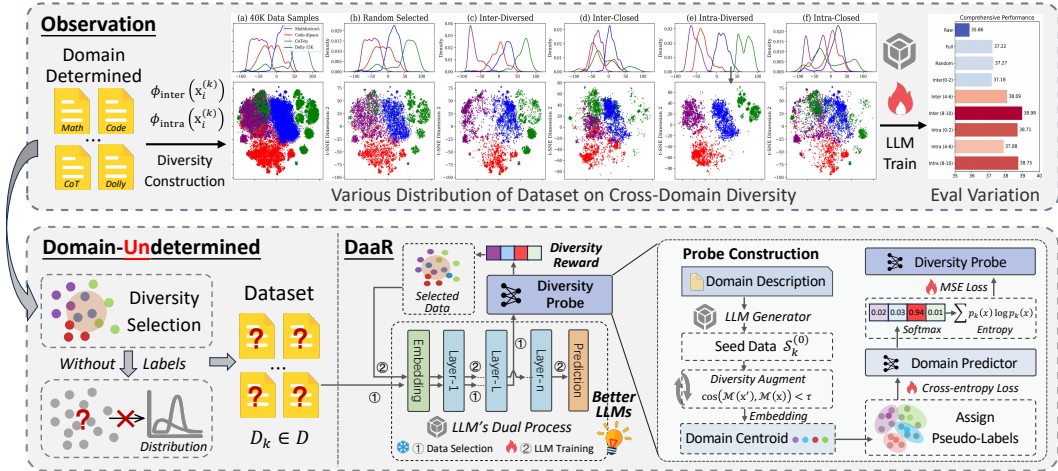

Figure 1: The illustration highlights our observations and the proposed DAAR method. Observation shows the t-SNE visualization of embeddings for data samples with different distributions, leading to varying LLM's evaluation performance. In the domain-undetermined data selection scenario, DAAR introduces a dual-identity framework for LLMs: ① An output model to probe and select data based on diversity reward, ② An input model to be tuned with the selected data. The diversity probe is trained on model-aware synthetic data, enabling domain discrimination and diversity reward prediction.

## 2.2 Problem Setup

In this paper, we consider learning from domain-undetermined data mixtures to enhance LLM downstream performance in target domains. Let $\{\mathcal{D}_k\}_{k=1}^K$ denote $K$ target domain distributions, where each $\mathcal{D}_k : \mathcal{X} \to [0,1]$ defines a probability distribution over samples $x \in \mathcal{X}$. For model parameters $\theta$, the domain-specific performance is measured by $\mathbb{E}_{x \sim \mathcal{D}_k}[\ell(x;\theta)]$. Our objective is to minimize the weighted multi-domain loss: $\sum_{k=1}^K \lambda_k \mathbb{E}_{x \sim \mathcal{D}_k}[\ell(x;\theta)]$, where $\lambda_{1:K}$ are non-negative weights with $\sum_{k=1}^K \lambda_k = 1$, reflecting application-specific priorities.

For each domain $\mathcal{D}_k$, we assume only a natural language description $d_k$ is available while **NO** target domain samples are accessible, contrasting with conventional data selection methods requiring labeled examples [48, 50, 41, 54] in LLM training. We are provided with an **domain-unlabeled** dataset $\mathcal{D} = \{x_i\}_{i=1}^N$ containing domain-agnostic samples. A straightforward approach is direct fine-tuning using all $\mathcal{D}$, but it often yields suboptimal results (empirically evidenced in Table 2). Instead, we propose selecting a subset via binary vector $\alpha \in \{0,1\}^N$, where $\alpha_i = 1$ indicates $x_i$ is selected. The LLM is trained by solving: $\min_\theta \sum_{i=1}^N \alpha_i \ell(x_i;\theta)$, leading to the bi-level optimization:

$$\min_\alpha \sum_{k=1}^K \lambda_k \mathbb{E}_{x \sim \mathcal{D}_k}[\ell(x;\theta^*(\alpha))], \quad \text{s.t. } \theta^*(\alpha) = \arg\min_\theta \sum_{i=1}^N \alpha_i \ell(x_i;\theta). \tag{1}$$

All notations used are summarized in Appendix A.

## 3 Diversity: A Critical Factor for LLM Fine-Tuning

In this section, we investigate how domain-specific diversity in labeled training data modulates model capabilities through empirical analysis of well-designed data pools, complemented by a theoretical perspective that explains the observation and provides mechanistic insights.

### 3.1 Seed Data Pools and Basic Setting

**Data Pools and Sources** To explore how selecting data samples from extensive and diverse data repositories affects the foundational capabilities of LLMs, we construct various data pools and consistently fine-tune LLMs on them. The seed data pool is sourced from the following datasets: Dolly-15k [15] for common sense, Cot-en [13] for reasoning, Math-Instruct [56] for mathematics,

and Code-Alpaca [6] for coding. Each dataset was randomly tailored to 10,000 entries, resulting in a combined data pool of 40,000 entries. Following instruction tuning practices [62, 33], we then uniformly sample 8,000 data entries as a reference data pool for the random baseline.

**Benchmarks** Aligned with representative capabilities of leading open-source LLMs, we select the following widely used evaluation sets: NQ [28] and TriviaQA [26] for common sense, Hellaswag [57] for reasoning, GSM8K [14] and MATH [24] for mathematics, MBPP [4] and HumanEval [11] for coding. To evaluate the comprehensive performance of LLMs across domains, we employ the average metric (AVG) as the primary evaluation criterion.

**Models & Implementation** We employ the Qwen2 series (Qwen2-7B & Qwen2.5-7B) [53] and the Llama3.1-8B [18] as representative SOTA base models to be fine-tuned.

All experiments are conducted under identical training and evaluation protocols with two independent repetitions. Full platform, training and evaluation details are provided in Appendix D.

## 3.2 Data Pools with Contrastive Distributions

To systematically analyze the impact of domain-specific diversity patterns on model capabilities, we propose a contrastive construction with three phases: (A) Foundational Definitions, (B) Diversity Metric Formulation, and (C) Distribution Synthesis.

**(A) Foundational Definitions** Each domain dataset $\mathcal{D}_k$ contains $N_k = |\mathcal{D}_k|$ samples, and we represent each data sample through its semantic embedding $\mathbf{x}_i^{(k)} \in \mathbb{R}^d$ extracted from the ***embedding-layer*** of the pretrained LLM. The domain centroid $\mathcal{C}_k$ serves as the semantic prototype $\mathcal{C}_k = \frac{1}{N_k} \sum_{i=1}^{N_k} \mathbf{x}_i^{(k)}$. This centroid-based representation enables geometric interpretation of domain characteristics in the embedding space. We dissect data diversity into two complementary aspects:

**(B.1) Inter-Diversity** It quantifies the diversity between distinct domains through centroid geometry. For sample $\mathbf{x}_i^{(k)}$, its cross-domain similarity is measured by $\phi_{\text{inter}}(\mathbf{x}_i^{(k)})$. The global inter-diversity metric $\Phi_{\text{inter}}$ computes the expected pairwise centroid distance:

$$\phi_{\text{inter}}(\mathbf{x}_i^{(k)}) = \sum_{\substack{j=1 \\ j \neq k}}^{K} \frac{\mathbf{x}_i^{(k)} \cdot \mathcal{C}_j}{\|\mathbf{x}_i^{(k)}\|\|\mathcal{C}_j\|}, \quad \Phi_{\text{inter}} = \mathbb{E}_{k \neq l}\left[\|\mathcal{C}_k - \mathcal{C}_l\|_2\right] = \frac{1}{\binom{K}{2}} \sum_{k=1}^{K-1} \sum_{l=k+1}^{K} \|\mathcal{C}_k - \mathcal{C}_l\|_2. \quad (2)$$

This formulation reflects a key insight: maximizing $\Phi_{\text{inter}}$ encourages domain separation, while minimization leads to overlapping representations. Observation of the upper part of Fig. 1 (full-size version is in Appendix, Fig. 17) demonstrates this continuum through t-SNE projections – high $\Phi_{\text{inter}}$ manifests as distinct cluster separation with clear margins (Fig. 17.(c)), whereas low values produce entangled distributions (Fig. 17.(d)). Full analysis is detailed in Appendix H.1.

**(B.2) Intra-Diversity** Focusing solely on the separation between different domains may hinder the model's ability to learn the knowledge specific to a given domain. Hence we calculate sample similarity to its domain center in $\phi_{\text{intra}}(\mathbf{x}_i^{(k)})$, and the domain-level variance metric is defined as $\Phi_{\text{intra}}^{(k)}$.

$$\phi_{\text{intra}}(\mathbf{x}_i^{(k)}) = \frac{\mathbf{x}_i^{(k)} \cdot \mathcal{C}_k}{\|\mathbf{x}_i^{(k)}\|\|\mathcal{C}_k\|}, \quad \Phi_{\text{intra}}^{(k)} = \frac{1}{N_k} \sum_{i=1}^{N_k} \|\mathbf{x}_i^{(k)} - \mathcal{C}_k\|_2^2. \quad (3)$$

Controlled manipulation of $\Phi_{\text{intra}}$ reveals critical trade-offs: lower variance enhances domain-specific learning but risks over-specialization, while higher variance improves robustness with the risk of cross-domain interference. The visualization in Fig. 17 (e-f) illustrates this scenario that concentrated distributions exhibit sharp marginal peaks, while dispersed variants show overlapping density regions.

**(C) Distribution Synthesis** For each domain $\mathcal{D}_k$, we compute sample-wise diversity scores $\{\phi_{\text{inter}}(\mathbf{x}_i^{(k)})\}_{i=1}^{N_k}$ and $\{\phi_{\text{intra}}(\mathbf{x}_i^{(k)})\}_{i=1}^{N_k}$. The construction proceeds via partition each $\mathcal{D}_k$ into 20% intervals based on the percentiles of $\phi_{\text{inter}}$ for **inter-diversity control**, and partition the $\phi_{\text{intra}}$ scores into 20% quantile intervals for **intra-diversity control**.

The 20% interval results in five choices of data selection per domain, parameterizing the trade-off between diversity preservation and domain specificity. As demonstrated in Appendix H.2 (Fig. 19 and Fig. 20), this quantization process induces measurable distribution shifts.

### 3.3 Experimental Observations

Table 1 presents comprehensive evaluations across seven benchmarks, where the notation *Inter-Diversity (X-Y)* indicates samples ranked in the top (100-Y)% to (100-X)% of cross-domain similarity scores. Due to space constraints, we present only the results for the top 20%, middle 20%, and bottom 20%. Detailed performance results on various downstream tasks are shown in Appendix H.4. Our diversity-controlled selections reveal two observations:

- **Varied Improvement Patterns**: Both models demonstrate marked improvements over RAW distributions across diversity conditions, but the effects of their improvements vary. For Llama3.1-8B, *Inter-D (80-100)* achieves 38.98 average accuracy (+3.12 over RAW), outperforming the RANDOM baseline by 1.71, while *Inter-D (0-20)* is below RANDOM.

- **Model-Dependent Performance Peak**: Each model exhibits distinct optimal operating points along the diversity spectrum. Llama3.1-8B reaches peak performance at *Inter-D (80-100)* and *Intra-D (80-100)*, suggesting complementary benefits from both diversity types. Qwen2-7B peaks in inter-type selection at low inter-diversity, while it peaks in intra-type selection at high intra-diversity.

These results show the promising potential of diversity-aware data selection, motivating us to further understand the performance variance more formally and propose principled solutions to adaptively achieve the performance peaks.

**Remarks** Despite existing positive improvements on overall performance, two constraints merit consideration for real-world applications. **(1) Distribution Transiency**: The optimal diversity parameters (e.g., 80-100 vs. 40-60) show sensitivity across tasks and models, necessitating automated and potentially costly methods. **(2) Label Dependency**: The studied heuristic strategies *Inter-Diversity* and *Intra-Diversity* currently require domain-labeled data for centroid calculation.

### 3.4 Theoretical Perspective

To explain why diversity serves as a crucial factor in LLM's cross-domain performance and addresses distribution transiency, we present a theoretical analysis based on important sampling. We first formulate the data selection dynamics and then incorporate diversity into it, thereby revealing the mechanistic basis and offering insights for our method.

Table 1: Performance of Llama3.1-8B and Qwen2-7B on AVG with different constructed Inter-Diversity and Intra-Diversity distributions.

| Models | Distribution | AVG |
|---|---|---|
| | **RAW** | 35.86 |
| | **FULL (40K)** | 37.22 |
| | **RANDOM (8K)** | 37.27 |
| **Llama3.1** | Inter-D (0-20) | 37.18 |
| | Inter-D (40-60) | 38.09 |
| | **Inter-D (80-100)** | **38.98** |
| | Intra-D (0-20) | 38.71 |
| | Intra-D (40-60) | 37.88 |
| | **Intra-D (80-100)** | **38.75** |
| | **RAW** | 41.47 |
| | **FULL (40K)** | 53.01 |
| | **RANDOM (8K)** | 53.02 |
| **Qwen2** | **Inter-D (0-20)** | **54.13** |
| | Inter-D (40-60) | 52.47 |
| | Inter-D (80-100) | 51.07 |
| | Intra-D (0-20) | 48.72 |
| | Intra-D (40-60) | 52.85 |
| | **Intra-D (80-100)** | **53.25** |

**Importance Sampling** We assume that there exists an ideal target distribution $q$ that minimizes the multi-domain loss in Eq. (1). Hence, the process of data selection can be framed as aligning the source distribution $p$ (empirical distribution of $\mathcal{D}$) toward the optimal distribution $q$. To bridge this distribution shift, we adopt an importance sampling perspective [50, 49, 48] where the optimal subset corresponds to reweighting samples by the density ratio $\frac{q(x)}{p(x)}$:

$$\mathbb{E}_{x \sim q(\cdot)}\ell(x) = \int_x q(x)\ell_\theta(x)dx = \int_x p(x)\frac{q(x)}{p(x)}\ell_\theta(x)dx = \mathbb{E}_{x \sim p(\cdot)}\frac{q(x)}{p(x)}\ell(x;\theta) = \sum_{x \sim \mathcal{D}}\frac{q(x)}{p(x)}\ell(x;\theta). \quad (4)$$

The domains of LLM's data are commonly defined by textual descriptions, which inherently constrain the semantic scope. Thus we make the following assumption regarding distributions $p$ and $q$:

**Assumption 3.1** (Shared Support). For any given domain index $c \in \{1, \ldots, K\}$, the support of $p$ and $q$ is shared, formalized as: $p(x|c = k) = q(x|c = k), \quad k = 1, \ldots, K$.

Assumption 3.1 implies that the discrepancy between the source distribution $p$ and the target distribution $q$ arises solely from differences in domain proportions, leading to the following proposition.

**Proposition 3.2.** *The diversity is entirely attributable to the relative weights assigned to each domain:*

$$\frac{q(x)}{p(x)} = \sum_{k=1}^{K} \lambda_k p(c=k|x), \lambda_k := \frac{q(c=k)}{p(c=k)} \tag{5}$$

The derivation is detailed in Appendix C.1. This decomposition reveals that the Inter-Diversity and Intra-Diversity are governed by $\lambda_k$ and $p(c=k|x)$, leading to the following analysis.

**Why Overlook Diversity is Suboptimal** We define $k^*(x) := \arg\max_k p(c=k|x)$ as the prediction of the classifier $p(c=k|x)$. Consider a special case where the classifier yields deterministic predictions, *i.e.* $p(c=k^*(x)|x) = 1, \forall x \in \mathcal{D}$. In this case, the importance weight reduces to $\frac{q(x)}{p(x)} = \sum_{k=1}^{K} \lambda_k p(c=k|x) = \lambda_{k^*(x)}$. Therefore, importance sampling can be applied within:

$$\sum_{x \in \mathcal{D}} \frac{q(x)}{p(x)} \ell(x;\theta) = \sum_{x \in \mathcal{D}} \lambda_{k^*(x)} \ell(x;\theta) = \sum_{k=1}^{K} \sum_{x \in \mathcal{D}: k^*(x)=k} \lambda_{k*(x)} \ell(x;\theta). \tag{6}$$

This formulation is commonly used when domain labels are available, under the implicit assumption that domains are mutually exclusive. However, such an assumption rarely holds in the context of LLM training, as cross-domain data may either synergize or conflict [59, 49, 27] (evidence in Table 1). Hence, neglecting diversity in data selection potentially leads to suboptimal overall performance. We further formalize this suboptimality by analyzing the approximation error incurred by such a deterministic assignment, which provides a supplementary justification for using predictive entropy as a selection criterion. For a detailed analysis, please refer to Appendix C.2.

**Guidance to Our Method** Under the domain-undetermined scenario, our proposed method DAAR operationalizes the gained insights by **(1)** constructing *pseudo-labels* (Section 4.1) align with LLM-aware generated seed, **(2)** explicitly modeling the classifier $p(c=k|x)$ through a *domain discrimination probe* (Section 4.2), and **(3)** using the *probe's predictive entropy* (Section 4.2) as a proxy for diversity, estimating the ability of classifier $p(c=k|x)$ to select data toward the target distribution $q$.

# 4 DAAR: Diversity as a Reward

To address the challenges identified in Sec. 3.3 and leverage the insights gained in Sec. 3.4, we establish a data selection method DAAR guided by diversity-aware reward signals.

It comprises three key components illustrated in Fig. 1: (1) model-aware centroid synthesis, which generates domain-representative centroids and seed data capturing the LLM's intrinsic feature space, (2) two-stage training with reward probe, which yields a probe module capable of predicting sample-level diversity rewards accurately, and (3) diversity-driven data selection to obtain a data subset that effectively boosts LLM's cross-domain performance.

## 4.1 Model-Aware Training Data

**Model-aware Centroid Construction** The proposed method initiates with centroid self-synthesis through a two-phase generation process to address two fundamental challenges: (1) eliminating dependency on human annotations through automated domain prototyping, and (2) capturing the base model's intrinsic feature space geometry for model-aware domain separation.

- **Phase 1 - Seed Generation**: For each domain $k$, generate seed samples $\mathcal{S}_k^{(0)}$ via zero-shot prompting with domain-specific description templates generated by LLM itself, alongside with minor injection from downstream task samples, establishing initial semantic anchors. We show that removing the minor injection of downstream samples **negligibly impacts final performance** but significantly reduces data generation efficiency in Appendix G.8.

- **Phase 2 - Diversity Augmentation**: Iteratively expand $S_k^{(t)}$ through context-aware generation, conditioned on a sliding window buffer with 3 random anchors sampled from the $(t-1)$ iteration $S_k^{(t-1)}$. The generated sample $\mathbf{x}'$ is retained through rejection sampling to enhance diversity.:

$$\max_{\mathbf{x} \in S_k^{(t-1)}} \cos(\mathcal{M}_{\text{ebd}}(\mathbf{x}'), \mathcal{M}_{\text{ebd}}(\mathbf{x})) < \tau, \tag{7}$$

where $\mathcal{M}_{\text{ebd}}(\cdot)$ indicates the output of the embedding layer of the given LLM, with $\tau$ as the similarity threshold. This process terminates when it reaches the predetermined iteration.

The domain centroid is then computed from the final augmented set $\mathcal{S}_k$ using the model's embedding, formalized as $\mathcal{C}_k = 1/|\mathcal{S}_k| \cdot \sum_{\mathbf{x}_i \in \mathcal{S}_k} \mathcal{M}_{\text{ebd}}(\mathbf{x}_i)$. This captures the LLM's intrinsic feature space geometry while eliminating dependency on human annotations.

More implementation details including hyper-parameters and the data generation process, along with their corresponding ablation studies, are provided in Appendix B.

**Domain-Aware Clustering** We then automatically construct pseudo-labels for the given data samples based on the previously synthesized centroids $\{C_k\}_{k=1}^{K}$. We perform constrained k-means clustering in the embedding space with $\arg\min_{\{S_k\}} \sum_{k=1}^{K} \sum_{\tilde{x} \in S_k} \|\mathcal{M}_{\text{ebd}}(\tilde{x}) - \mathcal{C}_k\|^2$. This produces the seed dataset $\mathcal{D}_{\text{probe}}$ containing pseudo-labels $\{\tilde{y}_i\}_{i=1}$ where $\tilde{y}_i \in \{1, \dots, K\}$, with model-induced and embedding-derived domain label assignments.

## 4.2 Training for Self-Rewarding Abilities

**Entropy as a Diversity Proxy** While diversity metrics in Eqs. (2)-(3) directly quantify cross-sample distribution, we aim to design a reward probe that outputs with sample-level diversity level. Since models require sample-level processing that is unable to leverage pairwise relationships, direct training for diversity leads to poor performance in Appendix F.6. We instead use softmax confidence scores and predictive entropy as an implicit diversity proxy, reflecting model-aware data discriminability. This enables effective diversity approximation through a two-stage framework.

**Stage 1: Domain Predictor** The proposed DAAR establishes model-aware domain discrimination abilities through a multi-layer perceptron (MLP) probe module, $\psi_{\text{dom}}$, attached to the hidden layer $\mathcal{M}(\tilde{x})$ of the LLMs. The probe will be trained meanwhile all the parameters of the LLM are frozen, achieving a preferable balance between effectiveness and cost, with detailed analysis regarding the choice of layers presented in Appendix G.2. Specifically, with pseudo-label $\tilde{y}$, we can compute domain predicted probabilities as:

$$p_k(\tilde{x}) = \text{softmax}\left(\psi_{\text{dom}}(\mathcal{M}(\tilde{x}))\right), \quad \psi_{\text{dom}} : \mathbb{R}^d \to \mathbb{R}^K, \tag{8}$$

$\psi_{\text{dom}}$ is optimized via cross-entropy loss $\mathcal{L}_{\text{dom}} = -\frac{1}{|\mathcal{D}_{\text{probe}}|} \sum_{(\tilde{x}, \tilde{y}) \in \mathcal{D}_{\text{probe}}} \sum_{k=1}^{K} \mathbb{I}_{[k=\tilde{y}]} \log p_k(\tilde{x})$, where $\mathbb{I}_{[k=\tilde{y}]}$ denotes the indicator function. We employ single-sample batches with the AdamW optimizer to prevent gradient averaging across domains. Training consistently converges and achieves 92.7% validation accuracy on domain classification, as shown in Fig. 2 (a).

**Stage 2: Diversity Rewarding** Building on the stabilized domain discrimination probe, we quantify sample-level diversity through predictive entropy $H(\tilde{x})$. And to enable efficient reward computation during data selection, we then train another 5-layer MLP $\psi_{\text{div}}$ to directly estimate $H(\tilde{x})$ from $\mathcal{M}(\tilde{x})$:

$$H(\tilde{x}) = -\sum_{k=1}^{K} p_k(\tilde{x}) \log p_k(\tilde{x}). \quad \hat{H}(\tilde{x}) = \psi_{\text{div}}(\mathcal{M}(\tilde{x})), \psi_{\text{div}} : \mathbb{R}^d \to \mathbb{R}^+. \tag{9}$$

This diversity probe module $\psi_{\text{div}}$ shares $\psi_{\text{dom}}$'s architecture up to its final layer that replaced with the regression head, trained using entropy-scaled MSE loss $\mathcal{L}_{\text{div}} = \frac{1}{|\mathcal{D}_{\text{probe}}|} \sum_{\tilde{x} \in \mathcal{D}_{\text{probe}}} (\hat{H}(\tilde{x}) - H(\tilde{x}))^2$. The module is also well-converged as shown in Fig. 2 (b).

**Data Selection**: After training the module $\psi_{\text{div}}$, we can use its output to select data samples. Building on the theoretical insights in Sec. 3.4, data points that are closer to other centroids and more dispersed within their own centroid are more beneficial for enhancing the comprehensive capabilities of the model. Therefore, we use the predicted entropy score as a reward, selecting the top 20% with the highest scores as the final data subset for fine-tuning. The analysis of stability is shown in Appendix G.10.

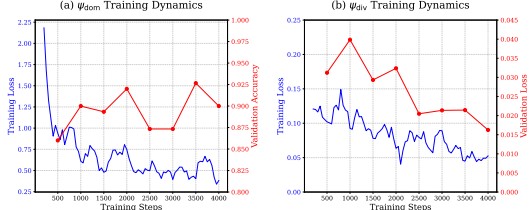

Figure 2: Training loss and validation process of the two stages of DAAR on Qwen2-7B, more detailed results in Appendix H.3.

Table 2: Evaluation results of Llama3.1-8B, Qwen2-7B, and Qwen2.5-7B across various downstream task benchmarks. DAAR demonstrates superiority in AVG compared to other baselines.

| Models | Distribution | Common Sense | | Reasoning | Mathematics | | Coding | | Avg |
| | | NQ | TriviaQA | Hellaswag | GSM8K | MATH | MBPP | HumanEval | |
|---|---|---|---|---|---|---|---|---|---|
| Llama3.1-8B | RAW | 14.13 | 65.90 | 74.62 | 54.80 | 7.90 | 5.00 | 28.66 | 35.86 |
| | FULL (40K) | 21.92 | 65.11 | 73.62 | 51.70 | 7.60 | 4.10 | 36.50 | 37.22 |
| | RANDOM (8K) | 21.99 | 64.83 | 74.72 | 55.70 | 14.50 | 5.10 | 24.09 | 37.27 |
| | INSTRUCTION LEN | 15.34 | 63.60 | 73.73 | 54.00 | 15.40 | 3.60 | 30.80 | 36.64 |
| | ALPAGASUS [10] | 21.57 | 64.37 | 74.87 | 55.20 | 17.65 | 4.60 | 16.16 | 36.34 |
| | INSTAG-BEST [34] | 18.12 | 64.96 | 74.01 | 55.70 | 15.50 | 4.80 | 37.81 | 38.70 |
| | SUPERFILTER [29] | 22.95 | 64.99 | 76.39 | 57.60 | 6.05 | 2.60 | 40.55 | 38.73 |
| | DEITA-BEST [33] | 15.58 | 64.97 | 74.21 | 55.00 | 13.05 | 4.60 | 34.46 | 37.41 |
| | DAAR (Ours) | 20.08 | 64.55 | 74.88 | 54.80 | 15.30 | 4.70 | 37.50 | **38.83** |
| Qwen2-7B | RAW | 8.03 | 59.58 | 73.00 | 78.00 | 5.70 | 5.00 | 60.98 | 41.47 |
| | FULL (40K) | 15.61 | 58.75 | 72.51 | 73.80 | 31.30 | 51.70 | 67.38 | 53.01 |
| | RANDOM (8K) | 13.28 | 58.27 | 73.00 | 75.35 | 35.36 | 52.20 | 63.72 | 53.02 |
| | INSTRUCTION LEN | 8.62 | 58.44 | 72.86 | 73.30 | 27.05 | 53.10 | 63.72 | 51.01 |
| | ALPAGASUS [10] | 13.67 | 57.94 | 73.04 | 73.90 | 32.30 | 51.40 | 63.41 | 52.24 |
| | INSTAG-BEST [34] | 9.51 | 58.50 | 73.06 | 74.70 | 35.35 | 51.90 | 64.70 | 52.53 |
| | SUPERFILTER [29] | 19.16 | 58.98 | 72.99 | 73.70 | 30.10 | 52.40 | 58.85 | 52.31 |
| | DEITA-BEST [33] | 16.41 | 57.80 | 72.70 | 76.10 | 29.05 | 52.40 | 64.63 | 52.73 |
| | DAAR (Ours) | 16.88 | 57.58 | 73.03 | 75.40 | 38.1 | 52.00 | 64.94 | **53.99** |
| Qwen2.5-7B | RAW | 8.84 | 58.14 | 72.75 | 78.20 | 9.10 | 7.40 | 78.05 | 44.64 |
| | FULL (40K) | 12.88 | 58.60 | 72.28 | 76.80 | 13.60 | 62.80 | 71.04 | 52.57 |
| | RANDOM (8K) | 11.46 | 57.85 | 73.08 | 78.90 | 13.15 | 62.50 | 71.65 | 52.65 |
| | INSTRUCTION LEN | 11.34 | 58.01 | 72.79 | 78.00 | 15.80 | 62.30 | 68.12 | 52.34 |
| | ALPAGASUS [10] | 10.40 | 57.87 | 72.92 | 77.20 | 18.75 | 61.80 | 65.55 | 52.07 |
| | INSTAG-BEST [34] | 11.08 | 58.40 | 72.79 | 76.40 | 16.40 | 62.90 | 70.43 | 52.63 |
| | SUPERFILTER [29] | 13.54 | 58.51 | 72.89 | 79.30 | 11.35 | 39.50 | 65.25 | 48.62 |
| | DEITA-BEST [33] | 10.50 | 58.17 | 73.14 | 74.60 | 16.60 | 62.00 | 72.26 | 52.47 |
| | DAAR (Ours) | 15.83 | 58.65 | 72.48 | 80.20 | 16.70 | 64.20 | 68.29 | **53.76** |

## 5 Experiments

To validate the efficacy of DAAR, we conduct comprehensive experiments on data pools and benchmarks in Sec. 3.1, comparing SOTA baselines as follows with critical modifications: all domain-specific labels are deliberately stripped. This constraint mimics more challenging real-world scenarios and precludes direct comparison with data mixture methods requiring domain label prior.

**Baselines** We use the following data selection methods for comprehensive evaluation: (1) RANDOM SELECTION: traditional random sampling; (2) INSTRUCTION LEN: measuring instruction complexity by token count [5]; (3) ALPAGASUS [10]: using ChatGPT for direct quality scoring of instruction pairs; (4-5) INSTAG [34]: semantic analysis approach with INSTAG-C (complexity scoring via tag quantity) and INSTAG-D (diversity measurement through tag set expansion); (6) SUPERFILTER [29]: response-loss-based complexity estimation using compact models; (7-9) DEITA [33]: model-driven evaluation with DEITA-C (complexity scoring), DEITA-Q (quality scoring), and DEITA-D (diversity-aware selection). Detailed implementations for these baselines are in Appendix E.1.

### 5.1 Overall Performance

The main experimental results are presented in Table 2, where INSTAG-BEST and DEITA-BEST represent the optimal variants from their method families. Our experiments clearly demonstrate the effectiveness of DAAR across three major language models and seven challenging benchmarks:

**High-Difficulty Scenario** : The task of balanced capability enhancement proves particularly challenging for existing methods. While some baselines achieve strong performance on specific tasks (e.g., SuperFilter's 40.55 on HumanEval for Llama3.1), they suffer from catastrophic performance drops in other domains (e.g., SuperFilter's 6.05 on MATH). **Only 3 baselines** perform better than RANDOM on Llama3.1-8B. In the Qwen series, even **all baselines** fall below the RANDOM performance. We conjecture this stems from *selected distribution-bias*, as the baselines are **unable to balance** the proportions across different domains, a perspective visually supported and analyzed in Appendix E.2.

Table 3: Illustration results on MMLU, Qwen3 and customization, detailed in Table 7, 10, 11

(a) MMLU Performance

| Qwen2-7B | MMLU-Avg |
|---|---|
| RAW | 28.98 |
| RAND (8K) | 68.81 |
| INSTRUCTION-L | 68.55 |
| ALPAGASUS | 59.34 |
| INSTAG-BEST | 68.65 |
| SUPERFILTER | 54.27 |
| DEITA-BEST | 68.08 |
| **DAAR (Ours)** | **69.42** |

(b) Eval on Qwen3

| Qwen3-8B | Avg |
|---|---|
| RAW | 49.41 |
| RAND (8K) | 48.16 |
| INSTRUCTION-L | 49.19 |
| ALPAGASUS | 47.81 |
| INSTAG-BEST | 48.69 |
| SUPERFILTER | 49.51 |
| DEITA-BEST | 49.90 |
| **DAAR (Ours)** | 50.06 |

(c) Customized Selection on Domain*

| Qwen2.5 | NQ | Hellaswag | Avg |
|---|---|---|---|
| RAW | 8.84 | 72.75 | 44.64 |
| RAND | 11.46 | 73.08 | 52.66 |
| INSTAG | 11.08 | 72.79 | 52.63 |
| DAAR | 15.83 | 72.48 | **53.76** |
| **Common*** | **17.56** | 72.51 | 52.85 |
| **Reason*** | 16.56 | **74.62** | 53.39 |

**Comprehensive Performance** : DAAR establishes new SOTA averages across all models, surpassing the best baselines by **+0.14** (Llama3.1), **+0.97** (Qwen2), and **+1.11** (Qwen2.5). The proposed method uniquely achieves *dual optimization* in critical capabilities: **Mathematical Reasoning**: Scores 38.1 MATH (Qwen2) and 16.70 MATH (Qwen2.5), with 7.4% and 27.0% higher than respective random baselines. **Coding Proficiency**: Maintains 64.94 HumanEval (Qwen2) and 64.20 MBPP (Qwen2.5) accuracy with <1% degradation from peak performance. This demonstrates DAAR's ability to enhance challenging **STEM** capabilities while preserving core competencies.

## 5.2 Generalizability, Uniqueness and Stability of DAAR

**Evaluation on MMLU** In the main experiments, the data and benchmarks we employed are meticulously aligned, which validates the **In-Distribution** capabilities of DAAR. To further validate the generalization ability of DAAR, we present MMLU [23] evaluations in Appendix F.1. MMLU encompasses a significantly broader range of content than our dataset, thus serving as an **Out-of-Distribution** scenario. Results demonstrate that DAAR achieved **top accuracy** on Qwen2 with an improvement of **0.61 points**, and secured a competitive second-place performance on Qwen2.5. At a granular level, DAAR particularly excels in the **STEM** domain, outperforming baselines by an average of **7.6%** and **1.7%** on Qwen2 and Qwen2.5, which aligns the findings above.

**Evaluation on Qwen3** In Appendix F.2, we explore the performance of DAAR on the mainstream RL-based LLM, **Qwen3-8B**. We observe that despite difference in structure, the phenomena (Fig. 9 & Table 9) and the dynamics of training (Fig. 10) **remain consistent** with the main observations. Then we compare DAAR with baselines in Table 10, although Qwen3-8B is not directly designed for SFT paradigm, DAAR maintains the **top position** in AVG performance across baselines, surpasses all baselines **up to 4.7%** and notably exceeds the avg of baselines by **25.91%** on **MATH**.

**Customized Characteristics** Unlike baselines, DAAR enables domain-aware data selection in undetermined scenario, showcasing potential customization on $\lambda_k$. As detailed in Appendix F.3, by adjusting the **selection ratio** for specific domains, we achieve **SOTA performance in targeted domain** without compromising overall ability, promising practical utility in real-world application.

Table 4: DAAR using model-aware pseudo-labels vs. Ground-Truth (GT) labels.

| Model | Setting | NQ | TriviaQA | Hellaswag | GSM8k | MATH | MBPP | HumanEval | Avg |
|---|---|---|---|---|---|---|---|---|---|
| Llama3.1-8B | DAAR w/ Pseudo | 20.08 | 64.55 | 74.88 | 54.80 | 15.30 | 4.70 | 37.50 | 38.83 |
| | DAAR w/ GT | 22.54 | 65.52 | 73.43 | 53.80 | 14.35 | 4.40 | 36.50 | 38.65 |
| | *Diff* | *+2.46* | *+0.97* | *-1.45* | *-1.00* | *-0.95* | *-0.30* | *-1.00* | ***-0.18*** |
| Qwen2-7B | DAAR w/ Pseudo | 16.88 | 57.58 | 73.03 | 75.40 | 38.10 | 52.00 | 64.94 | 53.99 |
| | DAAR w/ GT | 18.75 | 58.35 | 72.02 | 73.40 | 35.75 | 51.50 | 64.01 | 53.40 |
| | *Diff* | *+1.87* | *+0.77* | *-1.01* | *-2.00* | *-2.35* | *-0.50* | *-0.93* | ***-0.59*** |
| Qwen2.5-7B | DAAR w/ Pseudo | 15.83 | 58.65 | 72.48 | 80.20 | 16.70 | 64.20 | 68.29 | 53.76 |
| | DAAR w/ GT | 16.04 | 58.75 | 71.87 | 79.40 | 16.30 | 64.25 | 67.76 | 53.48 |
| | *Diff* | *+0.21* | *+0.10* | *-0.61* | *-0.80* | *-0.40* | *+0.05* | *-0.53* | ***-0.28*** |

**Robustness and Stability** We validate our model-aware design choices that relying on external ground-truth labels can degrade performance by **up to 0.59 points** in Table 4 and Appendix F.4. Furthermore, we demonstrate high end-to-end robustness, with independent runs yielding inter-centroid **similarities of >0.98** and a final performance standard deviation of **only 0.08** in Appendix F.5

## 5.3 Ablation Studies and Discussions

**Ablation Studies on Centroid Construction**
As shown in Fig.3 and Appendix G.1, DAAR can generate domain-representative samples with clear distinctions. Notably, the diversity is consistently most pronounced between common sense, reasoning, and coding domains **across model architectures and parameters**, showing that despite differences in model architecture across LLMs, their discriminability between these domains exhibit remarkable similarity.

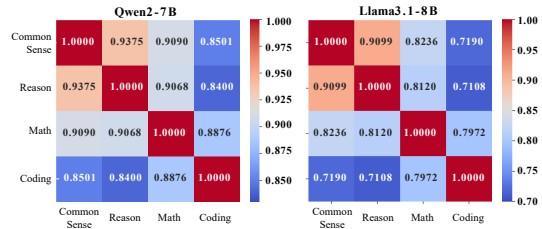

Figure 3: Semantic cosine similarity across different domains for generated samples of size 10.

**Impacts of Hyper-Parameters** In Appendix G.2-Appendix G.7, we conduct a series of ablation studies, including (1) *selection of layer*: the discrimination accuracy derived from layer 1-10 fluctuates in **91.2-93.6%**; (2) *the number of seed*: the similarity between centroids remains highly consistent with variance ranging **from 5.4E-7 to 1.8E-5**; (3) *the number of augmented data*: the mean similarity of centroids generated from sizes of 10 and 30 is **0.983** and **0.981** on Qwen2.5 and Llama3.1 respectively; (4) *the size of sliding window & similarity threshold*: the impact of different hyper-parameters on centroid generation is **minimal**, ensuring the subsequent training **stability** of DAAR; (5) *diversity threshold*, where varying thresholds minimally affect centroid-guided clustering (with distinct generation attempt frequencies across thresholds) and empirically balanced thresholds are determined for different domains; (6) *generalizability across model scales*, where DAAR shows consistent improvements in representative scenario on both **Llama3.2-3B** and **Qwen2.5-14B** models.

**Cost-Efficiency and Flexibility** As detailed in Appendix G.9, our method demonstrates significant computational efficiency compared to baselines that rely on GPT-based evaluators or full-LLM inference. By operating on frozen hidden layers and generating regression scores instead of full vocabulary, our approach **achieves 70% lower GPU usage** and **2.5x faster inference**.

# 6 Conclusion, Limitation & Future Works

In this paper, we propose a new approach to fine-tuning LLMs with data that lacks clear domain labels, using diversity as a guiding principle. By measuring semantic diversity with entropy, we employ a self-reward mechanism built upon the given LLM, identifying data that best fits the model's natural tendencies in terms of its underlying knowledge distribution. Our experiments with various SOTA LLMs show notable superiority of the method over competitive baselines, highlighting the potential of data diversity to enhance model overall performance.

The implementation of DAAR has limitations that can be addressed further, such as the customization of the selection ratios to support broader application in real-world scenarios. Besides, DAAR has potential to efficiently adjust diversity measures, benefiting LLMs in self-evolving towards artificial general intelligence in dynamic environments [35].

## Acknowledgment

This research is supported by Key-Area Research and Development Program of Guangdong Province (Granted No. 2024B1111060004), Guangdong Provincial Special Funds for Promoting High Quality Economic Development (Marine Economic Development) in Six Major Marine Industries (Granted No. GDNRC[2024]52), and in part by the New Generation Artificial Intelligence-National Science and Technology Major Project (Granted No. 2025ZD0123003).

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

# Appendix

**Table of Contents**

# A  Notation

For ease of reading and reference, we present the mathematical symbols used in this paper in Table 5.

Table 5: Symbol Notation

| Symbol | Description | Symbol | Description |
|---|---|---|---|
| $\mathcal{D}$ | Composite dataset | $\mathcal{D}_k$ | Distinct domain dataset |
| $x_i^{(k)}$ | Raw $i$-th data sample in domain-$k$ | $\mathbf{x}_i^{(k)}$ | Embedded $i$-th data sample in domain-$k$ |
| $N_k$ | Number of $\mathcal{D}_k$ data samples | $\mathcal{C}_k$ | Domain centroid of $k$ |
| $\phi_{\text{inter}}$ | Sample-level cross-domain similarity | $\Phi_{\text{inter}}$ | Global inter-diversity |
| $\phi_{\text{intra}}$ | Sample similarity to its domain centroid | $\Phi_{\text{intra}}^{(k)}$ | Domain-level variance of domain-$k$ |
| $l$ | Loss function | $p, q$ | Data distributions |
| $\mathcal{S}_k^{(0)}$ | Generated centroids' seed sample | $\mathcal{S}_k^{(t)}$ | Generated centroid sample at $t$-th round |
| $\mathcal{M}_{\text{ebd}}$ | Embedding layer of LLM | $\mathcal{M}$ | Hidden layer of LLM |
| $\mathcal{D}_{\text{probe}}$ | DAAR training set | $\tau$ | A similarity threshold |
| $\tilde{x}_i$ | Data sample in $\mathcal{D}_{\text{probe}}$ | $\tilde{y}_i$ | Pseudo-label of $\tilde{x}_i$ |
| $\psi_{\text{dom}}$ | A MLP-based domain predictor | $\psi_{\text{div}}$ | A MLP-based entropy predictor |
| $p_k(\tilde{x})$ | Domain predicted probability by $\psi_{\text{dom}}$ | $\hat{H}(\mathbf{x})$ | Predictive entropy by $\psi_{\text{dom}}$ |
| $\mathcal{L}_{\text{dom}}$ | Cross-entropy loss of $\psi_{\text{dom}}$ | $\mathcal{L}_{\text{div}}$ | MSE loss of $\psi_{\text{dom}}$ |

# B  Implementation Details of DAAR

## B.1  Hyper-Parameters

For the generation of synthetic data, we utilize the ***model.generate*** function from the PyTorch library. Key parameters included setting ***max-new-tokens*** to ***2048*** to control the length of the output, enabling ***do-sample*** to ***True*** to allow sampling, and configuring ***top-p*** to ***0.95*** and ***temperature*** to ***0.9*** to ensure diversity in the generated content. During the content extraction phase, we employ ***regular expressions*** to efficiently extract and structure the desired information.

For implementation details of DAAR's hyper-parameters, we set the number of seed samples $\mathcal{S}_k^{(0)}$ as 5 (ablation in Appendix G.3), the number of anchors in sliding window as 3 (ablation in Appendix G.5), the similarity threshold $\tau$ as 0.9 for Mathematics and 0.85 for others (ablation in Appendix G.6), the terminated iteration as 30 (ablation in Appendix G.4), the hidden layer $\mathcal{M}(\tilde{x})$ as layer-3 (ablation in Appendix G.2).

## B.2  Implementation of Embedding

Given the Alpaca-based [43] data format employed in our study, we concatenate the 'instruction', 'input', and 'output' components of each data sample to capture domain-aware semantic representations. These concatenated sequences are then fed into the LLM, where we compute the average of all token embeddings extracted from the embedding layer to derive the semantic vector characterizing each sample.

The embedding layer is deliberately selected due to its shallow architectural position: (1) it preserves precise semantic information capture through raw input representations, and (2) it incurs reduced computational overhead compared to deeper layers. As evidenced by the visualization in Fig. 13

(Appendix G.2), this approach demonstrates robust domain discrimination capabilities across different LLM layers, thereby validating the robustness of our methodology.

### B.3    The Phase of Model-Aware Training Data Generation

**Seed Generation**    This phase aims to provide zero-shot LLM-generated seed data that reflects how LLMs inherently perceive the characteristics of domain-specific data. We **employ the LLM itself to generate concise domain descriptions** across diverse fields illustrated in Prompt 1, and the prompt shown in Prompt 2. Due to the difficulty of pre-trained LLMs in providing accurate and coherent responses, we utilize their corresponding Instruct versions for centroid data generation. Specifically, we apply Llama3.1-8B-Instruct for Llama3.1-8B, Qwen2-7B-Instruct for Qwen2-7B, and Qwen2.5-7B-Instruct for Qwen2.5-7B.

Notably, these **descriptions are not required to be optimal or prescriptive**, rather, they serve as representative examples compatible with our experimental scenarios. Users retain the flexibility to customize target domain specifications according to specific requirements. The associated stability analysis regarding this design choice will be elaborated in Appendix G.10.

**Diversity Augmentation**    While seed data effectively captures the intrinsic domain knowledge of LLMs, its limited quantity and the constrained diversity of single-prompt generation methods necessitate enhancement. We address these limitations through two complementary strategies including a sliding window mechanism and a rule-based filter. The implementation details are formalized in Prompt 3 and Eq. (7).

**Domain-Aware Clustering**    During reward model training data clustering, we rigorously curate a subset of 5,000 entries that are distribution-matched to the fine-tuning dataset while separate from it.

---

**Prompt 1: Domain's Description**

**Common Sense**: Common sense generally includes a knowledge-based question and its corresponding answer, without reasoning.

**Reasoning**: Reasoning involves the ability to think logically about a situation or problem, to draw conclusions from available information, and to apply knowledge in new situations.

**Mathematics**: Mathematical skills include the ability to perform calculations, understand mathematical concepts, solve hard and professional math problems, and apply mathematical reasoning.

**Coding**: Design and generate specific code programs, or apply algorithms and data structures, with code generation in the Output.

---

**Prompt 2: Seed Generation Prompt**

You are an AI model with expertise in {selected_domain}. Here's a brief description of this domain: {Prompt 1}

Generate 5 different instruction pairs related to this field with various lengths. Maintain the index format: Instruction [1 ... 5].

The response should include three parts:
1. Instruction: A clear command or question that can be understood by the assistant.

2. Input: Any information provided to help it understand the instruction. If there is no need to generate, just keep it empty.

3. Output: The expected answer or action.

Keep the generated content focused on {selected_domain}. And do not involve {unselected_domains} related knowledge.

---

> **Prompt 3: Training Data Diversity Augmentation**
>
> You are an AI model with expertise in {selected_domain}. Here's a brief description of this domain:{Prompt 1}
> Generate only an instruction pair related to this field. The response should include three parts:
>
> Instruction: A clear command or question that can be understood by the assistant.
> Input: Any information provided to help it understand the instruction. If there is no need to generate, just keep empty.
> Output: The expected answer or action.
>
> Keep the generated content focused on {selected_domain}. Do not involve {unselected_domain} related knowledge.
>
> Note that you should generate content strongly unrelated and different to these examples to ensure diversity in the generated output:
> Counterexample: {}
>
> The format of the generated content should be: Instruction: [], Input: [], Output: [].

## C Supplementary Theoretical Analysis

This appendix provides additional details on the theoretical foundation of our work, as discussed in Section 3.4. We first present the full derivation of the importance weight decomposition (Proposition 3.2) and then offer an analysis justifying our entropy-based selection strategy.

### C.1 Derivation of Proposition 3.2

We derive the density ratio by systematically expanding and substituting terms:

$$\frac{q(\mathbf{x})}{p(\mathbf{x})} = \frac{\sum_{k=1}^{K} q(c=k)q(\mathbf{x}|c=k)}{p(\mathbf{x})} \tag{10}$$

Expand $q(\mathbf{x})$ via the law of total probability over domains.

$$= \frac{\sum_{k=1}^{K} q(c=k)p(\mathbf{x}|c=k)}{p(\mathbf{x})} \tag{11}$$

Apply Assumption 3.1 ($q(\mathbf{x}|c=k) = p(\mathbf{x}|c=k)$) to unify domain-conditional densities.

$$= \frac{\sum_{k=1}^{K} q(c=k)\frac{p(c=k|\mathbf{x})p(\mathbf{x})}{p(c=k)}}{p(\mathbf{x})} \tag{12}$$

Express $p(\mathbf{x}|c=k)$ via Bayes' rule inversion: $p(\mathbf{x}|c=k) = \frac{p(c=k|\mathbf{x})p(\mathbf{x})}{p(c=k)}$.

$$= \sum_{k=1}^{K} \frac{q(c=k)}{p(c=k)} p(c=k|\mathbf{x}) \cdot \frac{p(\mathbf{x})}{p(\mathbf{x})} \tag{13}$$

Factor out $p(\mathbf{x})$ from the numerator and the denominator.

$$= \sum_{k=1}^{K} \underbrace{\frac{q(c=k)}{p(c=k)}}_{\lambda_k} p(c=k|\mathbf{x}) \tag{14}$$

Define domain proportion ratio $\lambda_k := \frac{q(c=k)}{p(c=k)}$.

This demonstrates that the density ratio decomposes linearly into domain-specific components weighted by their proportion shifts ($\lambda_k$) and posterior probabilities ($p(c=k|\mathbf{x})$).

## C.2  Supplementary Analysis of the Theoretical Foundation

Our theoretical framework motivates a data selection strategy that privileges high-entropy samples. Here, we theoretically analyze why this is not merely a heuristic to move away from a known suboptimal point, but rather a principled approach to minimize approximation error.

**Approximation Error of Deterministic Assignment**   As shown in Section 3.4, a simplified selection strategy might rely on a deterministic domain assignment, where only the most likely domain $k^*(x) = \arg\max_k p(c = k|x)$ is considered. This reduces the true importance weight $w(x) = \sum_{k=1}^{K} \lambda_k p(c = k|x)$ to an approximation $w_{\text{approx}}(x) = \lambda_{k^*(x)}$. We analyze the squared error of this approximation, $\text{Err}(x) = (w(x) - w_{\text{approx}}(x))^2$.

**Proposition C.1.** *The squared approximation error Err(x) can be measured by:*

$$Err(x) = \left( \sum_{j \neq k^*(x)} (\lambda_j - \lambda_{k^*(x)}) \cdot p(c = j|x) \right)^2 \tag{15}$$

**Implication for Data Selection**   This proposition reveals that the approximation error is a direct function of the probability mass distributed across non-dominant domains. The error is zero only if $p(c = k^*(x)|x) = 1$, which corresponds to zero conditional entropy, $H(C|X = x) = 0$. Conversely, the error is maximized precisely when the probability is spread across multiple domains—the very definition of high predictive entropy.

Therefore, our strategy of selecting high-entropy samples is a principled approach to preferentially select samples where the deterministic approximation is most erroneous. These are exactly the samples for which the full diversity information encoded in $p(c|x)$ is most critical for an accurate estimation of the true importance weight $w(x)$. Our method, DAAR, employs a dedicated *domain discrimination probe* to model $p(c|x)$ and uses its predictive entropy as a computationally feasible proxy to identify these high-error, high-value samples for training.

# D  Experimental Setup Details

## D.1  Construction of Data Pool

To investigate data selection from large data pools and its impact on the mixture of downstream tasks, we construct a data pool with distinct properties to mimic practical settings. We select the following datasets to evaluate specific abilities:

- *Common Sense*: **Dolly-15K** [15] with 15,011 samples, an open source dataset of instruction-following records generated by thousands of Databricks employees in several of the behavioral categories.
- *Reasoning*: **Cot-en** [13] with 74,771 samples, is created by formatting and combining nine CoT datasets released by FLAN.
- *Mathematics*: **Math-Instruct** [56] with 262,039 samples, is compiled from 13 math rationale datasets, six of which are newly curated by this work. It uniquely ensures extensive coverage of diverse mathematical fields.
- *Coding*: **Code-Alpaca** [6] with 20,016 samples, is constructed from real-world code examples, providing a rich set of tasks designed to guide models in generating accurate and functional code.

Each dataset was initially filtered and randomly reduced to 10,000 entries, resulting in a combined data pool of 40,000 entries. Specifically, for the Math-Instruct dataset, due to its inclusion of CoT and certain coding capabilities, we extract a highly mathematics-related subset and use regular expressions to filter out the coding-related content (including 'program', 'python', 'def', 'import', 'print', 'return'), ensuring it remains within the domain of mathematics.

## D.2  Benchmarks

To evaluate the models' true capabilities and performance across different domains, we follow the approach of major open-source LLMs (e.g., the Llama3 series [18] and Qwen2 series [53]) and select

the following widely used evaluation sets. All evaluations were conducted on the OpenCompass platform[4].

- **Common Sense**: **NQ** [28] and **TriviaQA** [26], which cover factual knowledge-based questions of varying difficulty.

- **Reasoning**: **HellaSwag** [57], which effectively evaluates the model's comprehensive reasoning ability.

- **Mathematics**: **GSM8K** [14] and **MATH** [24] benchmarks, which encompass problems ranging from elementary to competition-level difficulty.

- **Coding**: **MBPP** [4] and **HumanEval** [11], which include evaluations of basic coding abilities in Python. We use the average of various metrics to demonstrate the models' overall performance across different domains.

Considering the numerous evaluation tasks, utilizing the complete evaluation set would result in significant time expenditure. To accelerate the evaluation process while maintaining fairness and accuracy, we randomly tailor the original evaluation sets into evaluation subsets, as detailed in Table 6. *All experiments were conducted using this consistent setup* to ensure the fairness of the experiments.

Table 6: Number of samples in various evaluation benchmarks' datasets.

| Number of Samples | Common Sense | | Reasoning | Mathematics | | Coding | |
|---|---|---|---|---|---|---|---|
| | NQ | Triviaqa | Hellaswag | GSM8K | MATH | MBPP | HumanEval |
| **Original** | 3,610 | 8,837 | 10,042 | 1,319 | 5,000 | 974 | 164 |
| **Utilized** | 3,610 | 5,000 | 10,042 | 500 | 1,000 | 500 | 164 |

### D.3  Training and Evaluation Details

**Platform**   We implement our approaches using PyTorch [36] v2.4.1, coupled with PEFT v0.12.0 and the Transformers library [47] v4.45.2. Experiments are conducted on a computing platform equipped with four NVIDIA A100 GPUs (40GB), with pre-trained LLMs loaded as 16-bit floating-point numbers. The specific data-model development processes are completed in Data-Juicer Sandbox [9, 8], via integration with the ms-swift [61] training repository, and the OpenCompass [16] evaluation repository.

**Training Details**   In our experimental setup, we employ Low-Rank Adaptation (LoRA) [25] adapters for the fine-tuning process, utilizing a LoRA-rank of 8 and a LoRA-alpha of 16. The learning rate was consistently maintained at $5 \times 10^{-5}$ across all experiments to ensure uniformity in training dynamics. We utilize a batch size of 4 and set the maximum sequence length to 2048 tokens to accommodate the model's capacity. To optimize the training process, a warmup ratio of 0.05 was applied, and a validation ratio of 0.03 was used. The training was conducted over a single epoch, balancing computational efficiency with the need for effective model adaptation. Following some effective instruction-tuning work [62, 34], we set the size of our subset to 8,000 entries, which constitutes 20% of the data pool.

**Evaluation Details**   Following the guidelines provided by OpenCompass [16], we adhered to the default settings for our evaluation process. We select the hf-type as *base* and utilize a batch size of *16* to ensure efficient processing. For most tasks, we employ the *gen* mode, while for the Hellaswag task, we opt for the *ppl* mode to better assess perplexity.

---

[4]https://opencompass.org.cn/

# E  Details of Baselines

## E.1  Implementation of Baselines

We select the following representative methods and works related to data selection as our baselines to evaluate their performance in the context of a mixture of downstream tasks.

**RANDOM SELECTION(RAND)** A random selection of 8,000 data samples from the data pool was made, which to some extent reflects the distribution characteristics of the original data pool.

**INSTRUCTION LENGTH (IL)** The length of the instruction can be considered a measure of input complexity. It is widely believed [5, 60] that more complex data is beneficial for enhancing model capabilities. Therefore, we select a subset of 8,000 entries with the maximum number of words (based on spaces) in the concatenation of Instruction and Input as part of the filtering process.

**ALPAGASUS [10]** Using prompts to directly score and annotate the quality of the data leverages the powerful cognitive abilities of LLMs for evaluation and selection. Based on the original work, we use the GPT-3.5-Turbo API to score the data with the following prompt:

---

**Prompt 4: Implementation of ALPAGASUS**

**System Prompt:**
We would like to request your feedback on the performance of AI assistant in response to the instruction and the given input displayed following.

Instruction: [Instruction]
Input: [Input]
Response: [Response]

**User Prompt:**
Please rate according to the accuracy of the response to the instruction and the input. Each assistant receives a score on a scale of 0 to 5, where a higher score indicates a higher level of accuracy. The generated scores can be precise to decimal points, not just in increments of 0.5. Please first output a single line containing the value indicating the scores. In the subsequent line, please provide a comprehensive explanation of your evaluation, avoiding any potential bias.

---

The distribution of direct ratings for the 40,000 data pool is shown in Fig. 4. Although the original paper's prompts were strictly followed and efforts were made to minimize potential bias, most scores still clustered around 5. Since we require a uniform selection of 8,000 data samples, we randomly select the subset with a rating of 5 to serve as the baseline data.

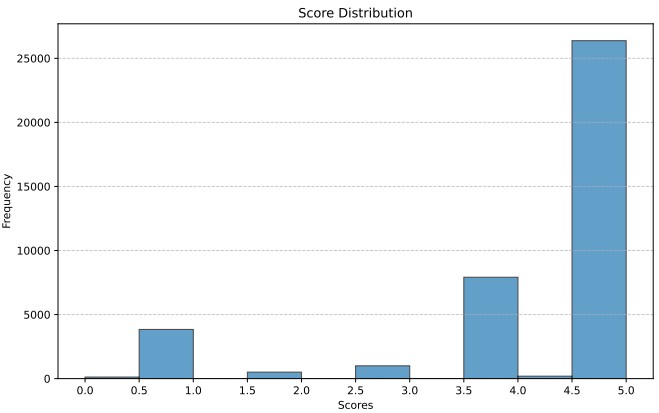

Figure 4: The distribution of the score by ALPAGASUS.

**INSTAG [34]** first utilizes ChatGPT to tag the samples based on semantics and intentions, then trains a LLaMA-based tagger on the ChatGPT tags to tag data. They use the number of tags as a proxy for complexity. We directly use ChatGPT-3.5-Turbo as a tagger to achieve better performance. Following the original paper, the prompt is as follows.

---

**Prompt 5: Implementation of INSTAG COMPLEXITY**

**System Prompt:**
You are a tagging system that provides useful tags for instruction intentions to distinguish instructions for a helpful AI assistant. Below is an instruction:

[begin]
{Instruction + Input}
[end]
**User Prompt:**
Please provide coarse-grained tags, such as "Spelling and Grammar Check" and "Cosplay", to identify main intentions of above instruction. Your answer should be a list including titles of tags and a brief explanation of each tag. Your response have to strictly follow this JSON format: ["tag": str, "explanation": str]. Please response in English.

---

A total of 19,585 tags were assigned across 40,000 data samples, with the distribution shown below. Subsequently, based on the procedures outlined in the original text, tags were deduplicated as follows: 1) filter out long-tail tags that appear fewer than $\alpha$ times in the entire annotated dataset, and 2) transform all tags to lowercase to mitigate the influence of capitalization.

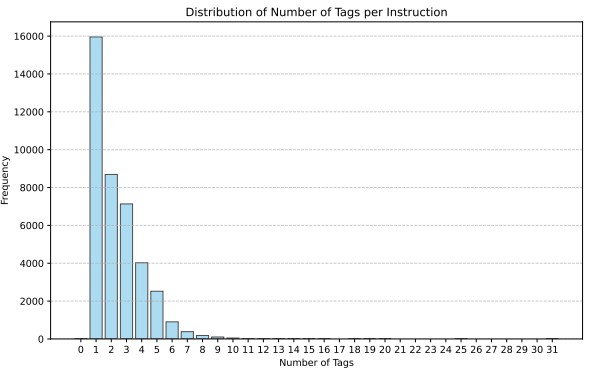

Figure 5: The distribution of the tags by INSTAG.

- **INSTAG COMPLEXITY (INSTAG-C)**: To reduce redundancy, we apply a threshold of $\alpha = 5$, resulting in a set of 1,948 valid tags. Following the definition of Complexity, we select the top 8,000 entries with the highest tag counts. Specifically, there are 8,211 entries with more than three tags, so we include all records with more than four tags and randomly supplement from those with exactly four tags until reaching a total of 8,000 entries.

- **INSTAG DIVERSITY (INSTAG-D)**: For Diversity, we use $\alpha = 1$ to reduce redundancy. The algorithm employed involves sorting all data in descending order based on the number of tags, and prioritizing records with more tags. To ensure dataset diversity, a tag is only added to the final dataset if it increases the overall size of the tag set. This approach captures a broader range of unique tags, thereby enhancing the diversity and representativeness of the dataset.

**SUPERFILTER [29]** introduces a method that astonishingly utilizes a small GPT2 model to successfully filter out the high-quality subset from the existing GPT4-generated instruction tuning dataset. The core concept is the instruction-following difficulty (IFD) score. By meticulously following this methodology, we utilize GPT-2 to select 8,000 data samples.

**DEITA [33]** is an open-sourced project designed to facilitate Automatic Data Selection for instruction tuning in LLMs. It delves into the relationships between data complexity, quality, and diversity, and develops a series of methodologies. Specifically, we utilize it as the following three baselines:

- **DEITA COMPLEXITY (DEITA-C)**: Enhances instruction complexity by evolving examples with techniques like adding constraints. ChatGPT ranks these evolved samples for complexity, and the scores train a LLaMA-1-7B model to predict complexity in new instructions, refining complexity assessment. We utilize its LLaMA-1-7B-based complexity-scorer to evaluate the data pool and select the top 20% of them to reach 8,000 entries.

- **DEITA QUALITY (DEITA-Q)**: Enhances response quality by prompting ChatGPT to iteratively improve helpfulness, relevance, depth, creativity, and detail. After five iterations, ChatGPT ranks and scores the responses, providing nuanced quality distinctions. These scores train an LLaMA-1 7B model to predict quality scores for new instruction-response pairs. We utilize its LLaMA-1-7B-based quality scorer to evaluate the data pool and select the top 20% of them to reach 8,000 entries.

- **DEITA DEITA (DEITA-D)**: Data-Efficient Instruction Tuning for Alignment, it selects data by combining complexity and quality into an evol score, prioritizing higher scores. It uses the REPR FILTER to iteratively add samples, ensuring diversity and avoiding redundancy, resulting in a balanced dataset of complexity, quality, and diversity. Following its methodology, we gain 8,000 entries as a baseline.

### E.2 Visualization of Baseline's Selected Data

We project all baseline-selected data samples through Llama3.1-8B's, Qwen2-7B's and Qwen2.5-7B's embedding layer and visualize their distributions via t-SNE dimensionality reduction in Fig. 6, Fig. 7, and Fig. 8, respectively. This visualization provides intuitive insights into how different data selection strategies handle domain-mixed data. Three **key observations and conjectures** emerge:

- Certain methods (particularly INSTRUCTION LENGTH, DEITA-C, and DEITA-Q) demonstrate pronounced **distributional skew** toward coding-domain samples, which may partially explain their suboptimal HumanEval performance shown in Table 31.

- The domain-agnostic nature of baseline approaches reveals **fundamental limitations** in preserving original data distributions when processing domain-unspecified samples, as evidenced by the visualization.

- Although distributional shifts are evident, **the absence of** commensurate performance deterioration across all methods empirically validates our theoretical framework (Section 3.4), which establishes connections between data distribution shift.

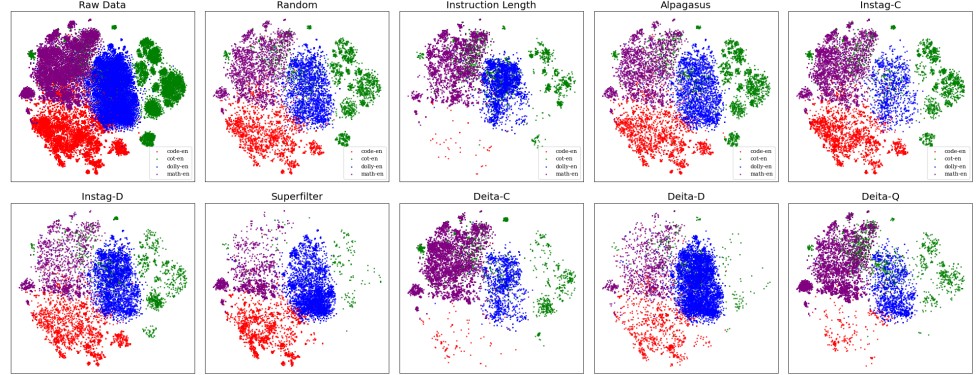

Figure 6: t-SNE visualization of data samples selected by different baselines using Llama3.1-8B embeddings. While original labels were removed during training, we preserve them for interpretability.

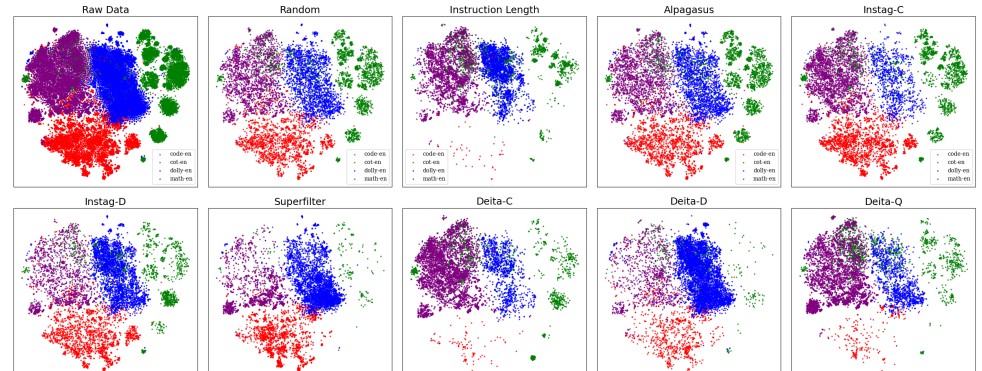

Figure 7: t-SNE visualization of data samples selected by different baselines using Qwen2-7B embeddings. While original labels were removed during training, we preserve them for interpretability.

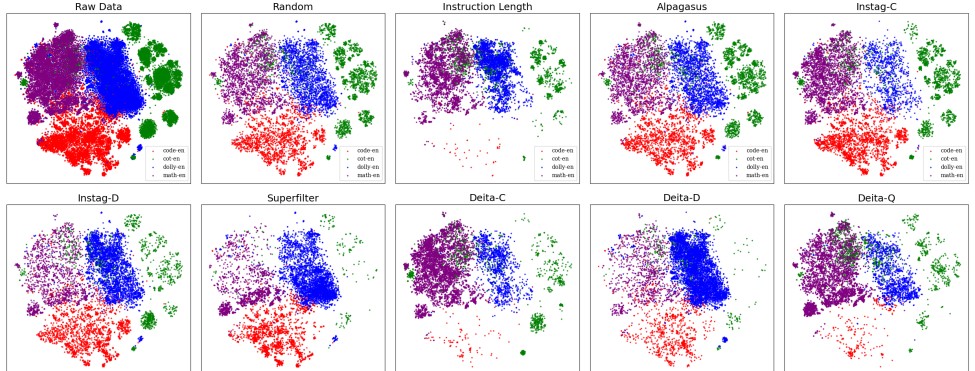

Figure 8: t-SNE visualization of data samples selected by different baselines using Qwen2.5-7B embeddings. While original labels were removed during training, we preserve them for interpretability.

# F Comprehensive Experiments

## F.1 Performance on Comprehensive Benchmark MMLU

The benchmarks selected in Section 3.1 are specifically tailored to correspond with the domain-specific capabilities of the data sources. To comprehensively evaluate the performance of DAAR across broader assessment metrics and more extensive benchmarking frameworks, we adopted MMLU as an out-of-distribution (OOD) evaluation protocol, detailed description as:

- MMLU is a benchmark designed to evaluate the multitask capabilities of language models across diverse subjects, covering 57 tasks spanning topics with questions ranging from elementary to professional levels.

Table 7: Evaluation results of Qwen2-7B and Qwen2.5-7B on MMLU.

| Models | Methods | Humanities | STEM | Social Science | Other | MMLU-AVG |
|---|---|---|---|---|---|---|
| Qwen2-7B | RAW | 38.58 | 25.77 | 24.94 | 27.82 | 28.98 |
| | RANDOM | 70.02 | 58.54 | 80.00 | 72.26 | 68.81 |
| | INSTRUCTION LEN | 71.34 | 58.09 | 79.77 | 70.71 | 68.55 |
| | ALPAGASUS [10] | 56.55 | 49.10 | 71.88 | 65.54 | 59.34 |
| | INSTAG-C [34] | 70.28 | 58.69 | 79.92 | 71.17 | 68.65 |
| | INSTAG-D [34] | 67.87 | 58.35 | 79.82 | 71.61 | 68.07 |
| | SUPERFILTER [29] | 52.55 | 49.16 | 57.99 | 60.01 | 54.27 |
| | DEITA-C [33] | 68.05 | 58.53 | 80.12 | 70.97 | 68.08 |
| | DEITA-D [33] | 67.11 | 58.22 | 79.17 | 72.13 | 67.83 |
| | DEITA-Q [33] | 56.79 | 52.82 | 69.01 | 63.06 | 59.47 |
| | DAAR (Ours) | 70.49 | 59.94 | 79.67 | 72.74 | **69.42** |
| Qwen2.5-7B | RAW | 73.41 | 48.96 | 72.79 | 58.46 | 61.72 |
| | RANDOM | 75.40 | 64.47 | 81.46 | 72.71 | 72.42 |
| | INSTRUCTION LEN | 75.08 | 63.74 | 81.40 | 72.96 | 72.15 |
| | ALPAGASUS [10] | 75.39 | 64.85 | 81.73 | 73.13 | 72.70 |
| | INSTAG-C [34] | 75.26 | 64.35 | 81.64 | 73.03 | 72.46 |
| | INSTAG-D [34] | 74.83 | 65.05 | 81.48 | 73.16 | 72.59 |
| | SUPERFILTER [29] | 75.24 | 66.34 | 82.19 | 73.99 | **73.45** |
| | DEITA-C [33] | 75.76 | 64.18 | 81.59 | 72.83 | 72.46 |
| | DEITA-D [33] | 75.27 | 65.60 | 81.55 | 72.99 | 72.85 |
| | DEITA-Q [33] | 75.81 | 64.14 | 81.41 | 73.65 | 72.61 |
| | DAAR (Ours) | 74.91 | 65.84 | 81.85 | 73.68 | 73.07 |

As demonstrated in Table 7, despite not being explicitly optimized for broader capabilities during its design phase, DAAR still demonstrates competitive performance in comprehensive capability assessments. The method achieved the highest score on Qwen2-7B, surpassing the second-place Random selection by **0.61 accuracy points**. On Qwen2.5-7B, it exhibited performance closely following SuperFilter, thereby confirming the generalization potential of DAAR to a certain extent.

**Notably,** Llama3.1-8B was excluded from reference comparisons due to significant result fluctuations observed during Opencompass evaluations (as shown in Table 8 that the results range from 0.04 to 42.52). We further identified inconsistent failures in model prediction generation during the evaluation process, which compromised the reliability of performance measurement. These implementation challenges **prevented valid comparative analysis** with Llama3.1-8B in our experiments.

Table 8: Invalid evaluation results of Llama3.1-8B on MMLU.

| Models | Methods | Humanities | STEM | Social Science | Other | MMLU-AVG |
|---|---|---|---|---|---|---|
| Llama3.1-8B | RAW | 0.02 | 0.01 | 0.08 | 0.07 | 0.04 |
| | RANDOM | 30.13 | 15.34 | 30.73 | 27.51 | 24.73 |
| | INSTRUCTION LEN | 23.38 | 11.65 | 20.43 | 18.26 | 17.68 |
| | ALPAGASUS [10] | 35.99 | 23.28 | 36.80 | 34.37 | 31.56 |
| | INSTAG-C [34] | 25.87 | 12.58 | 33.73 | 23.71 | 22.60 |
| | INSTAG-D [34] | 42.50 | 19.44 | 47.70 | 37.19 | 34.70 |
| | SUPERFILTER [29] | 11.61 | 9.40 | 12.11 | 11.71 | 11.00 |
| | DEITA-C [33] | 8.79 | 0.97 | 0.60 | 2.27 | 2.97 |
| | DEITA-D [33] | 50.79 | 29.76 | 53.04 | 43.21 | 42.52 |
| | DEITA-Q [33] | 33.99 | 23.36 | 26.31 | 24.27 | 26.61 |
| | DAAR (Ours) | 45.18 | 20.94 | 46.47 | 34.20 | 34.87 |

## F.2 Validation on Qwen3-8B

As a prevailing trend in the development of current LLMs, the **RL-based** (reinforcement learning) **thinking model** has demonstrated significant capabilities and potential (e.g. DeepSeek series [31], Llama4 series and Qwen3 series). To validate the efficacy of DAAR on the most up-to-date and advanced LLMs, we conducted a series of verification experiments using Qwen3-8B [5]. ***It should be noted that***, since Qwen3 does not adhere to conventional SFT paradigms, as well as the comprehensive support for training & evaluation of the Qwen3 series remains incomplete, the experiments presented in this section primarily serve as exploratory investigations. A comprehensive analysis of data diversity across various training methodologies and LLM architecture will be reserved for **future research endeavors**.

**Observation Studies on Qwen3**  Following the methodology delineated in Section 3, we employed Qwen3's embedding layer to process the identical data pool. The t-SNE visualization of Qwen3's embeddings (Fig. 9) demonstrates comparable domain discrimination capabilities.

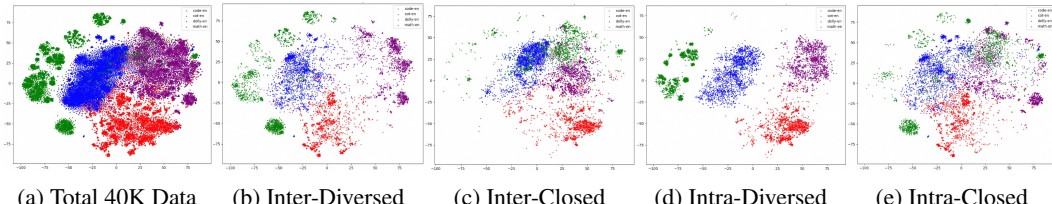

| (a) Total 40K Data | (b) Inter-Diversed | (c) Inter-Closed | (d) Intra-Diversed | (e) Intra-Closed |

Figure 9: The t-SNE visualization of embeddings for data samples with different distributions on Qwen3-8B. (a) The data pool of all 40K samples, (b) Distribution of data farthest from other domain centroids on Inter-Diversity, (c) Distribution of data closest to other domain centroids on Inter-Diversity, (d) Distribution of data closest to its own domain centroid on Inter-Diversity, (e) Distribution of data farthest from its own domain centroid on Inter-Diversity.

Subsequently, we processed the curated dataset through inter-diversity and intra-diversity filters, followed by training Qwen3 on artificially constructed data distributions. As evidenced in Table 9, the experimental outcomes reveal that while the performance patterns diverge from those observed in Llama 3.1, Qwen2, and Qwen2.5, **varying diversity levels demonstrably influence comprehensive model performance**. Notably, the absolute value of the comprehensive performance of Qwen3 is even inferior to that of Qwen2.5 and Qwen2. We posit that this phenomenon is primarily due to the incomplete support of the evaluation platform OpenCompass (as of the submission date), where the distinctive thinking and reasoning capabilities of Qwen3 have not been fully leveraged. However, the **relative comparison** of different processes within the same data pool remains highly informative.

Table 9: Validation results of Inter-Diversity and Intra-Diversity on Qwen3-8B across benchmarks.

| Qwen3-8B | Common Sense | | Reasoning | Mathematics | | Coding | | Avg |
|---|---|---|---|---|---|---|---|---|
| | NQ | TriviaQA | Hellaswag | GSM8K | MATH | MBPP | HumanEval | |
| **RAW** | 9.64 | 59.90 | 71.56 | 85.60 | 23.60 | 22.40 | 73.17 | 49.41 |
| **RAND (8K)** | 12.24 | 59.22 | 71.71 | 83.00 | 25.2 | 12.6 | 73.17 | 48.16 |
| Inter-Diversity (0-20) | 12.94 | 60.28 | 71.79 | 86.20 | 23.80 | 11.80 | 75.61 | 48.92 |
| Inter-Diversity (40-60) | 12.71 | 59.76 | 71.71 | 83.20 | 28.40 | 14.40 | 73.78 | 49.14 |
| **Inter-Diversity (80-100)** | 11.36 | 59.52 | 71.61 | 83.00 | 25.40 | 14.20 | 68.29 | 47.63 |
| Intra-Diversity (0-20) | 12.33 | 59.92 | 71.58 | 84.00 | 28.40 | 19.40 | 71.34 | 49.57 |
| Intra-Diversity (40-60) | 13.10 | 59.42 | 71.64 | 82.40 | 21.00 | 13.20 | 72.56 | 47.62 |
| **Intra-Diversity (80-100)** | 12.58 | 60.06 | 71.88 | 84.60 | 34.90 | 9.40 | 73.17 | 49.51 |

**DAAR Training on Qwen3**  Subsequently, we employed DAAR to conduct diversity probe training on Qwen3. Consistent with observations in other models, the two-stage MLP training framework demonstrated robust stability and convergence, shown in Fig. 10.

---

[5]https://github.com/QwenLM/Qwen3

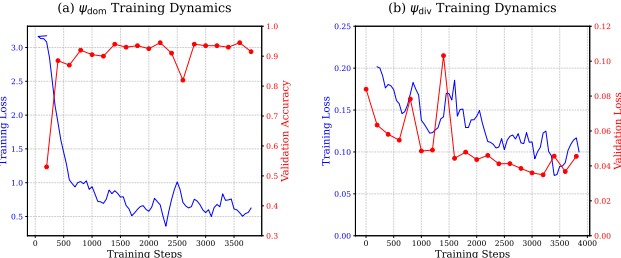

Figure 10: Training loss and validation process of the two training stages of DAAR on Qwen3-8B, showing the model gradually converging.

**Evaluation Results of DAAR with Baselines**  Following the identical experimental setup outlined in Appendix D, we conducted experiments on Qwen3, with the results presented in Table 10. As demonstrated, DAAR achieved the top comprehensive score on Qwen3, surpassing all baselines by up to **4.7%**. Notably, its performance on the MATH benchmark was particularly outstanding, exceeding the average of baselines by **25.91%**. The comparative experiments affirm that DAAR maintains competitive effectiveness even on the SOTA LLM Qwen3-8B.

Table 10: Performance of DAAR with baselines on Qwen3-8B across various benchmarks.

| Qwen3-8B | Common Sense | | Reasoning | Mathematics | | Coding | | Avg |
|---|---|---|---|---|---|---|---|---|
| | NQ | TriviaQA | Hellaswag | GSM8K | MATH | MBPP | HumanEval | |
| RAW | 9.64 | 59.90 | 71.56 | 85.60 | 23.60 | 22.40 | 73.17 | 49.41 |
| RAND (8K) | 12.24 | 59.22 | 71.71 | 83.00 | 25.20 | 12.60 | 73.17 | 48.16 |
| INSTRUCTION LEN | 11.66 | 59.56 | 71.67 | 83.80 | 26.30 | 18.20 | 73.17 | 49.19 |
| ALPAGASUS [10] | 12.52 | 59.34 | 71.69 | 83.80 | 26.60 | 11.20 | 69.51 | 47.81 |
| INSTAG-C [34] | 11.91 | 59.62 | 71.55 | 83.00 | 26.40 | 14.60 | 73.78 | 48.69 |
| INSTAG-D [34] | 11.99 | 59.48 | 71.59 | 87.20 | 25.30 | 13.60 | 71.34 | 48.64 |
| SUPERFILTER [29] | 15.73 | 60.18 | 71.69 | 84.80 | 30.90 | 16.20 | 67.07 | 49.51 |
| DEITA-C [33] | 12.55 | 59.62 | 71.78 | 83.60 | 28.60 | 13.20 | 77.44 | 49.54 |
| DEITA-Q [33] | 12.88 | 59.72 | 71.64 | 85.40 | 32.90 | 13.00 | 73.78 | 49.90 |
| DEITA-D [33] | 13.88 | 59.98 | 71.50 | 83.60 | 26.30 | 14.00 | 70.73 | 48.57 |
| DAAR (Ours) | 13.68 | 60.14 | 71.56 | 83.40 | 34.50 | 14.60 | 72.56 | **50.06** |

## F.3  Customized Data Selection Capability of DAAR

A distinctive characteristic of our method stems from its inherent capability to perform **domain-specific data selection** under a domain-undetermined scenario, thereby naturally supporting **customized data curation**. In practical scenarios beyond comprehensive LLM improvement, mission-critical applications (e.g., developing specialized LLM variants for coding or mathematical reasoning) often demand focused enhancement on target capabilities. Conventional baseline selection approaches, which lack domain-aware mechanisms, fundamentally preclude such fine-grained control.

To validate this customized selection capability, we conduct a validated experiment on Qwen2.5-7B. By strategically allocating **higher selection ratios (50%)** to specific target domains while reducing others to 10%, while maintaining the total training corpus size at 8,000 instances, we investigate whether such curated data distribution induces domain-specific performance gains.

As presented in Table 11, our analysis reveals that compared to the strongest baselines (RANDOM and original DaaR), customized DAAR achieves **SOTA enhancement** in focused domains, despite observing marginal performance degradation in non-target domains that leads to slightly inferior overall performance compared to the original DaaR. Interestingly, the performance on **Reasoning-Major** takes first place on **Avg** score, highlighting that the reasoning capability may be the crucial factor of LLMs. This validation promisingly confirms the operational feasibility of our **customization paradigm** and demonstrates DAAR' unique superiority over existing baselines in enabling domain-prioritized LLM specialization.

Table 11: Validation results of customized selected ratio on Qwen2.5-7B across benchmarks.

| Qwen2.5-7B | Common Sense | | Reasoning | Mathematics | | Coding | | Avg |
|---|---|---|---|---|---|---|---|---|
| | NQ | TriviaQA | Hellaswag | GSM8K | MATH | MBPP | HumanEval | |
| RAW | 8.84 | 58.14 | 72.75 | 78.20 | 9.10 | 7.40 | 78.05 | 44.64 |
| RAND (8K) | 11.46 | 57.85 | 73.08 | 78.90 | 13.15 | 62.50 | 71.65 | 52.66 |
| DAAR (Original) | 15.83 | 58.65 | 72.48 | 80.20 | 16.70 | 64.20 | 68.29 | **53.76** |
| Common-Major | **17.56** | **59.58** | 72.51 | 77.80 | 12.85 | 62.10 | 67.53 | 52.85 |
| Reasoning-Major | 16.56 | 58.22 | **74.62** | 77.30 | **17.20** | 63.40 | 66.42 | 53.39 |
| Math-Major | 13.20 | 57.96 | 74.16 | **81.40** | 16.05 | 61.80 | 63.45 | 52.75 |
| Coding-Major | 10.76 | 57.02 | 72.78 | 78.20 | 12.80 | **65.20** | **73.22** | 52.85 |

## F.4 Robustness to Labeling Schemes and Embedding Representations

A core principle of DAAR is its "model-aware" design, which relies on the model's own representations (via its embedding layer) and internal data structure (via clustering-based pseudo-labels) rather than external annotations or models. This section presents two ablation studies that investigate the validity and effectiveness of these design choices.

**Analysis 1: Model-Aware Pseudo-Labels vs. Ground-Truth Labels**  To analyze the impact of our clustering-based pseudo-labels, we conducted an experiment where we replaced them with human-annotated, ground-truth (GT) domain labels. This allowed us to directly compare a "model-aware" perspective against a "human-aware" one within our framework.

**Setup.** We reran the entire DAAR pipeline, using the GT labels to train the domain discrimination probe and subsequently to compute the predictive entropy for data selection. All other components remained identical.

**Observations.**

- **Data Selection Divergence:** The choice of labels significantly altered the data selection process. As shown in Table 12, the overlap between the data selected using pseudo-labels and GT labels was only around 83-87%, indicating that they prioritize different data subsets.

Table 12: Overlap of selected data between using pseudo-labels and ground-truth (GT) labels.

| Model | Overlap (Pseudo vs. GT) |
|---|---|
| Llama3.1-8B | 87.3% |
| Qwen2-7B | 83.1% |
| Qwen2.5-7B | 84.9% |

- **Final Performance:** Critically, using GT labels led to a consistent, albeit small, *decrease* in average end-task performance across all models, as detailed in Table 13. This counter-intuitive result suggests that for fine-tuning, an LLM's internal perception of data domains (captured by pseudo-labels) can be a more effective guide than human-defined categories. The GT labels may introduce biases that, while semantically correct to a human, are suboptimal for optimizing the model's capabilities.

**Analysis 2: Internal vs. External Embedding**  We also investigated the choice of using the model's own embedding layer versus relying on powerful, external SOTA embedding models. While external models could provide an alternative "semantic space," we evaluated the trade-offs.

**Setup.** We processed our dataset using two SOTA external embedding models (GTE-Qwen2-7B and Qwen3-8B-Embedding) and compared the time cost against using the native embedding layer of the base LLM (Qwen2-7B).

**Observations.** While the external models were also capable of producing separable domain clusters, they introduced prohibitive computational overhead. As shown in Table 14, using external models was 175x to 192x slower and led to out-of-memory (OOM) errors on a 40GB GPU. This highlights the significant efficiency advantage of DAAR's internal approach.

Table 13: Performance comparison: DAAR using model-aware pseudo-labels vs. GT labels. The "Diff" row shows the performance change when switching from pseudo-labels to GT labels.

| Model | Setting | nq | triviaqa | hellaswag | gsm8k | math | mbpp | humaneval | Avg |
|---|---|---|---|---|---|---|---|---|---|
| Llama3.1-8B | DAAR w/ Pseudo | 20.08 | 64.55 | 74.88 | 54.80 | 15.30 | 4.70 | 37.50 | 38.83 |
| | DAAR w/ GT | 22.54 | 65.52 | 73.43 | 53.80 | 14.35 | 4.40 | 36.50 | 38.65 |
| | *Diff* | *+2.46* | *+0.97* | *-1.45* | *-1.00* | *-0.95* | *-0.30* | *-1.00* | *-0.18* |
| Qwen2-7B | DAAR w/ Pseudo | 16.88 | 57.58 | 73.03 | 75.40 | 38.10 | 52.00 | 64.94 | 53.99 |
| | DAAR w/ GT | 18.75 | 58.35 | 72.02 | 73.40 | 35.75 | 51.50 | 64.01 | 53.40 |
| | *Diff* | *+1.87* | *+0.77* | *-1.01* | *-2.00* | *-2.35* | *-0.50* | *-0.93* | *-0.59* |
| Qwen2.5-7B | DAAR w/ Pseudo | 15.83 | 58.65 | 72.48 | 80.20 | 16.70 | 64.20 | 68.29 | 53.76 |
| | DAAR w/ GT | 16.04 | 58.75 | 71.87 | 79.40 | 16.30 | 64.25 | 67.76 | 53.48 |
| | *Diff* | *+0.21* | *+0.10* | *-0.61* | *-0.80* | *-0.40* | *+0.05* | *-0.53* | *-0.28* |

Table 14: Time cost for extracting embeddings from a 40K dataset.

| Model/Layer | Time Cost (Relative Speed) |
|---|---|
| Qwen2-7B (Internal Embedding Layer) | 9.23s (1x) |
| GTE-Qwen2-7B (External Model) | 1662s (175x slower) |
| Qwen3-8B-Embedding (External Model) | 1774s (192x slower) |

## F.5 Analysis of End-to-End Stability

Given that DAAR is a multi-stage pipeline, it is important to analyze its end-to-end stability and robustness against potential compounding errors. To this end, we conducted three fully independent, end-to-end experimental runs of our entire framework and evaluated the consistency at critical stages of the pipeline. We present results for Qwen2-7B as a representative case.

**Stability of Centroid Generation**   The initial stage of our pipeline involves generating domain centroids. We measured the pairwise cosine similarities of the final centroid embeddings across the three independent runs. As shown in Table 15, the resulting centroids are remarkably consistent, with all similarity scores exceeding 0.98. This indicates that the foundational stage of our method is highly stable and is not a significant source of variance.

Table 15: Pairwise cosine similarity of domain centroids across three independent runs.

| Domain | Sim(Run 1-2) | Sim(Run 1-3) | Sim(Run 2-3) |
|---|---|---|---|
| Common Sense | 0.9912 | 0.9895 | 0.9921 |
| Reasoning | 0.9845 | 0.9911 | 0.9887 |
| Mathematics | 0.9880 | 0.9803 | 0.9854 |
| Coding | 0.9905 | 0.9853 | 0.9918 |

**Stability of Final Data Selection**   We then examined whether minor variations in the initial stages propagate to the data selection outcome. We calculated the overlap ratio of the top 20% (8,000) selected samples across the three runs. Table 16 shows an average overlap of 96.4%, confirming that the final data selection is highly robust and largely deterministic, with minimal amplification of randomness.

**Consistency of Final Performance**   Finally, the end-to-end stability of the framework is reflected in the final benchmark performance. As reported in Table 17, the results across the three runs are highly consistent, with a minimal standard deviation of 0.08 in the average score. This provides strong empirical evidence that our framework is robust and its outcomes are reliable, without significant compounding error effects.

Table 16: Overlap ratio of selected data subsets across three independent runs.

| Metric | Overlap (Run 1-2) | Overlap (Run 1-3) | Overlap (Run 2-3) |
|--------|-------------------|-------------------|-------------------|
| Overlap Ratio | 95.7% | 97.3% | 96.1% |

Table 17: End-task performance across three independent runs on Qwen2-7B. Scores are reported with mean and standard deviation.

| Run | nq | triviaqa | hellaswag | gsm8k | math | mbpp | humaneval | Avg |
|-----|-----|----------|-----------|-------|------|------|-----------|-----|
| Run 1 | 17.92 | 56.78 | 72.91 | 75.00 | 39.60 | 51.40 | 64.80 | 54.06 |
| Run 2 | 15.84 | 58.38 | 73.14 | 75.80 | 36.60 | 52.60 | 65.07 | 53.92 |
| Run 3 | 15.18 | 57.96 | 72.99 | 76.70 | 38.80 | 51.40 | 65.24 | 54.04 |
| **Avg** | 16.31 | 57.71 | 73.01 | 75.83 | 38.33 | 51.80 | 65.04 | 54.01 |
| **Std. Dev.** | (1.43) | (0.83) | (0.12) | (0.85) | (1.55) | (0.69) | (0.22) | (0.08) |

## F.6 Rationale for Entropy over Diversity as the Reward Signal

As discussed in Section 4, the explicit diversity metric defined in Eq. (2) & (3) inherently depends on cross-sample relationships that *cannot be captured* through conventional sample-level training paradigms. To empirically validate this limitation, we conducted an ablation study using 500 representative samples. The inter-diversity scores were calculated using Eq. (2) and partitioned into ascending quartiles, each representing distinct diversity levels. A 5-layer MLP classifier was subsequently trained with cross-entropy loss as a more tractable verification approach compared to regression. Experimental result (Fig. 11) **demonstrates persistent convergence challenges**, evidenced by consistently *high training loss* and *subpar* classification accuracy capped at **34%**. These findings substantiate our hypothesis that LLMs fail to learn cross-sample characteristics, thereby necessitating entropy-based proxy metrics for effective reward formulation.

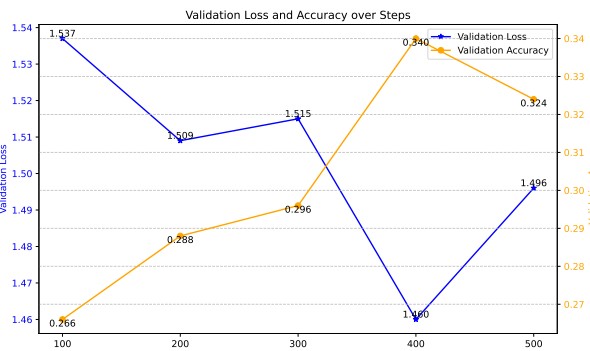

Figure 11: Validation loss and accuracy on directly use diversity instead of entropy as a reward signal.

# G Complete Ablation Studies and Discussion

## G.1 Similarity in Generated Domains' Centroid

To investigate whether there is a distinct separation between different domains generated, we compute the cosine similarity of semantic centroids across different domains. The experimental results are shown in Fig 12. It is evident that, compared to variations within the same domain at different data quantities, there are significant differences between different domains. This validates that this method of generating synthetic data can produce domain-representative data **with clear distinction**.

Notably, despite variations in **model architecture** and parameter count, the generated content consistently exhibits the greatest divergence between common sense, reasoning, and coding domains. Specifically, the discrepancy between common sense and coding, as well as reasoning and coding, is

markedly pronounced. Conversely, the semantic difference between common sense and reasoning is relatively smaller. This pattern suggests that the models, although differing in complexity and size, exhibit varying sensitivity to different data types, highlighting their nuanced capability to distinguish between certain domain-specific characteristics while maintaining subtle distinctions in others.

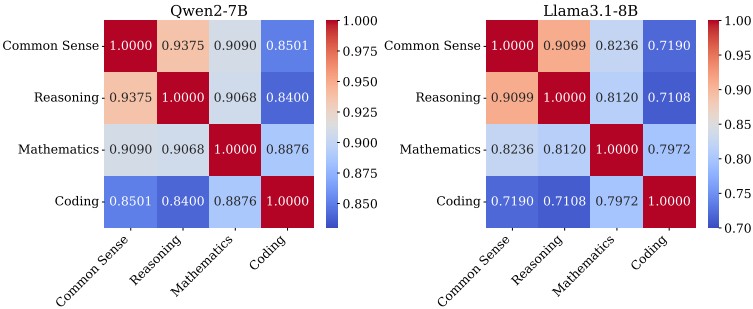

Figure 12: Semantic cosine similarity across different domains for generated samples of size 10.

## G.2  DAAR Layer Selection Protocol

We choose layer-3 as the hidden layer attached to $\psi_{dom}$. To determine an appropriate layer for embedding DAAR into an LLM that balances performance and computational cost, we first evaluate the capacity of different layers for domain awareness. Using Qwen2-7B as an example, we conduct t-SNE visualizations of Layer-5, Layer-10, Layer-15, Layer-20, Layer-25, and the last hidden layer, based on the average token vectors, as shown in Fig. 13.

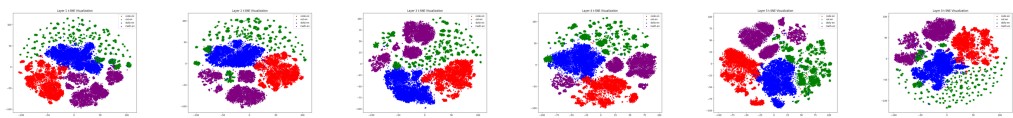

Figure 13: t-SNE visualization of mean token embeddings across Layer-5, Layer-10, Layer-15, Layer-20, Layer-25, and the last hidden layer of Qwen2-7B.

The observations demonstrate that LLMs **maintain cross-dataset classification capabilities** across all architectural depths. Considering the diminishing marginal utility of computational resources in deeper layers, we prioritize initial layers to optimize learning efficiency. The training dynamics across the first ten layers are visualized in Fig. 14, demonstrating consistent accuracy **within the 91.2–93.6% range**, verifying that various layers can effectively support DAAR first-stage training.

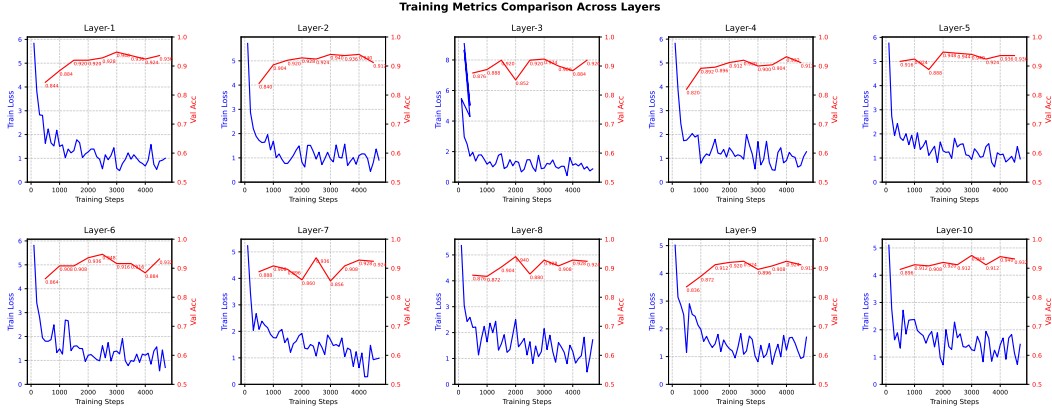

Figure 14: Validation loss and accuracy over steps on different layers.

Given that data clustering is based on the embedding layer, a natural approach was to directly connect to *the embedding layer* for training. However, the training results showed persistently high training

loss and a validation accuracy of **only about 0.6** (shown in Fig. 15), indicating **suboptimal learning performance**. Furthermore, related studies indicate that LLMs can learn semantic understanding and perception in the early layers [45], consequently, we select Layer-3 for DAAR.

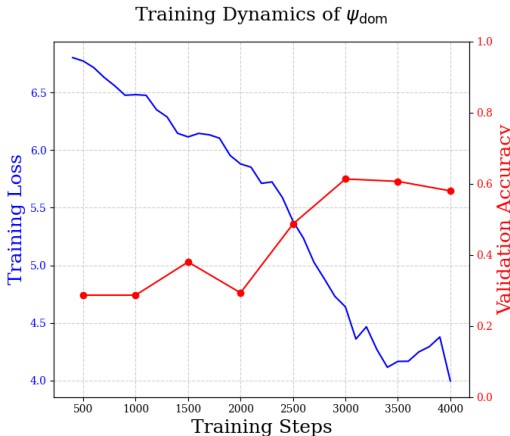

Figure 15: Training loss and validation process of Embedding Layer on Qwen2.5-7B.

### G.3 Ablation of the Number of Seed Samples

To investigate the impact of the number of seed samples $\mathcal{S}_k^{(0)}$ on domain centroid construction, we systematically selected 3, 5, and 10 as the number of seed samples while maintaining all other parameters at their default settings on Qwen2.5-7B. For each configuration, pairwise cosine similarity between domains was computed to assess whether significant discrepancies emerged across different seed sizes. Experimental results summarized in Table 18, demonstrate that similarities remain highly consistent across all domain pairs. Except for the common sense domain (with similarity approaching 0.9), varying seed sample counts do not yield substantial differences in the final centroids, with **variance ranging from 5.4e-7 to 1.8e-5**, demonstrating the stability of this hyperparameter.

Intuitively, we posit that excessively **small seed sizes** may compromise domain characterization due to generation randomness, while **larger sizes** risk introducing redundant patterns. We conjecture that the relatively lower absolute similarity in common sense domains stems from inherent sentence brevity, which increases vulnerability to lexical biases. However, this does not impair the anchoring effectiveness, as evidenced by the consistent performance across configurations.

Table 18: Semantic similarity between the number of seed samples among 3, 5 and 10 across domains on Qwen2.5-7B.

| Domains | Similarity of 3 & 5 | Similarity of 3 & 10 | Similarity of 5 & 10 |
| --- | --- | --- | --- |
| **Common Sense** | 0.8984 | 0.8913 | 0.9074 |
| **Reasoning** | 0.9713 | 0.9752 | 0.9815 |
| **Mathematics** | 0.9776 | 0.9761 | 0.9722 |
| **Coding** | 0.9775 | 0.9759 | 0.9760 |

### G.4 Ablation of the Number of Diversity Augmentation Data

To assess the impact of the amount of generated data on the accuracy of domain semantic centroids, we select quantities of 10 and 30 samples for generation. Subsequently, we calculate the semantic cosine similarity between these two sets, with results presented in Table 19 (using Qwen2.5-7B and Llama3.1-8B as examples). The observed differences between **varying data quantities were not significant**, indicating that the generalization of the generated data is sufficient, and further increasing the data quantity does not significantly enhance the accuracy of the centroids.

Table 19: Semantic similarity between generated samples of size 10 and 30 across different domains.

| Similarity of 10 & 30 | Common Sense | Reasoning | Mathematics | Coding |
|---|---|---|---|---|
| **Qwen2.5-7B** | 0.9686 | 0.9866 | 0.9919 | 0.9881 |
| **Llama3.1-8B** | 0.9851 | 0.9795 | 0.9685 | 0.9895 |

## G.5 Ablation of the Size of Sliding Window

To investigate the impact of the sliding window size in diversity augmentation, we kept all other hyper-parameters constant, setting the sliding window size to 1, 3, and 5 examples, respectively, and compared the similarity between them. The experimental results on Qwen2.5-7B, presented in Table 20, indicate that **different window sizes maintain a high degree of similarity**, suggesting that DAAR does not heavily depend on this hyperparameter choice. Considering the prompt length and stability, we opted for a window size of 3 in the implementation.

Table 20: Semantic similarity between the size of sliding window among 1, 3 and 5 across domains on Qwen2.5-7B.

| Domains | Similarity of 1 & 3 | Similarity of 1 & 5 | Similarity of 3 & 5 |
|---|---|---|---|
| **Common Sense** | 0.9205 | 0.9094 | 0.9022 |
| **Reasoning** | 0.9746 | 0.9760 | 0.9772 |
| **Mathematics** | 0.9736 | 0.9618 | 0.9847 |
| **Coding** | 0.9849 | 0.9647 | 0.9716 |

## G.6 Ablation of the Diversity Threshold

The threshold $\tau$ design needs to satisfy two objectives: (1) mitigate redundancy in LLM-generated outputs by preventing exact duplicates in synthetic data, and (2) ensure sufficient similarity among instances within the same domain embedding space. As illustrated in Table 21, we conduct an experiment using different similarity thresholds across various domains on Qwen2.5-7B, observing the number of attempts required for successful generation. The results indicate that distinct domains exhibit varying sensitivity to similarity metrics, with the number of attempts increasing sharply as the threshold decreases. Based on these observations, we empirically determined **a threshold value of 0.9** for Mathematics and **0.85** for others as a balanced compromise.

Table 21: Generation attempt frequencies for 10 seed samples across different domains and similarity levels on Qwen2.5-7B.

| Domains | 0.8 | 0.85 | 0.9 | 0.95 |
|---|---|---|---|---|
| **Common Sense** | 153 | 31 | 21 | 12 |
| **Reasoning** | 135 | 61 | 44 | 15 |
| **Mathematics** | > 1000 | 804 | 236 | 25 |
| **Coding** | 74 | 21 | 16 | 13 |

We further conducted ablation experiments by comparing the similarity of embedding centroids generated under different thresholds to systematically evaluate the impact of threshold variations on downstream performance. The results presented in Table 22 demonstrate, with the exception of common sense data (similar to the results in Table 18), that **varying thresholds do not significantly affect** the role of these centroids as initial points for clustering. We conjecture that lower thresholds ensure greater diversity within domain, whereas higher thresholds tend to result in repetitive data.

## G.7 Ablation of Model Scales

To assess the generalizability of DAAR beyond the 7-8B parameter range, we conducted additional experiments on models of different scales: a smaller model, Llama3.2-3B, and a larger model, Qwen2.5-14B. Due to computational constraints, we performed a targeted comparison of DAAR against two key baselines: **Raw** (using the original, unselected data) and **Random** (a strong baseline involving random data selection).

Table 22: Semantic similarity between the similarity threshold among 0.85, 0.9, and 0.95 across domains on Qwen2.5-7B.

| Domains | Similarity of 0.85 & 0.9 | Similarity of 0.85 & 0.95 | Similarity of 0.85 & 0.95 |
|---|---|---|---|
| **Common Sense** | 0.8869 | 0.8963 | 0.9502 |
| **Reasoning** | 0.9791 | 0.9785 | 0.9795 |
| **Mathematics** | 0.9730 | 0.9754 | 0.9703 |
| **Coding** | 0.9646 | 0.9740 | 0.9691 |

The results are presented in Table 23. On the larger Qwen2.5-14B model, DAAR achieves the highest average score (61.54), outperforming the strong Random baseline. Similarly, on the smaller Llama3.2-3B model, DAAR (29.64) maintains a consistent advantage over both Raw and Random baselines.

These findings provide evidence that the effectiveness of DAAR is not confined to a specific model size and generalizes to both smaller and larger language models.

Table 23: Performance comparison on Llama3.2-3B and Qwen2.5-14B, demonstrating the generalization capability of DAAR across different model scales.

| Model | Method | NQ | TriviaQA | Hellaswag | GSM8K | MATH | MBPP | HumanEval | Avg |
|---|---|---|---|---|---|---|---|---|---|
| | Raw | 10.89 | 66.56 | 76.86 | 86.00 | 19.70 | 5.40 | 78.66 | 49.15 |
| Qwen2.5-14B | Random | 20.06 | 65.80 | 76.76 | 86.40 | 36.90 | 64.20 | 75.46 | 60.80 |
| | **DAAR** | **19.73** | **66.26** | **77.19** | **85.00** | **39.10** | **66.80** | **76.69** | **61.54** |
| | Raw | 7.62 | 53.10 | 68.77 | 26.60 | 4.50 | 3.80 | 23.17 | 26.79 |
| LLaMA3.2-3B | Random | 16.03 | 53.56 | 68.46 | 28.70 | 6.15 | 4.65 | 29.12 | 29.52 |
| | **DAAR** | **15.13** | **53.79** | **68.23** | **29.10** | **5.63** | **5.30** | **30.34** | **29.64** |

## G.8 Ablation of Downstream Sample Injection in Seed Generation

The seed generation process described in Section 4.1 includes a minor injection of downstream task samples. This component is intended to provide an initial diversity signal to accelerate the subsequent augmentation phase. To rigorously evaluate its impact, we present an ablation study comparing our standard method against a variant where this injection is completely removed.

It is important to note that in all settings, our experimental protocol maintains strict data partitioning. The downstream samples used for seeding, when present, are drawn from a set entirely disjoint from the datasets used for final evaluation.

Our analysis focuses on three key aspects: the impact on the final centroid representations, the effect on end-task model performance, and the change in data generation efficiency.

**Impact on Centroid Similarity** We measured the cosine similarity between domain centroids generated with our standard method (w/ injection) and the ablation setting (w/o injection). The results in Table 24 show that the centroids are nearly identical, with an average similarity score of 0.987. This indicates that the minor injection has a negligible impact on the final learned domain representations.

Table 24: Cosine similarity of domain centroids generated with and without downstream seed injection, evaluated on Qwen2-7B.

| Domain | Similarity (w/ vs. w/o Injection) |
|---|---|
| Common Sense | 0.9922 |
| Reasoning | 0.9865 |
| Mathematics | 0.9843 |
| Coding | 0.9878 |
| **Average** | **0.9877** |

**Impact on Final Performance**   As shown in Table 25, the end-task performance of models trained using data selected by DAAR is statistically indistinguishable between the two settings. This demonstrates that the effectiveness of our method is not dependent on the seed injection.

Table 25: End-task performance comparison between DAAR with and without downstream seed injection. The results show statistically insignificant differences.

| Model | Setting | nq | triviaqa | hellaswag | gsm8k | math | mbpp | humaneval | Avg |
|-------|---------|------|----------|-----------|-------|-------|-------|-----------|-------|
| Llama3.1-8B | DaaR w/ Inject | 20.08 | 64.55 | 74.88 | 54.80 | 15.30 | 4.70 | 37.50 | 38.83 |
|  | DaaR w/o Inject | 20.39 | 64.80 | 76.05 | 55.40 | 13.65 | 5.75 | 36.48 | 38.93 |
| Qwen2-7B | DaaR w/ Inject | 16.88 | 57.58 | 73.03 | 75.40 | 38.10 | 52.00 | 64.94 | 53.99 |
|  | DaaR w/o Inject | 16.22 | 58.33 | 73.41 | 75.20 | 36.37 | 52.95 | 65.14 | 53.95 |
| Qwen2.5-7B | DaaR w/ Inject | 15.83 | 58.65 | 72.48 | 80.20 | 16.70 | 64.20 | 68.29 | 53.76 |
|  | DaaR w/o Inject | 14.91 | 58.32 | 72.55 | 79.70 | 16.30 | 63.70 | 70.65 | 53.73 |

**Impact on Generation Efficiency**   While not critical for performance, the injection significantly improves the efficiency of the diversity augmentation process. As detailed in Table 26, removing the initial seed diversity required approximately 4 times more generation attempts to meet the same diversity threshold ($\tau$). This confirms its functional role as a practical accelerator.

Table 26: Number of generation attempts required to meet the diversity threshold during seed augmentation.

| Domain | Attempts (w/ Injection) | Attempts (w/o Injection) |
|--------|-------------------------|--------------------------|
| Common Sense | 31 | 134 |
| Reasoning | 61 | 248 |
| Mathematics | 236 | 819 |
| Coding | 21 | 105 |

### G.9   Cost-Efficiency and Flexibility

Compared to baseline methods requiring GPT-based evaluators (ALPAGASUS, INSTAG) or full-LLM inference (DEITA), our approach achieves superior efficiency through data-model co-optimizations. Our method demonstrates computational efficiency through two key aspects enabled by the LLM's self-rewarding capability during dedicated data synthesis:

- Our framework operates on frozen embeddings extracted from layer 3 while truncating subsequent layers, which constitutes a significantly shallower architecture compared to the conventional 32-layer structure in 7B-scale LLMs. Taking Qwen2 as an example, full model inference requires loading complete parameters alongside cache management components, consuming **18–24 GB GPU memory**. In contrast, our method accomplishes data filtering with **merely 5–6 GB GPU memory** footprint.
- The lightweight 5-layer MLP probe module introduces only **76 million additional parameters**. As a regression model producing single scalar outputs per instance, it achieves substantial speed improvements over LLM-inference-based approaches. Experimental results on 5,000 samples demonstrate this efficiency advantage: our method requires approximately **40 minutes** for sample-level inference, whereas conventional LLM inference approaches demand **nearly two hours**.

### G.10   Stability of DAAR

Our dual-MLP design of DAAR strategically separates entropy regression (second MLP) from neural softmax-based measurement (first MLP), both showing stable convergence in Fig 2. While precise analysis of compounding errors remains a theoretical concern, our framework inherently mitigates these errors through these aspects: **(1) Seed Generation:** LLM-generated descriptions exhibit minimal variation across iterations, and repeated seeding mitigates uncertainty. **(2) Layer Selection:** Semantic feature extraction across layers is demonstrated to effectively capture domain-specific patterns, thereby establishing methodological robustness, detailed in Appendix G.2. **(3) Clustering:**

Data seeding as initialization for K-means clustering, where minimal perturbation of initial points ensures stable training of the first MLP. **(4) Selection Strategy:** Percentage-based data selection prioritizing sample distribution robustness over fine-grained entropy value utilization.

## H    Comprehensive Visualization and Experimental Results

### H.1    Visualization of Embeddings on Llama3.1-8B , Qwen2-7B & Qwen2.5-7B

We process concatenated data samples with "instruction"+"input"+"output" pairs through each LLM's embedding layer, computing **mean token** embeddings for visualization. We present the t-SNE visualization of data samples from Llama3.1-8, Qwen2-7B and Qwen2.5-7B with different distributions in Fig. 16, Fig. 17 and Fig. 18. It is evident that, despite differences in architecture or model, the method of filtering data through Inter-Diversity and Intra-Diversity is effective. Notably, although the embedding distributions of different models are not identical, they exhibit similar behavior, indicating that the representations learned by the embedding layers are comparable.

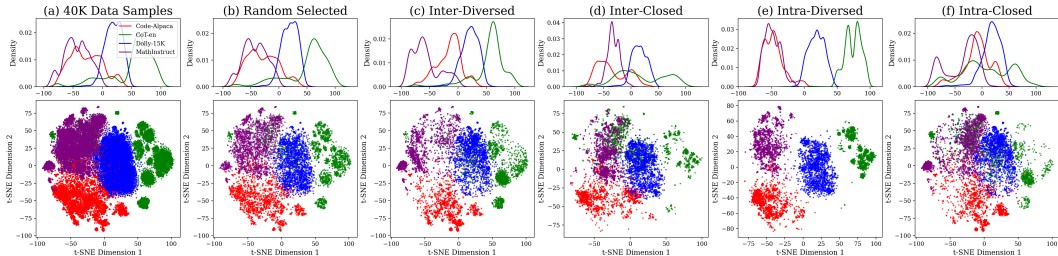

Figure 16: The t-SNE visualization of embeddings for data samples with different distributions on Llama3.1-8B. (a) The data pool of all 40K samples, (b) Randomly selected subset, (c) Distribution of data farthest from other domain centroids on Inter-Diversity, (d) Distribution of data closest to other domain centroids on Inter-Diversity, (e) Distribution of data closest to its own domain centroid on Inter-Diversity, (f) Distribution of data farthest from its own domain centroid on Inter-Diversity.

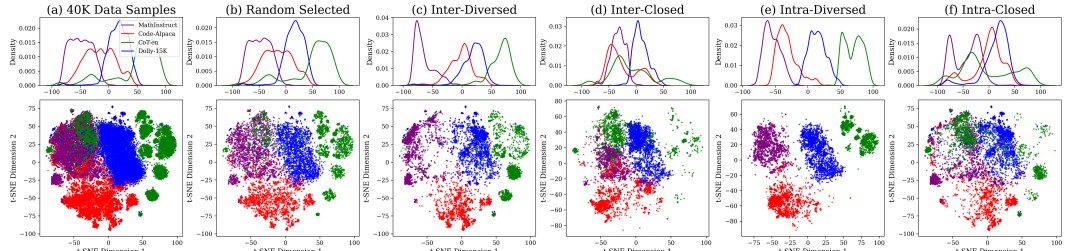

Figure 17: The t-SNE visualization of embeddings for data samples on Qwen2-7B.

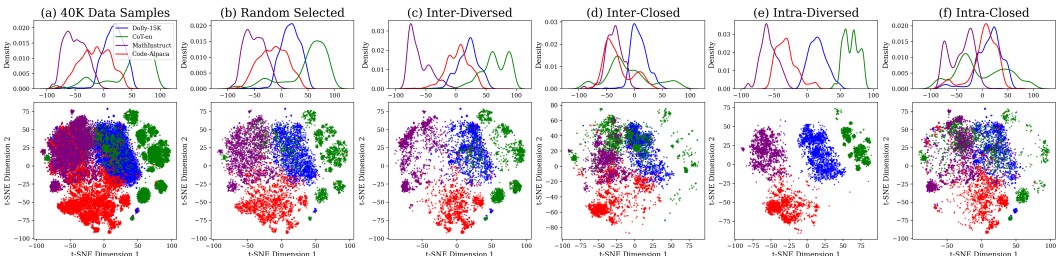

Figure 18: The t-SNE visualization of embeddings for data samples on Qwen2.5-7B.

### H.2    Visualization of Inter- & Intra-Diversity Distribution Data

Using the Qwen2-7B model as an example, we construct data based on the Inter-Diversity and Intra-Diversity distribution methods, selecting a batch of data every 20%. The visualization process is

shown in Fig. 19 and Fig. 20. As seen in the figures, the data gradually transitions from domain-aware diverse to domain-aware closed, indicating that our data construction method effectively controls the distribution of different data.

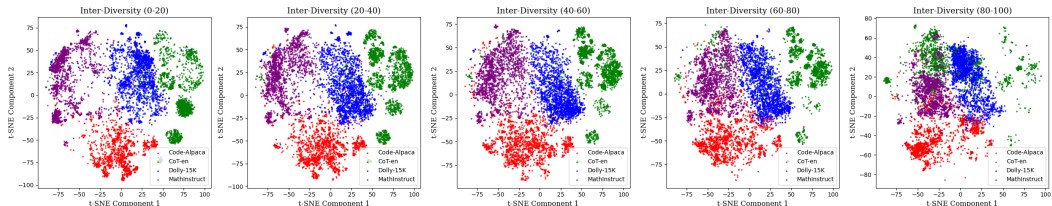

Figure 19: Data visualization (t-SNE) based on different **Inter-Diversity** distributions on Qwen2-7B.

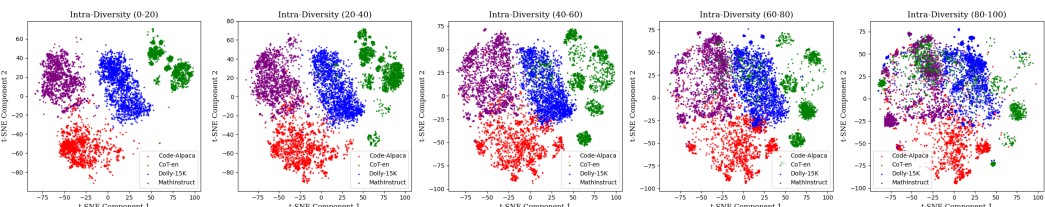

Figure 20: Data visualization (t-SNE) based on different **Intra-Diversity** distributions on Qwen2-7B.

## H.3  Comprehensive Training Dynamics

We present the training and validation dynamics for Llama3.1-8B and Qwen2.5-7B, in Fig 21. It can be observed that across different models, the training process of DAAR method consistently ensures gradual convergence, achieving high domain predictability and calibrated entropy fitting.

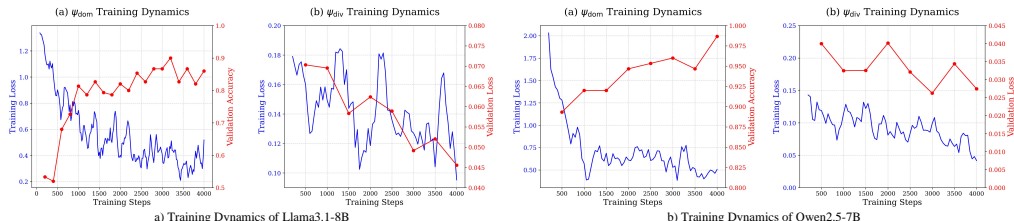

Figure 21: Training loss and validation process of the two training stages of DAAR on Llama3.1-8B and Qwen2.5-7B, showing the model gradually converging.

## H.4 Complete Validation Results of Inter-Diversity and Intra-Diversity

We present the complete experimental results of Llama3.1-8B, Qwen2-7B, and Qwen2.5-7B in the validation experiments in Tables 27, Tables 28, and Tables 29, respectively. It can be observed that for any complete dataset, the conclusions from Section 3.3 remain valid, specifically that the peak distribution of results is uneven, with significant differences among them.

Table 27: Validation results of Inter-Diversity and Intra-Diversity on Llama3.1-8B across benchmarks.

| Llama3.1-8B | Common Sense | | Reasoning | Mathematics | | Coding | | Avg |
|---|---|---|---|---|---|---|---|---|
| | NQ | TriviaQA | Hellaswag | GSM8K | MATH | MBPP | HumanEval | |
| **RAW** | 14.13 | 65.90 | 74.62 | 54.80 | 7.90 | 5.00 | 28.66 | 35.86 |
| **FULL (40K)** | 21.92 | 65.11 | 73.62 | 51.70 | 7.60 | 4.10 | 36.50 | 37.22 |
| **RAND (8K)** | 21.99 | 64.83 | 74.72 | 55.70 | 14.50 | 5.10 | 24.09 | 37.27 |
| Inter-Diversity (0-20) | 19.28 | 65.79 | 74.44 | 54.90 | 6.50 | 4.30 | 35.06 | 37.18 |
| Inter-Diversity (20-40) | 21.91 | 65.48 | 74.54 | 52.50 | 16.65 | 5.70 | 26.53 | 37.62 |
| Inter-Diversity (40-60) | 23.70 | 65.14 | 74.86 | 56.40 | 17.15 | 5.00 | 24.40 | 38.09 |
| Inter-Diversity (60-80) | 22.42 | 64.52 | 74.76 | 55.10 | 14.60 | 7.40 | 31.10 | 38.56 |
| **Inter-Diversity (80-100)** | 23.76 | 64.43 | 75.20 | 56.40 | 15.05 | 4.50 | 33.54 | **38.98** |
| Intra-Diversity (0-20) | 22.08 | 65.08 | 75.00 | 54.70 | 16.20 | 4.40 | 33.54 | 38.71 |
| Intra-Diversity (20-40) | 22.41 | 64.44 | 74.66 | 52.60 | 15.30 | 4.20 | 27.44 | 37.29 |
| Intra-Diversity (40-60) | 22.12 | 64.74 | 74.87 | 54.00 | 16.00 | 6.00 | 27.44 | 37.88 |
| Intra-Diversity (60-80) | 20.83 | 64.30 | 74.36 | 52.20 | 14.90 | 4.50 | 35.98 | 38.15 |
| **Intra-Diversity (80-100)** | 19.78 | 64.77 | 74.51 | 56.50 | 13.00 | 5.20 | 37.50 | **38.75** |

Table 28: Validation results of Inter-Diversity and Intra-Diversity on Qwen2-7B across benchmarks.

| Qwen2-7B | Common Sense | | Reasoning | Mathematics | | Coding | | Avg |
|---|---|---|---|---|---|---|---|---|
| | NQ | TriviaQA | Hellaswag | GSM8K | MATH | MBPP | HumanEval | |
| **RAW** | 8.03 | 59.58 | 73.00 | 78.00 | 5.70 | 5.00 | 60.98 | 41.47 |
| **FULL (40K)** | 15.61 | 58.75 | 72.51 | 73.80 | 31.30 | 51.70 | 67.38 | 53.01 |
| **RAND (8K)** | 13.28 | 58.27 | 73.00 | 75.35 | 35.36 | 52.20 | 63.72 | 53.02 |
| **Inter-Diversity (0-20)** | 15.18 | 59.28 | 73.34 | 74.50 | 34.94 | 53.10 | 68.60 | **54.13** |
| Inter-Diversity (20-40) | 13.77 | 58.42 | 73.18 | 73.60 | 32.55 | 53.00 | 64.33 | 52.69 |
| Inter-Diversity (40-60) | 14.62 | 58.58 | 73.35 | 72.50 | 34.50 | 52.50 | 61.28 | 52.47 |
| Inter-Diversity (60-80) | 14.31 | 58.60 | 73.33 | 74.80 | 33.90 | 51.40 | 60.68 | 52.43 |
| Inter-Diversity (80-100) | 9.30 | 57.72 | 73.14 | 74.60 | 28.00 | 51.30 | 63.42 | 51.07 |
| Intra-Diversity (0-20) | 12.64 | 58.54 | 73.35 | 75.10 | 8.75 | 51.10 | 61.59 | 48.72 |
| Intra-Diversity (20-40) | 14.17 | 58.78 | 73.10 | 74.10 | 29.20 | 52.10 | 63.41 | 52.12 |
| Intra-Diversity (40-60) | 15.24 | 58.57 | 73.12 | 74.70 | 32.50 | 51.80 | 64.02 | 52.85 |
| Intra-Diversity (60-80) | 14.02 | 57.40 | 73.06 | 75.20 | 32.20 | 53.50 | 66.77 | 53.16 |
| **Intra-Diversity (80-100)** | 11.91 | 57.88 | 73.29 | 75.00 | 36.05 | 52.50 | 66.16 | **53.25** |

Table 29: Validation results of Inter-Diversity and Intra-Diversity on Qwen2.5-7B across benchmarks.

| Qwen2.5-7B | Common Sense | | Reasoning | Mathematics | | Coding | | Avg |
|---|---|---|---|---|---|---|---|---|
| | NQ | TriviaQA | Hellaswag | GSM8K | MATH | MBPP | HumanEval | |
| **RAW** | 8.84 | 58.14 | 72.75 | 78.20 | 9.10 | 7.40 | 78.05 | 44.64 |
| **FULL (40K)** | 12.88 | 58.60 | 72.28 | 76.80 | 13.60 | 62.80 | 71.04 | 52.57 |
| **RAND (8K)** | 11.46 | 57.85 | 73.08 | 78.90 | 13.15 | 62.50 | 71.65 | 52.65 |
| Inter-Diversity (0-20) | 13.23 | 58.15 | 73.27 | 78.70 | 11.45 | 62.30 | 69.21 | 52.33 |
| Inter-Diversity (20-40) | 10.81 | 58.11 | 73.02 | 77.90 | 16.95 | 62.30 | 68.29 | 52.48 |
| **Inter-Diversity (40-60)** | 10.75 | 57.89 | 72.90 | 73.30 | 26.70 | 62.80 | 69.51 | **53.41** |
| Inter-Diversity (60-80) | 10.43 | 58.19 | 73.10 | 78.40 | 17.05 | 62.80 | 71.95 | 53.13 |
| Inter-Diversity (80-100) | 10.00 | 58.10 | 73.11 | 77.30 | 16.45 | 62.30 | 67.07 | 52.05 |
| **Intra-Diversity (0-20)** | 10.68 | 58.52 | 73.18 | 80.10 | 25.80 | 62.50 | 68.90 | **54.24** |
| Intra-Diversity (20-40) | 11.21 | 58.14 | 73.02 | 79.50 | 17.75 | 62.90 | 67.38 | 52.84 |
| Intra-Diversity (40-60) | 11.57 | 58.11 | 72.94 | 76.00 | 15.65 | 62.50 | 65.25 | 51.72 |
| Intra-Diversity (60-80) | 10.89 | 57.91 | 72.92 | 75.80 | 11.35 | 62.00 | 66.16 | 51.00 |
| Intra-Diversity (80-100) | 12.79 | 58.09 | 73.21 | 75.40 | 16.05 | 62.50 | 49.39 | 49.63 |

## H.5 Complete Results of DAAR with Baselines

Additionally, we include the complete results of DAAR and the comparative baselines in the three tables: Table 30, Table 31 and Table 32. The results illustrate the challenges of the scenario, particularly for the Qwen2 series, where baseline methods struggle to outperform random selection. Furthermore, they demonstrate the robustness and effectiveness of our approach, consistently achieving the highest average scores across different models.

Table 30: Performance of DAAR with baselines on Llama3.1-8B across various benchmarks.

| Llama3.1-8B | Common Sense | | Reasoning | Mathematics | | Coding | | Avg |
|---|---|---|---|---|---|---|---|---|
| | NQ | TriviaQA | Hellaswag | GSM8K | MATH | MBPP | HumanEval | |
| RAW | 14.13 | 65.90 | 74.62 | 54.80 | 7.90 | 5.00 | 28.66 | 35.86 |
| FULL (40K) | 21.92 | 65.11 | 73.62 | 51.70 | 7.60 | 4.10 | 36.50 | 37.22 |
| RAND (8K) | 21.99 | 64.83 | 74.72 | 55.70 | 14.50 | 5.10 | 24.09 | 37.27 |
| INSTRUCTION LEN | 15.34 | 63.60 | 73.73 | 54.00 | 15.40 | 3.60 | 30.80 | 36.64 |
| ALPAGASUS [10] | 21.57 | 64.37 | 74.87 | 55.20 | 17.65 | 4.60 | 16.16 | 36.34 |
| INSTAG-C [34] | 18.12 | 64.96 | 74.01 | 55.70 | 15.50 | 4.80 | 37.81 | 38.70 |
| INSTAG-D [34] | 21.94 | 64.69 | 74.87 | 54.80 | 12.80 | 4.10 | 9.76 | 34.71 |
| SUPERFILTER [29] | 22.95 | 64.99 | 76.39 | 57.60 | 6.05 | 2.60 | 40.55 | 38.73 |
| DEITA-C [33] | 15.58 | 64.97 | 74.21 | 55.00 | 13.05 | 4.60 | 34.46 | 37.41 |
| DEITA-Q [33] | 19.57 | 64.22 | 75.15 | 54.00 | 7.20 | 4.20 | 28.35 | 36.10 |
| DEITA-D [33] | 20.97 | 63.32 | 75.10 | 54.90 | 7.00 | 4.00 | 31.71 | 36.71 |
| DAAR (Ours) | 20.08 | 64.55 | 74.88 | 54.8 | 15.30 | 4.70 | 37.50 | **38.83** |

Table 31: Performance of DAAR with baselines on Qwen2-7B across various benchmarks.

| Qwen2-7B | Common Sense | | Reasoning | Mathematics | | Coding | | Avg |
|---|---|---|---|---|---|---|---|---|
| | NQ | TriviaQA | Hellaswag | GSM8K | MATH | MBPP | HumanEval | |
| RAW | 8.03 | 59.58 | 73.00 | 78.00 | 5.70 | 5.00 | 60.98 | 41.47 |
| FULL (40K) | 15.61 | 58.75 | 72.51 | 73.80 | 31.30 | 51.70 | 67.38 | 53.01 |
| RAND (8K) | 13.28 | 58.27 | 73.00 | 75.35 | 35.36 | 52.20 | 63.72 | 53.02 |
| INSTRUCTION LEN | 8.62 | 58.44 | 72.86 | 73.30 | 27.05 | 53.10 | 63.72 | 51.01 |
| ALPAGASUS [10] | 13.67 | 57.94 | 73.04 | 73.90 | 32.30 | 51.40 | 63.41 | 52.24 |
| INSTAG-C [34] | 9.51 | 58.50 | 73.06 | 74.70 | 35.35 | 51.90 | 64.70 | 52.53 |
| INSTAG-D [34] | 12.87 | 57.48 | 72.80 | 74.40 | 33.75 | 51.80 | 64.02 | 52.45 |
| SUPERFILTER [29] | 19.16 | 58.98 | 72.99 | 73.70 | 30.10 | 52.40 | 58.85 | 52.31 |
| DEITA-C [33] | 8.94 | 58.07 | 73.06 | 73.90 | 35.55 | 52.90 | 62.20 | 52.09 |
| DEITA-Q [33] | 14.06 | 59.07 | 73.16 | 75.80 | 35.50 | 23.00 | 58.24 | 48.40 |
| DEITA-D [33] | 16.41 | 57.80 | 72.70 | 76.10 | 29.05 | 52.40 | 64.63 | 52.73 |
| DAAR (Ours) | 16.88 | 57.58 | 73.03 | 75.40 | 38.1 | 52.00 | 64.94 | **53.99** |

Table 32: Performance of DAAR with baselines on Qwen2.5-7B across various benchmarks.

| Qwen2.5-7B | Common Sense | | Reasoning | Mathematics | | Coding | | Avg |
|---|---|---|---|---|---|---|---|---|
| | NQ | TriviaQA | Hellaswag | GSM8K | MATH | MBPP | HumanEval | |
| RAW | 8.84 | 58.14 | 72.75 | 78.20 | 9.10 | 7.40 | 78.05 | 44.64 |
| FULL (40K) | 12.88 | 58.60 | 72.28 | 76.80 | 13.60 | 62.80 | 71.04 | 52.57 |
| RAND (8K) | 11.46 | 57.85 | 73.08 | 78.90 | 13.15 | 62.50 | 71.65 | 52.65 |
| INSTRUCTION LEN | 11.34 | 58.01 | 72.79 | 78.00 | 15.80 | 62.30 | 68.12 | 52.34 |
| ALPAGASUS [10] | 10.40 | 57.87 | 72.92 | 77.20 | 18.75 | 61.80 | 65.55 | 52.07 |
| INSTAG-C [34] | 10.81 | 58.45 | 73.27 | 76.00 | 13.30 | 61.80 | 68.29 | 51.70 |
| INSTAG-D [34] | 11.08 | 58.40 | 72.79 | 76.40 | 16.40 | 62.90 | 70.43 | 52.63 |
| SUPERFILTER [29] | 13.54 | 58.51 | 72.89 | 79.30 | 11.35 | 39.50 | 65.25 | 48.62 |
| DEITA-C [33] | 10.50 | 58.17 | 73.14 | 74.60 | 16.60 | 62.00 | 72.26 | 52.47 |
| DEITA-Q [33] | 11.24 | 57.83 | 72.97 | 78.50 | 12.95 | 38.10 | 67.68 | 48.47 |
| DEITA-D [33] | 10.48 | 57.81 | 73.05 | 77.20 | 15.25 | 52.90 | 69.21 | 50.84 |
| DAAR (Ours) | 15.83 | 58.65 | 72.48 | 80.20 | 16.70 | 64.20 | 68.29 | **53.76** |

