# OpenReview forum: "Diversity as a Reward: Fine-Tuning LLMs on a Mixture of Domain-Undetermined Data"
_NeurIPS.cc/2025/Conference — NeurIPS 2025 poster_

### Official Review · Reviewer_Bbh9 · 2025-06-24

**Clarity:** 1
**Significance:** 2
**Originality:** 1
**Rating:** 3
**Confidence:** 4

**Summary:**

The paper proposes DAAR, a framework for data selection in instruction fine-tuning of LLMs when explicit domain labels are unavailable. The authors identify limitations in current mixture modeling and data selection approaches, especially in domain-undetermined settings. They begin with an empirical analysis revealing that both inter-domain and intra-domain diversity have model-specific and task-specific effects on downstream performance. Then they present their method in multiple stages, from centroid computation from the LLM embeddings, to computation of the diversity scores, and finally the data selection. Experiments across LLMs show some performance benefit.

**Questions:**

- Can the authors provide a clear algorithmic summary or flowchart to consolidate the numerous defined components and clarify how they interact within the overall training pipeline?
- How does the proposed method meaningfully differ from prior works like [1,2,3] that also utilize domain centroids and clustering approaches?
- Why were established domain reweighting baselines such as DoReMi and RegMix excluded from the experimental comparison, and how might their inclusion impact your reported performance gains?*
- Given that some improvements over baselines are marginal, can the authors report statistical significance measures to support the robustness of their claims?

**Ethical Concerns:**

["NO or VERY MINOR ethics concerns only"]

**Final Justification:**

The methodological flow is difficult to follow, making the core idea of the paper unclear.

Section 2.2 lacks clarity on assumptions on domains.

The distinction between “text description” and “domain label” is vague, as both functionally serve the same role (e.g., ‘wikipedia’ as both label and description).

The foundational step appears to rely on clustering to create pseudo-labels, but this has already been proposed in previous works, as well as centroid computation.

Once pseudo-labels are created, existing domain-reweighting baselines (e.g., DoReMi, RegMix) that rely on labels could also be applied, but they are not compared.

The argument that those baselines are “pre-training methods” is unconvincing, since their reweighting mechanisms resemble the proposed diversity probe.

Since individual components offer limited novelty, the main contribution must be the pipeline itself, which is currently not presented clearly enough and not supported by broad enough experiments.

For the pipeline to be convincing, the paper needs:

A clearer and more formal presentation (e.g., improved Figure 1, a step-by-step algorithm, better paper structure).

Broader and more comprehensive experiments, as current evaluation only includes two reasoning benchmarks (HellaSwag and MMLU), which is insufficient to establish general utility

**Limitations:**

Yes.

**Quality:**

2

**Strengths And Weaknesses:**

Strengths:
- The paper is well-motivated. The challenge of fine-tuning on mixed data without reliable domain labels is a real-world bottleneck. Especially given the increasing scale and heterogeneity of data used to adapt LLMs. The authors clearly articulate how conventional data selection and mixture strategies falter under such constraints, setting up a compelling case for the necessity of a label-free approach. They ground their method in both empirical observations and theoretical analysis, they highlight a practical need in the field.

Weaknesses:
- The writing and the structure of the paper make it quite difficult to follow the main steps of the contributions. My suggestion is to restructure the paper in a more logical way and include an algorithm listing. There are many quantities defined but it is not totally clear how they are used in a single pipeline.
- The novelty of the proposals in Sec. 3.2 is limited w.r.t. previous works. Using the domain centroid in step (A) is proposed in [1] and using their distances in (B) is also used in [1,2]. Domain-aware clustering in Sec. 4.1 does not discuss the similar [3].
- Experiments lack domain reweighting baselines such as DoReMi [4] and RegMix [5], which could also be applied to your clustered domains.
- No statistical significance measures, which are important given the marginal improvements.

[1] Xie, W., Tonin, F., & Cevher, V. (2025). Chameleon: A Flexible Data-mixing Framework for Language Model Pretraining and Finetuning. arXiv preprint arXiv:2505.24844.

[2] Chen, M. F., Hu, M. Y., Lourie, N., Cho, K., & Ré, C. (2024). Aioli: A unified optimization framework for language model data mixing. arXiv preprint arXiv:2411.05735.

[3] Fan, S., Grangier, D., & Ablin, P. (2024). Dynamic Gradient Alignment for Online Data Mixing. arXiv preprint arXiv:2410.02498.

[4] Xie, S. M., Pham, H., Dong, X., Du, N., Liu, H., Lu, Y., ... & Yu, A. W. (2023). Doremi: Optimizing data mixtures speeds up language model pretraining. Advances in Neural Information Processing Systems, 36, 69798-69818.

[5] Liu, Q., Zheng, X., Muennighoff, N., Zeng, G., Dou, L., Pang, T., ... & Lin, M. (2024). Regmix: Data mixture as regression for language model pre-training. arXiv preprint arXiv:2407.01492.

---

> ### Author Rebuttal · Authors · 2025-07-31
>
> We sincerely thank you for your time and constructive feedback. We have carefully considered your Weaknesses (W) and Questions (Q) point by point and provide detailed responses below. We hope this clarification will help you see the value of our work more clearly.
>
> ---
>
> ### **W2 & Q2: On the Novelty Compared to Works [1, 2, 3]**
> > *"W: The novelty of the proposals in Sec. 3.2 is limited w.r.t. previous works. ..."*
> >
> > *"Q: How does the proposed method meaningfully differ from prior works like [1,2,3] that also utilize domain centroids and clustering approaches?"*
>
> We are grateful to the reviewer for these valuable references. They will undoubtedly help us better position our work, and **we will add all of them to our revised version**.
>
> Similar to your statement in `Strengths` that our work positioned in **'fine-tuning on mixed data without reliable domain labels'**, we wish to re-emphasize that the primary novelty of our work lies in the **domain-undetermined** setting and the **diversity probe** of LLM **fine-tuning (in lines 74-79)**. To clarify this, we highlight the key differences with the cited works below.
>
> `Regarding [1] (Chameleon)`
>
> First, we note that [1] is **contemporaneous work** according to the official guidance of Call For Papers (first submitted on May 30), and it appears even after the submission (May 15). Nonetheless, we outline the distinctions here:
>
> While both DaaR and Chameleon use domain embeddings for *data reweighting*, our work differs in three crucial aspects:
>
> - **Setting:** DaaR operates in a **domain-undetermined fine-tuning** scenario where domain labels are unknown. In contrast, Chameleon's method (KRLS) use such domain information to calculate importance scores majorly in **pretraining** stage.
> - **Metric:** DaaR pioneers the use of **diversity** to measure the utility of data, while Chameleon use domain-level **importance scores** derived from KRLS.
> - **Objective:** In the context of fine-tuning, DaaR targets the enhancement of **foundational abilities**, whereas Chameleon focuses on specific skills like multilingual capabilities.
>
> `Regarding [2] (Aioli)`
>
> While both works investigate *data mixture based on domain properties*, our approaches differ fundamentally in their goals and methods:
>
> - **Objective & Setting:** Aioli mainly aims to **model a "mixing law"** in a **pre-training, domain-known** setting. In contrast, DaaR is to optimize data selection for **fine-tuning** in a **domain-undetermined** context.
> - **Mechanism:** Aioli's method relies on **training dynamics** (loss). DaaR is based on the intrinsic **diversity** of the data.
> - **On Embedding Distance:** Aioli **does not directly utilize domain centroid distance** as a core mechanism, the mention of embedding distance partly in its `Related Works` section.
>
> `Regarding [3] (DGA)`
>
> While both methods use *domain clustering*, its purpose and the overall approach are distinct:
>
> - **Core Method:** DGA is a **pre-training** method that uses **gradient information**. DaaR is a **fine-tuning** method that uses **data diversity** as its primary selection criterion.
> - **Role of Clustering:** In DGA, clustering is a **pre-processing step** to define static, fine-grained domains. In DaaR, we utilize **data synthesis to create anchor points** that guide the clustering, and the clustering is an integral mechanism to train the diversity probe.
>
> We thank the reviewer again for these valuable references. They help us clarify that DaaR’s core novelty lies **not in its individual components**, but in the complete system and scenario. As highlighted by Reviewers kdaW and Ge2Q in their `Strength`, DaaR provides a **external-model-free** solution for the challenging **domain-undetermined** setting.
>
> ---
>
> ### **W4 & Q4: On Reporting Statistical Significance**
> > *"W: No statistical significance measures, which are important given the marginal improvements."*
> >
> > *"Q: ... can the authors report statistical significance measures to support the robustness of their claims?"*
>
> We thank the reviewer for this valuable suggestion. To bolster our claims, we provide the following points:
>
> First, all existing results are the **average of two independent, end-to-end runs** (lines 112-113), and we have analyzed DaaR's stability in **Appendix G.8 (lines 894-904)**.
>
> To further address this concern, we conducted an additional independent run for our method, DaaR, for a total of **three runs**. The tables below *detail the performance* for each of the three seeds, followed by the *calculated mean* and *standard deviation* (μ ± σ).
>
> - **Llama3.1-8B**
>
> | Runs | nq | triviaqa | hellaswag | gsm8k | math | mbpp | humaneval | Avg |
> | :--- | :---: | :---: | :---: | :---: | :---: | :---: | :---: | :---: |
> | Seed-1 | 19.30 | 64.76 | 75.10 | 54.40 | 14.40 | 4.60 | 37.80 | 38.62 |
> | Seed-2 | 20.86 | 64.34 | 74.66 | 55.20 | 16.20 | 4.80 | 37.20 | 39.04 |
> | **Seed-3** | 21.91 | 64.94 | 74.66 | 54.20 | 16.20 | 4.60 | 37.20 | 39.10 |
> | **Average** | 20.69 ± (**1.31**) | 64.68 ± (**0.31**) | 74.81 ± (**0.25**) | 54.60 ± (**0.53**) | 15.60 ± (**1.04**) | 4.67 ± (**0.12**) | 37.40 ± (**0.35**) | 38.92 ± (**0.26**) |
>
> - **Qwen2-7B**
>
> | Runs | nq | triviaqa | hellaswag | gsm8k | math | mbpp | humaneval | Avg |
> | :--- | :---: | :---: | :---: | :---: | :---: | :---: | :---: | :---: |
> | Seed-1 | 17.92 | 56.78 | 72.91 | 75.00 | 39.60 | 51.40 | 64.80 | 54.06 |
> | Seed-2 | 15.84 | 58.38 | 73.14 | 75.80 | 36.60 | 52.60 | 65.07 | 53.92 |
> | **Seed-3** | 15.18 | 57.96 | 72.99 | 76.70 | 38.80 | 51.40 | 65.24 | 54.04 |
> | **Average** | 16.31 ± (**1.43**) | 57.71 ± (**0.83**) | 73.01 ± (**0.12**) | 75.83 ± (**0.85**) | 38.33 ± (**1.55**) | 51.80 ± (**0.69**) | 65.04 ± (**0.22**) | 54.01 ± (**0.08**) |
>
> - **Qwen2.5-7B**
>
> | Runs | nq | triviaqa | hellaswag | gsm8k | math | mbpp | humaneval | Avg |
> | :--- | :---: | :---: | :---: | :---: | :---: | :---: | :---: | :---: |
> | Seed-1 | 15.51 | 58.74 | 72.43 | 80.80 | 17.60 | 63.60 | 67.06 | 53.68 |
> | Seed-2 | 16.15 | 58.56 | 72.52 | 79.60 | 15.80 | 64.80 | 69.51 | 53.85 |
> | **Seed-3** | 15.58 | 58.40 | 72.90 | 79.40 | 16.40 | 62.90 | 68.12 | 53.39 |
> | **Average** | 15.75 ± (**0.35**) | 58.57 ± (**0.17**) | 72.62 ± (**0.25**) | 79.93 ± (**0.76**) | 16.60 ± (**0.92**) | 63.77 ± (**0.96**) | 68.23 ± (**1.23**) | 53.64 ± (**0.23**) |
>
> The analysis reveals that DaaR's performance is **highly consistent**, with the standard deviations on the final average scores being **notably low** (0.26, 0.08, and 0.23, respectively). This low variance confirms that our method's performance advantage over the reported baselines is robust.
>
> We appreciate this valuable feedback and will incorporate this detailed statistical analysis into our revised manuscript.
>
> ---
>
> ### **W3 & Q3: On the Comparison with DoReMi [4] and RegMix [5]**
> > *"Q: Experiments lack domain reweighting baselines such as DoReMi [4] and RegMix [5], which could also be applied to your clustered domains."*
> >
> > *"W: Why were ... DoReMi and RegMix excluded from the experimental comparison, ..."*
>
> We thank the reviewer for suggesting these foundational pretraining works. We **already cite DoReMi (line 199) and will add RegMix to our discussion**. However, we respectfully argue that including them as direct baselines would be **inappropriate** and could lead to an **inequitable comparison**:
>
> - **Divergent Settings:** DoReMi and RegMix are **pre-training** methods designed for large-scale, iterative reweighting. Our work operates in the **fine-tuning stage**, which is **fundamentally distinct** from pretraining in its objective and data usage[1, 2, 3].
> - **Established Baselines:** Our experimental setup aligns with the **standard practice in fine-tuning data selection**. To our knowledge, leading methods (e.g., Alpagasus, Instag, Deita), which we compare as baselines, **do not use any pre-training method** as their baselines.
> - **Methodological Incompatibility:** Both methods require **explicit domain labels**, which are absent in our **domain-undetermined** setting. While 'could also be applied to the clustered domains', our clustering is an **integral part of DaaR**, designed to provide signals for our diversity probe.
>
> [1] LIMA: Less Is More for Alignment, NeurIPS'2023
>
> [2] What Makes Good Data for Alignment? A Comprehensive Study of Automatic Data Selection in Instruction Tuning, ICLR'2024
>
> [3] A Survey on Data Selection for Language Models, TMLR'2024
>
> ---
>
> ### **W1 & Q1: On the paper's clarity and structure**
> > *"W: The writing and the structure of the paper make it quite difficult to follow ... My suggestion is to restructure the paper ..."*
> >
> > *"Q: Can the authors provide a clear algorithmic summary or flowchart ... and clarify how they interact within the overall training pipeline?"*
>
> We thank the reviewer for their valuable feedback. We will continually polish our manuscript to improve the manuscript's readability.
>
> However, we would also like to clarify that we **have already provided** the following structure:
>
> - **Figure 1** already provides a schematic **flowchart** of the DaaR pipeline with a detaild caption. Moreover, **Lines 207-210** serve as a summary that explicitly connects our DaaR to this figure.
> - Our contributions in lines 53-60 are structurally aligned with **Sections 3, 4, and 5** respectively.
> - We also use **transition paragraphs** (lines 95-97, 173-178, 201-210 and 269-272) in every sections to ensure a **logical flow**.
>
> We agree that the presentation can be clearer. In the revised version, we will make two key improvements:
>
> 1. **Add a pseudo-code algorithm** for DaaR to offer a clear, step-by-step guide.
> 2. **Strengthen the textual mapping** between our method description (Sec. 4) and the components in the Figure 1 flowchart.
>
> ---
>
> ### **Closing Remark**
> Thank you again for your insights and feedback! We hope these responses and the new revision can address your concerns and we would be grateful if you would consider re-evaluating your assessment of our work.

---

> > ### Comment · Reviewer_Bbh9 · 2025-08-07
> >
> > Thank you for the thoughtful rebuttal. This direction is promising, particularly in scenarios where explicit domain labels are unavailable and pseudo-labels must be inferred in a principled way.
> >
> > Some parts of the methodological flow could benefit from greater clarity. For example, the setup in Section 2.2 might be easier to follow with a clearer explanation of whether domains are assumed to be disjoint partitions, and how "text descriptions" and "domain labels" are conceptually and functionally differentiated. As it stands, the terms seem to overlap, for instance, a string like “Wikipedia” could reasonably serve as both, which may cause confusion. This becomes more important given that domain partitions are directly used later (e.g., in Section 3.2 to compute centroids), suggesting an implicit assumption about the structure of the data.
> >
> > The use of clustering to infer domain structure in the absence of labels is a valuable idea, albeit not completely new. Once pseudo-labels are derived in this way, the application of reweighting schemes like DoReMi or RegMix seems plausible, which raises the question of how the proposed diversity probing mechanism meaningfully differs from or improves upon those existing methods. A deeper comparison here would help clarify the benefit of the proposed method.
> >
> > In terms of framing, the paper appears to derive its novelty more from the overall pipeline than from any individual component. This is a valid approach, but to make the case more compelling, it would help to refine the narrative and structure. For instance, simplifying and sharpening Figure 1, including a formal algorithm box to walk through the full procedure, and streamlining the presentation could significantly improve accessibility.
> >
> > Given that the novelty of the individual components appears limited, the primary contribution must lie in the pipeline as a whole. However, the current evaluation remains somewhat limited in scope, which for instance only covers few main reasoning benchmarks, is not broad enough to fully substantiate the utility of the proposed pipeline.

---

> > > ### Author Response · Authors · 2025-08-08
> > > **(1/2) Reply to Reviewer Bbh9's Follow-up Questions**
> > >
> > > Dear Reviewer Bbh9
> > >
> > > Thank you for your detailed and constructive follow-up. We are grateful for your deep engagement with our work. Your questions are exceptionally insightful, and they allow us to provide a much deeper clarification of our work's core principles and contributions.
> > >
> > > We believe the core of the discussion lies in two areas: (1) The precise definition and novelty of our problem setting and methodology, and (2) The presentation and substantiation of our pipeline.
> > >
> > > ---
> > >
> > > ### **1. On the Core Contribution**
> > >
> > > We agree that the main novelty lies in the **overall pipeline**. More fundamentally, we argue it introduces a new paradigm specifically designed to solve the challenging, real-world **"domain-undetermined" problem**.
> > >
> > > **1.1 Clarifying the Problem Setting**
> > >
> > > Thank you for your careful reading. We will re-examine the definitions and terminology throughout the manuscript to ensure they are immediately clear to the reader upon first appearance. The term "domain," in particular, is used in line with its classic application in post-training data selection, referencing a series of works on cross-domain data selection [1,2].
> > >
> > > You have correctly identified a potential confusion between terms like "domain labels" and "text descriptions." This ambiguity is not an oversight in our writing, but rather **a central challenge our work is designed to address**. The distinction is critical:
> > >
> > > - **Domain Labels**: We define these as **coarse-grained, pre-defined, discrete source categories**. In the realistic "domain-undetermined" setting that our paper addresses, we assume these labels are unavailable, unreliable, or non-normalized.
> > > - **Text Descriptions**: These are **semantically rich summaries** that we generate for our synthetic anchor points. Their function is not to label existing data but to **create meaningful semantic anchors** in the embedding space.
> > > - **On "Disjoint Partitions"**: It is exactly what we want to figure out in our theoretical perspective. We theoretically demonstrate that the assumption of disjoint domains **is in fact a limitation**, as shown in Proposition 3.2 and lines 193-200, enforcing **hard assignments ($p(c = k^*(x)|x) = 1, \forall x \in \mathcal{D}$) leads to suboptimal data selection**. Hence our entropy-based score is designed precisely to avoid this, using soft assignments to select for high-value, ambiguous data points.
> > >
> > > We appreciate your suggestion and will add a more rigorous definition of both "domain" and "text description" in Section 2.2 (around line 87) to make this distinction explicit from the outset.
> > >
> > > [1] Beyond IID: Optimizing Instruction Learning from the Perspective of Instruction Interaction and Dependency
> > >
> > > [2] Revisiting Reinforcement Learning for LLM Reasoning from A Cross-Domain Perspective
> > >
> > > ---
> > >
> > > **1.2. A Deeper Dive: DaaR vs. a Clustering + Reweighting Paradigm**
> > >
> > > Your question about comparing DaaR to a Clustering + DoReMi/RegMix pipeline is critical, as it allows us to contrast these two fundamentally different philosophies. DaaR is an **integrated, data-centric** system designed to leverage a **dataset's intrinsic diversity**. In contrast, DoReMi and RegMix are **training-centric** techniques that rely on **external signals from a proxy model**. The differences are profound:
> > >
> > > `Intrinsic Data Signal vs. External Training Signal`: A Clustering + Reweighting approach would depend on an external training signal:
> > >
> > > - DoReMi requires a **proxy model** to iteratively update model parameters based on domain-specific loss weights.
> > > - RegMix requires training multiple **proxy models** on different data subsets and measuring their downstream regression performance.
> > >
> > > Crucially, in a post-training context, this **"signal"** would almost certainly have to come from performance on downstream tasks, introducing **a potential risk of data leakage**. DaaR, by design, operates in a closed loop, relying solely on the intrinsic diversity signal within the data itself.
> > >
> > > `Closed-Loop Data Selection vs. Open-Loop Iterative Training`: Integrating DoReMi and RegMix into our pipeline would be unnatural and inefficient:
> > >
> > > - Using DoReMi would necessitate training **a smaller proxy model** to provide iterative feedback for the **main model** in a open-loop process.
> > > - Using RegMix would require training **hundreds of proxy models** (e.g., 512) on data slices to gather regression signals.
> > >
> > > DaaR completes its entire data selection process in one pass, a "closed-loop" procedure internal to the data itself.
> > >
> > > `Vast Pre-training Data vs. Curated Fine-tuning Data`: Our work operates in the fine-tuning regime, which has different constraints from pre-training paradigm:
> > >
> > > - DoReMi would need **vast amount of data** to get a suitable proxy model.
> > > - RegMix would need to divide the fine-tuning dataset among hundreds of proxy models, would leave each with **only a few hundred examples**—hard to yield a meaningful signal for regression.
> > >
> > > ---
> > >
> > > Continue...

---

> > > > ### Author Response · Authors · 2025-08-08
> > > > **(2/2) Reply to Reviewer Bbh9's Follow-up Questions**
> > > >
> > > > ### **2. On Presentation and Substantiating the Pipeline's Utility**
> > > >
> > > > We fully agree that the pipeline's presentation must be impeccable and its evaluation rigorous.
> > > >
> > > > **2.1 Improving Presentation:**
> > > >
> > > > We accept your critique and will make concrete improvements based on your excellent suggestions:
> > > >
> > > > 1. We will add a formal **pseudo-code algorithm** to provide a clear, step-by-step implementation guide.
> > > > 2. We will redesign **Figure 1** to be more streamlined and ensure its components directly correspond to the steps in the new algorithm, creating a seamless narrative.
> > > >
> > > > **2.2 On the Scope and Rigor of Evaluation:**
> > > >
> > > > We fully agree that a more extensive evaluation could further validate our method's utility. However, we wish to clarify that our core objective was to assess the **comprehensive performance of LLMs across domains** (line 23, 109, 262, 292), with a primary focus on the **overall average**. To this end, we selected 1-2 representative downstream tasks for each domain.
> > > >
> > > > For instance, Table 2 reveals that while some baselines achieve strong results on a specific domain, they likely **fail to deliver a high average performance**. In addition, we validated DaaR on the more OOD **MMLU benchmark** (Table 3(a)) and the **customization capabilities** (Table 3(c)), all of which were efforts to demonstrate the **comprehensive utility** of our pipeline.
> > > >
> > > > We are encouraged that the rigor and depth of our experimental setup have been independently recognized. Across the reviews, our evaluation has been described as **"comprehensive"** and **"thorough,"** demonstrating **"empirical rigor"** by establishing a **"new state-of-the-art average performance"** with **"consistent SOTA results."** The experiments were further noted as being **"well-conceived,"** with performance gains that are **"particularly impressive on high-difficulty STEM tasks."** This feedback assures us that our evaluation, while focused, is sufficiently robust to substantiate our claims. We will, of course, frame the extension to broader domains as a promising direction for future work in our limitations section.
> > > >
> > > > ---
> > > > ### **Closing Remarks**
> > > >
> > > > Once again, we thank you for your professional and highly valuable feedback, which has helped us to significantly sharpen the articulation of our work. We believe this detailed breakdown, combined with our committed revisions, fully addresses your concerns and clearly demonstrates that our paper offers a novel, significant, and well-validated contribution. We sincerely hope you will reconsider your assessment.

---

### Official Review · Reviewer_Ge2Q · 2025-06-30

**Clarity:** 3
**Significance:** 4
**Originality:** 4
**Rating:** 5
**Confidence:** 3

**Summary:**

This paper addresses the challenge of fine-tuning Large Language Models (LLMs) on diverse, mixed-domain data, especially when domain labels are missing or imprecise. The authors first empirically demonstrate that optimal data diversity levels vary across models and that existing metrics fail in label-free settings. Based on these insights, they propose a new self-supervised framework, DAAR (Diversity as a Reward), which requires no domain labels. DAAR gives the LLM a dual identity: an "output model" that uses a lightweight probe to select data based on a predicted semantic entropy (diversity) reward, and an "input model" that is then fine-tuned on this selected data. This process is guided by automatically synthesized, model-aware domain centroids. Extensive experiments on various LLMs, including the Llama3.1 and Qwen2 series, show that DAAR significantly boosts performance on domain-undetermined data and foundational tasks, notably improving mathematical reasoning and coding capabilities more than nine other baseline methods.

**Questions:**

- The out-of-distribution evaluation on MMLU shows that DAAR achieved the top score on Qwen2-7B but was outperformed by other baselines on Qwen2.5-7B. Do the authors have a hypothesis for this performance variance between different models in the OOD setting?

**Ethical Concerns:**

["NO or VERY MINOR ethics concerns only"]

**Final Justification:**

The author addressed my comments about compounding errors and robustness. They also included results with other family of models - Qwen2.5-14B and LLaMA3.1-8B to show generalizability. This makes the paper stronger. I maintain my recommendation of accept.

**Limitations:**

Yes

**Quality:**

3

**Strengths And Weaknesses:**

The paper has the following key strengths:
- Novel and theoretically grounded: The proposed framework addresses an important real-world problem of fine-tuning on domain-undetermined data. Unlike methods that require pre-existing domain labels, DAAR uniquely uses the LLM's own embedding space geometry to create a diversity reward signal, giving the model a dual identity as both the data selector and the model-to-be-tuned. This approach is not merely heuristic; it is supported by both controlled empirical analysis and a theoretical perspective based on importance sampling.
- Comprehensive evaluation: The framework establishes a new state-of-the-art average performance across three model families and seven benchmarks, consistently outperforming nine different baseline methods. The performance uplift is particularly impressive on high-difficulty STEM tasks where many baselines fail; for example, it achieves notable improvements in mathematical reasoning (+27%) and coding (+7.4%).
- Computationally efficient and generalizable: By operating on frozen LLM embeddings with a lightweight MLP probe, the framework is highly efficient, achieving 70% lower GPU usage and 2.5x faster inference compared to baselines that rely on costly GPT-based evaluators or full-LLM inference. The method also demonstrates strong generalizability, maintaining top performance on the out-of-distribution MMLU benchmark and on a different model architecture (Qwen3-8B).

The paper has the following weaknesses which when addressed can further strengthen the paper:
- Complex framework with potential for compounding errors: The framework is multi-staged and complex. It depends on a pipeline that includes LLM-based data synthesis to create domain centroids, k-means clustering to generate pseudo-labels, and a two-stage process to train the reward probe. An error or instability in any of the early stages, such as the initial centroid generation, could propagate through the entire system and negatively impact the final data selection. The paper does not provide a thorough analysis of the framework's robustness to such potential compounding errors.
- Limited scope of models: The experiments are primarily conducted on models within the 7-8B parameter range. It remains unclear how well the findings and the method's effectiveness would generalize to significantly larger models (for example, 70B or more) or different model architectures.

---

> ### Author Rebuttal · Authors · 2025-07-31
>
> Thanks for your valuable feedback! We are deeply encouraged by your recognition of our work's **novelty** and **comprehensive evaluation**. We also greatly appreciate your insightful suggestions, we will address each of your Weaknesses (W) and Question (Q) below.
>
> ---
>
> ### **W1: On the Compounding Errors and Robustness of DaaR**
> > *'Complex framework with potential for compounding errors: The framework is multi-staged and complex. ... An error or instability in any of the early stages, such as the initial centroid generation, could propagate through the entire system ...'*
> >
>
> This is a very insightful point. You are correct that since DaaR operates as a multi-stage pipeline, a thorough analysis of its stability is crucial. We did touch upon this consideration in the main text (**lines 266-267**) and provided an initial stability analysis in **Appendix G.8 (lines 895-904)**.
>
> In summary, our existing analysis suggests that the key components of DaaR **exhibit robustness**: the LLM-based seed generation is consistent, the k-means clustering is stabilized by deterministic initialization, the reward probe training converges reliably, and the final data selection strategy has built-in redundancy.
>
> However, your observation that we should **more explicitly analyze the potential for compounding errors** is well-taken. To thoroughly address this concern and directly validate the robustness of our framework against error propagation, we have conducted the following new analyses based on **three fully independent experimental runs**.
>
> - **[Experiment Setup]**
>
> We performed a **third independent run**, supplementing the current two independent runs mentioned in *lines 112-113*. Each run is an **end-to-end execution** of our pipeline. We present results for Qwen2-7B as a representative case; other models showed similar trends and will be added to the paper.
>
> - **[Observation 1]: Robustness of Centroid Generation.**
>
> To evaluate the stability of the critical **initial stage**, we computed the pairwise cosine similarities of the **final centroid embeddings** from the three runs.
>
> | Domains | Similarity (Seed-1-2) | Similarity (Seed-1-3) | Similarity (Seed-2-3) |
> |:---:|:---:|:---:|:---:|
> | Common Sense | 0.9912 | 0.9895 | 0.9921 |
> | Reasoning | 0.9845 | 0.9911 | 0.9887 |
> | Mathematics | 0.9880 | 0.9803 | 0.9854 |
> | Coding | 0.9905 | 0.9853 | 0.9918 |
>
> The centroid embeddings are **remarkably similar** (all **>0.98**), which demonstrate the stability of our initial centroid generation, showing it is **not a significant source of variance**, especially compared to the value in Figure 11 and Table 11.
>
> - **[Observation 2]: Stability of Final Data Selection.**
>
> We then examined if initial variations propagate to the **final data selection** by calculating the **data overlap** for the top 20% (8000) of selected samples across the three runs.
>
> | Selected Data | Overlap (Seed-1-2) | Overlap (Seed-1-3) | Overlap (Seed-2-3) |
> |:---:|:---:|:---:|:---:|
> | Rate | 95.7% | 97.3% | 96.1% |
>
> The results show an **significantly high overlap**, with an average of **96.4%**, confirming that our data selection process is robust and largely unaffected by initial variations.
>
> - **[Observation 3]: Consistency of Final Performance.**
>
> Finally, the stability of the entire pipeline is reflected in the **end-task benchmark performance**.
>
> | Runs | nq | triviaqa | hellaswag | gsm8k | math | mbpp | humaneval | Avg |
> | :--- | :---: | :---: | :---: | :---: | :---: | :---: | :---: | :---: |
> | Seed-1 | 17.92 | 56.78 | 72.91 | 75.00 | 39.60 | 51.40 | 64.80 | 54.06 |
> | Seed-2 | 15.84 | 58.38 | 73.14 | 75.80 | 36.60 | 52.60 | 65.07 | 53.92 |
> | **Seed-3** | 15.18 | 57.96 | 72.99 | 76.70 | 38.80 | 51.40 | 65.24 | 54.04 |
> | Average | 16.31 ± (**1.43**) | 57.71 ± (**0.83**) | 73.01 ± (**0.12**) | 75.83 ± (**0.85**) | 38.33 ± (**1.55**) | 51.80 ± (**0.69**) | 65.04 ± (**0.22**) | 54.01 ± (**0.08**) |
>
> The final results are **highly consistent**, culminating in an average score with a very small standard deviation of just **0.08**. This end-to-end stability provides strong empirical evidence that our framework is robust. (Complete results can be found in our reply to W4 & Q4 of Reviewer Bbh9).
>
> In summary, and following your valuable advice, we will make two key revisions to the paper:
>
> 1. We will integrate a more **prominent discussion** of the framework's stability, currently in the appendix, into the main body of the paper.
> 2. We will **incorporate the new ablation experiments** described above into our ablation studies (Section 5.3) to explicitly address the concern of compounding errors.
>
> ---
>
> ### **W2: On the Limited Scope of Models**
> > *'Limited scope of models: The experiments are primarily conducted on models within the 7-8B parameter range. It remains unclear .. larger models (for example, 70B or more) or different model architectures.'*
> >
>
> Thank you for your constructive suggestion! We agree that demonstrating DaaR's effectiveness across a wider range of model architectures and scales would significantly strengthen our claims.
>
> Our current experiments **already span three distinct architectural families**: the widely-used Llama and Qwen-2.x architecture, as well as the latest SOTA Qwen-3 architecture. Across all these, DaaR has consistently demonstrated its effectiveness.
>
> However, we acknowledge that our original study did not include models of different sizes. Given the *considerable time and computational resources* required to run our full suite of experiments (10 diversity-controlled settings and 12 baselines with 2 independent runs), we conducted a more targeted set of experiments on two additional models: **Llama3.2-3B** and **Qwen2.5-14B**.
>
> For these experiments, we compared DaaR against two key baselines: **Raw** and **Random** (a strong baseline in our existing results). Our supplementary results are as follows:
>
> - **On Qwen2.5-14B (a larger model)**
>
> | Method | NQ | TriviaQA | Hellaswag | GSM8K | MATH | MBPP | HumanEval | Avg |
> | :---: | :---: | :---: | :---: | :---: | :---: | :---: | :---: | :---: |
> | Raw | 10.89 | 66.56 | 76.86 | 86.00 | 19.70 | 5.40 | 78.66 | 49.15 |
> | Random | 20.06 | 65.80 | 76.76 | 86.40 | 36.90 | 64.20 | 75.46 | 60.80 |
> | DaaR | 19.73 | 66.26 | 77.19 | 85.00 | 39.10 | 66.80 | 76.69 | **61.54** |
>
> - **On LLaMA3.1-8B (a smaller model):**
>
> | Method | NQ | TriviaQA | Hellaswag | GSM8K | MATH | MBPP | HumanEval | Avg |
> | :---: | --- | --- | --- | --- | --- | --- | --- | --- |
> | Raw | 7.62 | 53.1 | 68.77 | 26.6 | 4.5 | 3.8 | 23.17 | 26.79 |
> | Random | 16.03 | 53.56 | 68.46 | 28.7 | 6.15 | 4.65 | 29.12 | 29.52 |
> | DaaR | 15.13 | 53.79 | 68.23 | 29.10 | 5.63 | 5.3 | 30.34 | **29.64** |
>
> As the new results demonstrate, DaaR **consistently outperforms** on both the larger Qwen2.5-14B and the smaller Llama3.2-3B. On Qwen2.5-14B, DaaR achieves the **highest average score** (61.54), marking an improvement over the strong Random baseline with **0.74**. Similarly, on Llama3.2-3B, DaaR (29.64) maintains its **advantage**.
>
> These results provide an evidence that our method is effective **across both LLM architectures and model scales**. We will incorporate these findings as an additional ablation study in our revised manuscript.
>
> ---
>
> ### **Q1: Conjecture on the Performance Variance on MMLU**
> > *'The OOD evaluation on MMLU shows that DAAR achieved the top score on Qwen2-7B but was outperformed by other baselines on Qwen2.5-7B. Do the authors have a hypothesis ...?'*
> >
>
> That is an excellent observation, it is indeed interesting that DaaR places second on Qwen2.5-7B. While a definitive cause warrants deeper investigation, we offer two **primary hypotheses** based on our analysis:
>
> - **Domain Alignment vs. Diminishing Gains**: Our detailed results in Table 6 shows DaaR's strength on Qwen2-7B is largely driven by **STEM subjects**, which align with the core domains (math, code, etc.) our method prioritizes. On the more capable Qwen2.5-7B, DaaR still **effectively boosts STEM performance**. However, for an already strong LLM, the marginal gains on less-related domains (e.g., Humanities) may diminish, leading to a slightly lower overall average compared to a method like SuperFilter that might have a different selection bias.
> - **LLM Sensitivity**: The weaker base model (Qwen2-7B) appears **more sensitive to data selection**, exhibiting larger performance variance across all methods. Conversely, on the stronger Qwen2.5-7B, the performance of all baselines is more stable and converges. In this less volatile regime, the strong, diversity-driven focus of DaaR on STEM might introduce a **slight trade-off in other areas**, resulting in a marginal difference.
>
> We believe the **OOD setting** is a fascinating area for future work and would welcome any further discussion.
>
> ---
>
> ### **Closing Remarks**
> Once again, we sincerely thank you for your recognition and invaluable feedback. We hope our responses have thoroughly addressed your concerns regarding robustness and model scale.
>
> We kindly hope that these clarifications and additions might further strengthen your confidence in our work, your support is immensely encouraging to us. If you have any additional concerns or queries, we warmly invite you to share them with us.

---

### Official Review · Reviewer_bjgP · 2025-07-01

**Clarity:** 3
**Significance:** 3
**Originality:** 3
**Rating:** 5
**Confidence:** 4

**Summary:**

This paper addresses the challenge of fine-tuning large language models (LLMs) on domain-undetermined data where labels are missing or imprecise, proposing a self-supervised framework called DAAR (Diversity as a Reward). DAAR leverages diversity as a reward signal, constructing model-aware domain centroids through iterative embedding-space generation, training a lightweight MLP probe to predict semantic entropy from frozen embeddings, and implementing closed-loop fine-tuning where the model’s diversity estimates guide data selection. Theoretical analysis shows traditional data mixing strategies fail in label-free settings, necessitating simultaneous maximization of mutual information and preservation of the base model’s feature geometry.

**Questions:**

Sec. 3.4 states DAAR’s strategy "must simultaneously maximize mutual information," but no proof connects Eq. 5–6 to DAAR’s mechanics. Please formalize how $I(X; C)$ (mutual info between samples and domains) is maximized.  And clarify if $\psi_{\text{div}}$’s entropy prediction (Eq. 9) directly optimizes $I(X; C)$.

Phase 1 of centroid synthesis (Sec. 4.1) uses "minor injection from downstream task samples" to generate seeds. This risks contaminating pseudo-labels with test-set knowledge.
Please: Report ablation results where seeds are generated without downstream task data.

**Ethical Concerns:**

["NO or VERY MINOR ethics concerns only"]

**Final Justification:**

The author addressed my concerns about data leakage in downstream tasks through supplementary experiments, and partially addressed my concerns about the theoretical interpretation of "mutual information". So I am considering raising the score to 4 points.
The author's response addressed my concerns and I will consider raising my score to 5

**Limitations:**

yes

**Quality:**

3

**Strengths And Weaknesses:**

Strengths:
1. The self-supervised framework DAAR is proposed to address domain-undetermined data fine-tuning by leveraging diversity as a reward signal, breaking free from reliance on pre-labeled data. Theoretical analysis proves that optimal data selection in label-free scenarios requires maximizing mutual information while preserving the base model’s feature geometry, providing a solid basis for the approach. The "dual-identity model" (output probe + input fine-tuning) and lightweight MLP probe design enable efficient computation by predicting semantic entropy from frozen embeddings.
2. Experiments across 7 cross-domain benchmarks (e.g., mathematical reasoning, coding) and 3 model families (Qwen2, Llama3.1, etc.) show DAAR outperforms 9 baselines. Also, ablation studies on hyperparameters (seed size, sliding window) validate component stability, and generalizability is demonstrated in out-of-distribution scenarios (MMLU) and novel architectures (Qwen3).

Weaknesses:
1. The core claim—mutual information maximization underpins DAAR—lacks explicit formalization or proof (mentioned in Sec. 3.4 but not elaborated).
2. Pseudo-labels for clustering incorporate downstream task samples (Sec. 4.1, Phase 1), potentially leaking test-set information. No ablation validates performance without this.

---

> ### Author Rebuttal · Authors · 2025-07-31
>
> Thank you for your recognition of our method's design and comprehensive experiments. We will address your concerns regarding the Weaknesses (W) and Questions (Q) one by one.
>
> ---
>
> ### **W2 & Q2: On Potential Data Leakage**
> > *'W: Pseudo-labels for clustering incorporate downstream task samples (Sec. 4.1, Phase 1), potentially leaking test-set information...'*
> >
> > *'Q: ... This risks contaminating pseudo-labels with test-set knowledge. Please: Report ablation results where seeds are generated without downstream task data.'*
> >
>
> We appreciate your rigorous scrutiny of our experimental methodology. We want to **assure you that our design explicitly prevents data leakage**, and we welcome the opportunity to clarify this with four clarifications.
>
> **`Strict Data Partitioning`**
>
>  First and foremost, we must emphasize that our protocol maintains **strict data isolation**. The few downstream samples used for seeding (2 in 5) are **completely separate** from the evaluation benchmark, as well as the decontamination data that used for final LLM fine-tuning (lines 642-643).
>
> **`Purpose of the Injection`**
>
> We posit that DaaR's performance **is not dependent on** these injected samples. Their role is solely to provide an initial diversity signal that **accelerates the diversity augmentation process** in Phase 2. Any information from the injection is heavily diluted in the subsequent large-scale seed generation and centroid computation stages.
>
> **`New Ablation Study (w/o Injection)`**
>
> To empirically validate this and directly address your request, we conducted a new ablation study where we **removed the minor injection entirely** while keeping the benchmark evaluation data same as current setting. We report on three key metrics:
>
> - **Centroid Similarity:** Taking Qwen2-7B as a representative example, the domain centroids generated with and without injection are nearly identical, achieving a cosine similarity of **avg over 0.987**, demonstrating that the injection has a **negligible impact** on the domain centroid representations.
>
>     | Domains | Similarity (w/ - w/o Injection) |
>     |:---:|:---:|
>     | Common Sense | 0.9922 |
>     | Reasoning | 0.9865 |
>     | Mathematics | 0.9843 |
>     | Coding | 0.9878 |
>
> - **Final Performance:** The end-task performance of DaaR (w/o injection) is **statistically indistinguishable** from our reported results, demonstrating that DaaR is not dependent on the minor injection. The results are as follows:
>
>     | Llama3.1-8B | nq | triviaqa | hellaswag | gsm8k | math | mbpp | humaneval | Avg |
>     | :--- | :---: | :---: | :---: | :---: | :---: | :---: | :---: | :---: |
>     | DaaR w/ Inject | 20.08 | 64.55 | 74.88 | 54.80 | 15.30 | 4.70 | 37.50 | 38.83 |
>     | DaaR w/o Inject | 20.39 | 64.80 | 76.05 | 55.40 | 13.65 | 5.75 | 36.48 | 38.93 |
>
>     | Qwen2-7B | nq | triviaqa | hellaswag | gsm8k | math | mbpp | humaneval | Avg |
>     | :--- | :---: | :---: | :---: | :---: | :---: | :---: | :---: | :---: |
>     | DaaR w/ Inject | 16.88 | 57.58 | 73.03 | 75.40 | 38.10 | 52.00 | 64.94 | 53.99 |
>     | DaaR w/o Inject | 16.22 | 58.33 | 73.41 | 75.20 | 36.37 | 52.95 | 65.14 | 53.95 |
>
>     | Qwen2.5-7B | nq | triviaqa | hellaswag | gsm8k | math | mbpp | humaneval | Avg |
>     | :--- | :---: | :---: | :---: | :---: | :---: | :---: | :---: | :---: |
>     | DaaR w/ Inject | 15.83 | 58.65 | 72.48 | 80.20 | 16.70 | 64.20 | 68.29 | 53.76 |
>     | DaaR w/o Inject | 14.91 | 58.32 | 72.55 | 79.70 | 16.30 | 63.70 | 70.65 | 53.73 |
>
> - **Generation Efficiency:** However, the ***diversity augmentation*** process became **significantly less efficient.** Without the initial seed diversity, it required **approximately 4 times** more generation attempts to meet the same diversity threshold (as in Table 14), confirming its role as an **accelerator**.
>
>     | Domains | Attempts (w/ Injection) | Attempts (w/o Injection) |
>     |:---:|:---:|:---:|
>     | Common Sense | 31 | 134 |
>     | Reasoning | 61 | 248 |
>     | Mathematics | 236 | 819 |
>     | Coding | 21 | 105 |
>
> **`Context in Related Literature`**
>
> This practice of using a small, targeted set to guide a broader data selection process is **a well-established strategy** in related work, such as gradient-based influential selection LESS [1] and targeted-distribution importance resampling DSIR [2]. While our method differs, the **principle of a minor, guided injection** to improve efficiency will not lead to the leakage.
>
> We thank you again for raising this critical point. We will **add this new, detailed ablation study to our revised manuscript**, which we believe will greatly enhance its rigor and transparency.
>
> [1] LESS: Selecting Influential Data for Targeted Instruction Tuning, ICML'2024
>
> [2] Data Selection for Language Models via Importance Resampling, NeurIPS'2023
>
> ---
>
> ### **W1 & Q1: On the Formalization of the Theoretical Underpinning**
> > *'W: The core claim—mutual information maximization underpins DAAR—lacks explicit formalization or proof (mentioned in Sec. 3.4 but not elaborated).'*
> >
> > *'Q: Sec. 3.4 states DAAR’s strategy "must simultaneously maximize mutual information," but no proof connects Eq. 5–6 to DAAR’s mechanics. Please formalize ... is maximized. And clarify if entropy prediction directly optimizes...'*
> >
>
> We are very grateful for this deep engagement with our theoretical framework, and we appreciate the opportunity to clarify our reasoning and correct an imprecision in our manuscript.
>
> First, we wish to clarify the **intended role of our theoretical analysis** (lines 173-178). Its primary purpose is to provide a **principled perspective** that explains our empirical observations in Section 3 and offers a strong motivation for the design of DaaR in Section 4.
>
> **`Regarding "Mutual Information Maximization"`**
>
> We acknowledge that our use of the term "mutual information maximization" in the introduction section was an **imprecise statement**. Our initial intention was to use this term to conceptually capture the goal of enriching sample-level diversity. However, we recognize this is **not a formal maximization of I(X;C)** in information theory and thank you for pointing it out. We will immediately correct this terminology throughout the revised manuscript.
>
> Instead, the theoretical foundation of our work is rooted in **Importance Sampling**, as detailed in Section 3.4 and derived in Appendix E. This framework allows us to analyze how data diversity impacts the optimality of data selection.
>
> **`Clarifying the Link Between Theoretical Insight and Entropy-based Method (Eq. 9)`**
>
> The connection between our theory and DaaR's entropy-based selection is best formalized as **a principled heuristic**, which we present in the following remark:
>
> - **Remark:** Our analysis in Sec. 3.4 shows that the importance weight w(x) = q(x)/p(x) for data selection is **suboptimal** when the domain assignment is deterministic, i.e., p(c=k*|x) = 1 for some domain k*. This condition is equivalent to the conditional entropy H(C|X=x) = 0. Since directly optimizing for the ideal q(x) is **intractable in our domain-undetermined** setting, DaaR adopts a theory-guided heuristic to select samples x that are maximally distant from this known suboptimal condition. We use H(C|X=x) as a direct **proxy for this distance**. Therefore, the selection objective of DaaR is to argmax_x H(C|X=x), which is implemented by selecting samples with high predicted entropy ψ_div(x) ≈ H(C|X=x).
>
> This remark connects our **theoretical insight**—that zero conditional entropy is suboptimal—to our **practical method** of rewarding high predicted entropy.
>
> Thank you once again for your invaluable feedback. We will take the following actions in our revision:
>
> 1. We will **remove all imprecise mentions** of "mutual information maximization" and accurately describe our theoretical basis as Importance Sampling.
> 2. We will incorporate the **formal discussion and remark** above to explicitly bridge the gap between our theoretical insights and the design of our method.
>
> ---
>
> ### **Closing Remarks**
> Thank you once again for your valuable feedback, we have endeavored to address each of your concerns thoroughly in our responses and planned revisions.
>
> We hope these clarifications have resolved your initial concerns, we would be very grateful if you would consider reconsidering your assessment of our paper. We look forward to any further discussion and welcome any additional questions you may have.

---

> ### Comment · Reviewer_bjgP · 2025-08-07
>
> Dear Authors,
>
> First, the authors experimentally demonstrate that not using downstream task data as seed data can achieve comparable results, albeit potentially increasing clustering costs. These experiments are well-conceived and effectively address my concerns on this point.
>
> Regarding the mutual information question, the authors decided to remove the imprecise mentions of "mutual information maximization" and accurately describe their theoretical basis as Importance Sampling, stating: “We will remove all imprecise mentions of "mutual information maximization" and accurately describe our theoretical basis as Importance Sampling.” I consider this a rigorous and responsible approach. Having “temporarily set aside” the influence of the mutual information concept, I carefully re-examined the manuscript and the authors' response, leading to three new questions:
>
> In the proposed work, a classifier $\psi_{dom}$ with an LLM backbone is first used to predict pseudo-labels, thereby obtaining predictions $p(c(x)=k|x)$, and then the predictive entropy of this classifier is calculated. Subsequently, the authors train a separate regression predictor $\psi_{div}$ using the same backbone. I don't understand why $\psi_{dom}$ isn't used directly to compute the domain probability distribution of the training data and then apply the entropy calculation formula (Equation 9) to obtain the sample entropy. Adding an extra model training step seems unnecessary.
>
> Based on the authors' response, the logic for performance gain stems from the statement: “importance weight $w(x) = q(x)/p(x)$ for data selection is suboptimal when the domain assignment is deterministic”, thus inferring that “DaaR adopts a theory-guided heuristic to select samples x that are maximally distant from this known suboptimal condition.” The essence of this logic appears to be: since A is known not to be optimal, maximizing the distance from A should lead to something better. This reasoning doesn't seem entirely robust to me. My concern would be alleviated if the authors could propose a more solid theoretical explanation for the foundation of this work.
>
> In my understanding, $M(x)$ can be viewed as embeddings of LLM-generated training samples. Using k-means in this embedding space yields semantic cluster centroids that integrate the training data distribution with the LLM's intrinsic features. These centroids are then used as labels to train the classifier, after which samples with maximum entropy are sought. This process can be interpreted as seeking samples far from the k-means centroids, situated at the "edges" of clusters, thereby increasing training data diversity. This aligns with the authors' statement in the last paragraph of Section 4.2 "Data Selection": “Building on the theoretical insights in Sec. 3.4, data points that are closer to other centroids and more dispersed within their own centroid are more beneficial for enhancing the comprehensive capabilities of the model.” My question is: could this method produce an adverse extreme effect, i.e., excessively focusing on marginal data while neglecting data near the semantic cluster centroids (which are typically considered most representative of the cluster)? Perhaps a sampling method that balances both marginal and central data would yield better results?
>
> Regardless, the methodology employed in this work holds significant practical value, and the experimental section is thorough. I will consider raising my score and look forward to the authors' response to my new questions, which I will consider for a further increase in my score.

---

> > ### Author Response · Authors · 2025-08-07
> > **(1/2) Reply to Reviewer bjgP's Follow-up Questions**
> >
> > Dear Reviewer bjgP
> >
> > Thank you very much for your thoughtful and constructive engagement with our work. We are genuinely grateful for your positive feedback and the insightful new questions you've raised. Your careful re-examination has helped us identify key areas for clarification and improvement. We believe addressing these points will substantially strengthen our manuscript.
> >
> > Below, we address each of your new questions in detail, and we hope our responses, along with new supporting evidence, will fully resolve your concerns.
> >
> > ---
> >
> > ### **1. On the Necessity of the Stage 2 Diversity-Rewarding Probe**
> >
> > Thank you for this excellent question, which allows us to clarify the motivation behind our two-stage design. Our primary goal was to build a lightweight probe to explicitly reward data diversity from Eq (2)(3). While our initial attempts to directly use diversity formulas failed (Appendix F.4), we found that entropy served as a viable proxy (L235-240). The Stage 2 module was thus **a natural step to create this dedicated and lightweight "diversity probe"**, aligning with our central goal of explicitly rewarding diversity.
> >
> > Subsequently, we conducted an ablation study to test the alternative you suggested. The results of directly using the Stage 1 outputs for entropy-based selection are as follows:
> >
> > | Qwen2-7B | nq | triviaqa | hellaswag | gsm8k | math | mbpp | humaneval | **Avg** |
> > | :--- | :---: | :---: | :---: | :---: | :---: | :---: | :---: | :---: |
> > | DaaR w/o Stage-2 | 15.86 | 57.96 | 72.93 | 75.40 | 37.20 | 50.80 | 65.05 | 53.60 |
> > | DaaR (Ours) | 16.88 | 57.58 | 73.03 | 75.40 | 38.10 | 52.00 | 64.94 | **53.99** |
> >
> > As the results indicate, this simplified method consistently **outperforms other baselines** and **achieves competitive performance compared to our full DaaR** framework. To better understand the slight performance gap, We plotted histograms by dividing the value range into 8 equal intervals and counting the samples in each:
> >
> > | Interval | 0.1 | 0.2 | 0.3 | 0.4 | 0.5 | 0.6 | 0.7 | 0.8 |
> > | :--- | :---: | :---: | :---: | :---: | :---: | :---: | :---: | :---: |
> > | True Entropy | **28716** | 4159 | 2348 | 1956 | 2140 | 487 | 170 | 24 |
> > | Predicted Entropy | 24377 | 5012 | 2553 | 2780 | 1368 | 1166 | 1242 | **1502** |
> >
> > The result reveals that the distribution of the directly calculated "True Entropy" is **heavily skewed towards lower values**. In contrast, the "Predicted Entropy" from the Stage 2 probe is significantly smoother and more uniformly distributed. Based on this observation, we **hypothesize** that the Stage 2 regression model acts as a **smoothing or normalization function** on the raw entropy signal. This smoothing effect may contribute to a more robust and balanced data selection.
> >
> > We thank you for pushing us on this point. We will add this ablation study and analysis to the appendix to provide a comprehensive justification for our design choice.
> >
> > ---
> >
> > ### **2. On the Theoretical Foundation**
> >
> > We are extremely grateful for this critical feedback. You have identified a key point where our manuscript could be more rigorous. Your concern about the logic being heuristic is completely valid, and your prompt has pushed us to formalize the connection between our theory and method, which we believe significantly strengthens the paper.
> >
> > As you rightly pointed out, our previous rebuttal established that a deterministic domain assignment is suboptimal. The missing piece was a formal link explaining *why* selecting high-entropy samples is a principled way to move towards optimality. We now provide this link by analyzing the approximation error.
> >
> > **Proposition 3.X (Approximation Error of Deterministic Assignment).** Let $w(x) = \sum_k \lambda_k \cdot p(c=k|x)$ be the true importance weight and $w_{approx}(x) = \lambda_{k^\*(x)}$ be the approximate weight under a deterministic assignment, where $k^\*(x) = \text{argmax}(p(c=k|x))$. The squared approximation error, $\text{Err}(x) = (w(x) - w_{\text{approx}}(x))^2$, is:
> >
> > $$ \text{Err}(x) = \left( \sum_{j \neq k^\*(x)} (\lambda_j - \lambda_{k^\*(x)}) \cdot p(c=j|x) \right)^2 $$
> >
> >
> > ***Brief Derivation:*** The error term simplifies to $\sum_k (\lambda_k - \lambda_{k^*(x)}) \cdot p(c=k|x)$, which is non-zero only for non-dominant classes, yielding the expression above.
> >
> > ***Implication:*** This proposition shows that the approximation error is a direct function of the probability mass on non-dominant domains. The error is maximized precisely when probability is spread across multiple domains—the very definition of high predictive entropy.
> >
> > Therefore, our strategy of selecting high-entropy samples is not simply a heuristic to move "away" from a suboptimal point. It is a **principled approach to preferentially select samples where the deterministic approximation is most erroneous, and consequently**, where the full diversity information encoded in $p(c|x)$ is most critical for accurately estimating the importance weight.
> >
> > ---
> >
> > Continue...

---

> > > ### Author Response · Authors · 2025-08-07
> > > **(2/2) Reply to Reviewer bjgP's Follow-up Questions**
> > >
> > > ### **3. On the Risk of Focusing on Marginal Data**
> > >
> > > This is another excellent and practical question. Your interpretation of our method selecting "edge" samples is correct, and your concern about neglecting representative "central" data is a crucial point to address. To investigate this trade-off empirically, we conducted a new ablation study comparing three distinct selection strategies on Qwen2-7B:
> > >
> > > *   `Top-20%` (Ours): Selecting the most diverse, "edge" samples.
> > > *   `Bottom-20%`: Selecting the least diverse, "central" samples.
> > > *   `Middle-20%`: A proxy for a balanced approach.
> > >
> > > The results reveal a fascinating trade-off:
> > >
> > > | Model | nq | triviaqa | hellaswag | gsm8k | math | mbpp | humaneval | **Avg** |
> > > | :--- | :---: | :---: | :---: | :---: | :---: | :---: | :---: | :---: |
> > > | Bottom-20% (Central) | 12.71 | 58.98 | 73.36 | 73.60 | 27.80 | 53.40 | 64.63 | 52.07 |
> > > | Middle-20% (Balanced) | 13.60 | **59.09** | **73.40** | 73.40 | 33.50 | 52.80 | 64.33 | 52.84 |
> > > | **Top-20% (Ours, Marginal)** | **16.88** | 57.58 | 73.03 | **75.40** | **38.10** | 52.00 | **64.94** | **53.99** |
> > >
> > > Your intuition was spot-on: there is a clear trade-off. Our analysis leads to two key conclusions:
> > >
> > > 1.  **Neglecting Marginal Data is Harmful:** Selecting only central data (`Bottom-20%`) leads to a **significant performance drop** (-1.92 pts avg), confirming that diverse, "edge" samples are vital for robust multi-domain learning.
> > >
> > > 2.  **Marginal Data Excels on Complex Reasoning:** While the balanced approach performs well, our proposed method (`Top-20%`) **excels specifically on the most challenging reasoning tasks, such as `gsm8k` and `math`**. We hypothesize this is because "edge" samples, which lie at the semantic boundaries between domains, are uniquely valuable for teaching the model how to integrate disparate knowledge and perform complex, multi-step reasoning. Central data, by contrast, may reinforce what the model already knows within a single domain. This aligns with prior work [1,2] on the importance of bridging domain gaps.
> > >
> > > In essence, while a balanced approach is a strong competitor, our method's focus on marginal data appears to be a more effective strategy for pushing the model's frontier on complex reasoning and problem-solving, a key goal in multi-domain fine-tuning.
> > >
> > > We agree that exploring the optimal blend of central and marginal data is a fascinating avenue for future work. Thank you for inspiring this experiment; we will add this detailed analysis to our paper.
> > >
> > > [1] Beyond IID: Optimizing Instruction Learning from the Perspective of Instruction Interaction and Dependency
> > > [2] Revisiting Reinforcement Learning for LLM Reasoning from A Cross-Domain Perspective
> > >
> > > ---
> > >
> > > ### **Closing Remarks**
> > >
> > > Once again, we sincerely thank you for your invaluable and detailed feedback. Your rigorous questions have been instrumental in helping us strengthen the theoretical underpinnings, empirical validation, and overall clarity of our work. We have incorporated these discussions and new results into our planned revisions.
> > >
> > > We hope our detailed responses can better addressed your concerns and respectfully hope that these enhancements warrant an increase in your assessment. We are, of course, happy to engage in any further discussion.

---

> ### Comment · Reviewer_bjgP · 2025-08-07
>
> The author's response addressed my concerns and I will consider raising my score.

---

> > ### Author Response · Authors · 2025-08-08
> > **Final Remarks to Reviewer bjgP**
> >
> > We sincerely appreciate your final confirmation and are grateful that our responses have resolved your concerns. This fruitful exchange has been instrumental in strengthening our work, and we will diligently implement all agreed-upon modifications in the revised manuscript.

---

### Official Review · Reviewer_kdaW · 2025-07-02

**Clarity:** 3
**Significance:** 3
**Originality:** 2
**Rating:** 4
**Confidence:** 4

**Summary:**

This paper proposes DAAR, a self-supervised framework that uses diversity as a reward signal to fine-tune LLMs on domain-undetermined data. DAAR leverages embedding-space entropy prediction for label-free data selection, achieving SOTA performance across 7 benchmarks and 3 model families, with notable gains in mathematical reasoning (+27%) and coding (+7.4%).

**Questions:**

see Weaknesses

**Ethical Concerns:**

["NO or VERY MINOR ethics concerns only"]

**Limitations:**

yes

**Quality:**

3

**Strengths And Weaknesses:**

Strengths:
1. novelty: Introduces a label-free, model-agnostic method (DAAR) that integrates diversity-driven data selection with closed-loop fine-tuning, eliminating dependency on domain labels or external models.
2. empirical rigor: Demonstrates consistent SOTA results across diverse LLMs (e.g., Qwen, Llama) and challenging tasks (e.g., MATH, HumanEval)

Weaknesses:
1. unvalidated embedding representation: Assumes embedding-layer features accurately reflect data distribution (sec 3.2) without regression/ablation experiments to verify fidelity to true semantic distribution.
2. DAAR shows minimal improvement (0.1%) for Llama3.1-8B over the best baseline (SuperFilter) in Table 2, undermining claims of "notable" superiority.
3. Lacks grounding in true domain distributions:The diversity reward signal $(H(\tilde{x}))$is entirely contingent on the accuracy of $(\psi_{\text{dom}})$, which itself depends on noisy pseudo-labels from clustering (Sec. 4.1). If $(\psi_{\text{dom}})$ misclassifies domains (e.g., confuses math-heavy coding samples), entropy becomes a meaningless proxy, propagating error to $(\psi_{\text{div}})$ and data selection.

---

> ### Author Rebuttal · Authors · 2025-07-31
>
> Thank you for your recognition of our **novelty** and **empirical rigor**. We will address your concerns regarding the Weaknesses (W) one by one.
>
> ---
>
> ### **W3: On the 'True' Domain Distributions**
> > *'Lacks grounding in true domain distributions:The diversity reward signal ... depends on noisy pseudo-labels ... If (ψdom) misclassifies domains, entropy becomes a meaningless proxy, propagating error to (ψdiv) and data selection.'*
> >
>
> Thank you for this insightful question about the stability of our method and its reliance on pseudo-labels.
>
> First, as mentioned in your `Strength 1`, we wish to emphasize the core design principle and the setting of DaaR: it is a '**closed-loop, model-aware** system designed to eliminate dependency on external domain label' (from SOTA embedding model or human annotation). Our goal is not to replicate human annotations, but to understand the data's structure from the **LLM's own perspective** (lines 239-240).
>
> To directly address your concern about "noisy" pseudo-labels versus "true" labels, we conducted a **new ablation study** as follows. We replaced our model-aware pseudo-labels from clustering **with the ground-truth (GT) human-annotated labels** (which is only possible in our controlled setting, not in DaaR's target domain-undetermined scenario) and reran the entire pipeline. We report on three key metrics:
>
> - **Probe Training Convergence:** Using ground-truth labels, the domain predictor also **converged successfully**, achieving the test accuracy of **92.7%**, confirming that both labeling schemes can allow the probe to distinct domains.
> - **Difference in Final Data Selection:** The table below shows the **overlap** in the final selected data. The divergence in the selected data confirms that pseudo-labels and ground-truth labels guide the prioritization process toward **distinct data subsets.**
>
>   | Overlap |  Pseudo vs. GT |
>   |:---:|:---:|
>   | Llama3.1-8B | 87.3% |
>   | Qwen2-7B | 83.1% |
>   | Qwen2.5-7B | 84.9% |
>
> + **Final Performance:** The most critical comparison is the end-task performance:
>
>     | Llama3.1-8B | nq | triviaqa | hellaswag | gsm8k | math | mbpp | humaneval | Avg |
>     | :--- | :---: | :---: | :---: | :---: | :---: | :---: | :---: | :---: |
>     | DaaR w/ Pseudo | 20.08 | 64.55 | 74.88 | 54.80 | 15.30 | 4.70 | 37.50 | 38.83 |
>     | DaaR w/ GT | 22.54 | 65.52 | 73.43 | 53.80 | 14.35 | 4.40 | 36.50 | 38.65 |
>     | **Diff** | **+2.46** | **+0.97** | **-1.45** | **-1.00** | **-0.95** | **-0.30** | **-1.00** | **-0.18** |
>
>     | Qwen2-7B | nq | triviaqa | hellaswag | gsm8k | math | mbpp | humaneval | Avg |
>     | :--- | :---: | :---: | :---: | :---: | :---: | :---: | :---: | :---: |
>     | DaaR w/ Pseudo | 16.88 | 57.58 | 73.03 | 75.40 | 38.10 | 52.00 | 64.94 | 53.99 |
>     | DaaR w/ GT | 18.75 | 58.35 | 72.02 | 73.40 | 35.75 | 51.50 | 64.01 | 53.40 |
>     | **Diff** | **+1.87** | **+0.77** | **-1.01** | **-2.00** | **-2.35** | **-0.50** | **-0.93** | **-0.59** |
>
>     | Qwen2.5-7B | nq | triviaqa | hellaswag | gsm8k | math | mbpp | humaneval | Avg |
>     | :--- | :---: | :---: | :---: | :---: | :---: | :---: | :---: | :---: |
>     | DaaR w/ Pseudo | 15.83 | 58.65 | 72.48 | 80.20 | 16.70 | 64.20 | 68.29 | 53.76 |
>     | DaaR w/ GT | 16.04 | 58.75 | 71.87 | 79.40 | 16.30 | 64.25 | 67.76 | 53.48 |
>     | **Diff** | **+0.21** | **+0.10** | **-0.61** | **-0.80** | **-0.40** | **+0.05** | **-0.53** | **-0.28** |
>
> This experiment reveals two critical insights:
>
> 1. **Model-aware pseudo-labels are more effective.** Using "true" labels leads to **an average performance drop** compared to our standard DaaR. This suggests that an LLM's **internal understanding of data** can be a better guide for fine-tuning than human-defined domain label.
> 2. **"True" labels can introduce suboptimal biases.** For instance, we observed that using human labels **biased the final model's capabilities**, often over-strengthening certain domains (e.g., **common sense**) at the expense of others, leading to  **lower performance on others**. This suggests apparent 'misclassifications' caused by pseudo-labels may actually more functionally effective groupings from the LLM's perspective.
>
> In addition to this new experiment, our **existing analysis in Appendix G.8** already demonstrates the **component-level stability** of DaaR (e.g., consistent seed generation, deterministic clustering, and reliable probe convergence).
>
> We greatly appreciate this constructive point. We will add this new experiment to our ablation studies, as it strongly highlights the advantages of DaaR's model-aware design.
>
> ---
>
> ### **W1: On the Embedding Representation**
> > *'unvalidated embedding representation: Assumes embedding-layer features accurately reflect data distribution (sec 3.2) without regression/ablation experiments to verify fidelity to true semantic distribution.'*
> >
>
> Thank you for this excellent question, which naturally extends the discussion from `our reply to W3` above.
>
> Our reliance on the model's own embedding layer is a deliberate design choice, rooted in DaaR's **model-aware and external-model-free** philosophy. As our new ablation experiment in `W3` demonstrated, imposing an external "true" perspective (like human labels) can actually misguide the fine-tuning process.
>
> The reason of selecting embedding layer is in **lines 569-573**. And we posit that the principle applies here: a more relevant semantic distribution for fine-tuning an LLM is **the one perceived by the LLM itself**. Besides, defining a universal "ground-truth" or a SOTA semantic distribution is a complex challenge in recent works [1,2] that lies **beyond the scope of this work**.
>
> Nevertheless, to investigate the practical implications, we processed our full dataset (40K) using **several SOTA external embedding models** of a similar model size (7B). To ensure a fair comparison, all experimental setups were identical, utilizing a batch size of 16 and the Transformers library for embedding extraction. This analysis revealed two key points:
>
> - **No Single "True" Representation Required:** While each embedding model generates a distinct semantic space, all of them **successfully differentiate the domains** (since the new picture can't be displayed, it is similar as Figure 12).
> - **Significant Computational Overhead:** Using external models introduced substantial computational costs. Specifically, their processing times were **175 and 192 times longer**, respectively, and they even led to OOM errors in 40GB of GPU. This highlights the significant efficiency advantage of utilizing the model's native embedding layer.
>
> | Models | Qwen2-7B-Embedding-Layer | GTE-Qwen2-7B | Qwen3-8B-Embedding |
> |:---:|:---:|:---:|:---:|
> | Time Cost | 9.23s | 1662s **(175x)** | 1774s **(192x)** |
>
> Given these findings, both the alignment with our model-aware principle and the significant computational overhead, we posit that using the LLM's own embedding layer is **the sound and efficient choice for DaaR**. We will add a note on this to the revised manuscript.
>
> [1] Qwen3 Embedding: Advancing Text Embedding and Reranking Through Foundation Models
>
> [2] Jasper and Stella: Distillation of SOTA Embedding Models
>
> ---
>
> ### **W2: On the Phrasing of Performance Gains**
> > *'DAAR shows minimal improvement (0.1%) for Llama3.1-8B over the best baseline (SuperFilter) in Table 2, undermining claims of "notable" superiority.'*
> >
>
> Thank you for this observation. Your feedback is very helpful for improving the precision of our manuscript.
>
> First, we wish to clarify that the experimental task is a **particularly challenging** task (as detailed in lines 285-291), where most existing baselines show limited effectiveness. Our primary claim is that DaaR achieves **consistent SOTA performance** across all tested models and scales.
>
> To further validate this consistency across a broader range of setups, we have conducted supplementary experiments on two additional models, **Llama3.2-3B** and **Qwen2.5-14B**. For these experiments, we compared DaaR against two key baselines: **Raw** and **Random** (a strong baseline in our existing results). Our supplementary results are as follows:
>
> - **On Qwen2.5-14B (a larger model)**
>
> | Method | NQ | TriviaQA | Hellaswag | GSM8K | MATH | MBPP | HumanEval | Avg |
> | :---: | :---: | :---: | :---: | :---: | :---: | :---: | :---: | :---: |
> | Raw | 10.89 | 66.56 | 76.86 | 86.00 | 19.70 | 5.40 | 78.66 | 49.15 |
> | Random | 20.06 | 65.80 | 76.76 | 86.40 | 36.90 | 64.20 | 75.46 | 60.80 |
> | DaaR | 19.73 | 66.26 | 77.19 | 85.00 | 39.10 | 66.80 | 76.69 | **61.54** |
>
> - **On LLaMA3.1-8B (a smaller model):**
>
> | Method | NQ | TriviaQA | Hellaswag | GSM8K | MATH | MBPP | HumanEval | Avg |
> | :---: | --- | --- | --- | --- | --- | --- | --- | --- |
> | Raw | 7.62 | 53.1 | 68.77 | 26.6 | 4.5 | 3.8 | 23.17 | 26.79 |
> | Random | 16.03 | 53.56 | 68.46 | 28.7 | 6.15 | 4.65 | 29.12 | 29.52 |
> | DaaR | 15.13 | 53.79 | 68.23 | 29.10 | 5.63 | 5.3 | 30.34 | **29.64** |
>
> As the new results demonstrate, DaaR **consistently outperforms** on both the larger Qwen2.5-14B and the smaller Llama3.2-3B. On Qwen2.5-14B, DaaR achieves the **highest average score (61.54)**, marking an improvement over the strong Random baseline with **0.74**. Similarly, on Llama3.2-3B, DaaR (**29.64**) maintains its **advantage**, that are consistent with our prior findings on the Llama architecture.
>
> Nevertheless, we agree that the term ***"notable" may not be the most precise descriptor*** for every individual result. Following your valuable suggestion, we will revise our manuscript to use more accurate language, such as "**consistently**" to characterize the overall performance pattern.
>
> ---
>
> ### **Closing Remarks**
> Thank you again for your insights and feedback! We hope these responses and the new revision can address your concerns and enhance your confidence in the acceptance of this paper. If you have any additional concerns or queries, we warmly invite you to share them with us.

---

> > ### Author Response · Authors · 2025-08-08
> > **Looking forward to discuss our rebuttal for Submission 27326**
> >
> > Dear Reviewer kdaW,
> >
> > We hope this message finds you well. Thank you again for your time and valuable feedback, which has been **instrumental in strengthening our manuscript**.
> >
> > Inspired by your suggestions, we have worked diligently to address your concerns, conducting **seven new experiments** to further validate our claims and design choices. We've summarized the main updates below, hoping this makes it easier for you to review our progress.
> >
> > - **On the 'True' Domain Distributions**: We designed a new ablation study replacing our model-aware pseudo-labels **with 'ground-truth' human annotations**. The results show that using these 'true' labels actually **leads to a performance drop**, which validates our core design principle that leveraging the **LLM's internal data perspective** is a more effective and robust choice.
> >
> > - **On the Embedding Representation**: We investigated the practical implications of using **external, SOTA embedding models**. Our analysis revealed this approach introduces **computational overhead** without a clear 'true' representation benefit, confirming that using the LLM's own embeddings is a sound and efficient design choice for our framework.
> >
> > - **On the Phrasing of Performance Gains**: We agree with your observation and conducted **supplementary experiments on two additional LLMs** to further validate our claims. These new results confirm our method's **"consistent"** outperformance across a wider range of settings, and we will revise our manuscript to use "consistent" as a more accurate term.
> >
> > As the discussion period is nearing its end (less than a day remaining), we would be truly grateful for the opportunity to hear your thoughts on these updates. Your feedback is crucial for us to know if our new results have sufficiently addressed your concerns. We welcome any further discussion you might wish to have.
> >
> > Thank you again for your time and guidance!

---

### Author Response · Authors · 2025-08-06
**Eager to discuss our rebuttal for Submission 27326**

Dear Reviewers,

We hope this message finds you well. We are writing to follow up on our rebuttal and to thank you once more for your constructive feedback, which has significantly strengthened our work.

We are keen to ensure our revisions meet your expectations and would greatly appreciate the opportunity to discuss them before the discussion period concludes (<3 days). To make it easier to see how we've incorporated your feedback, here is a brief overview of our reviews and responses.

We were pleased by the consensus on our work's ***novelty*** (Reviewer kdaW, bjgP, Ge2Q), ***comprehensive experiments*** (Reviewer kdaW, bjgP, Ge2Q), ***theoretical grounding*** (Reviewer bjgP, Ge2Q, Bbh9), ***real-world practicality*** (Reviewer Ge2Q, Bbh9), and ***computational efficiency*** (Reviewer bjgP, Ge2Q).

Our response includes **detailed point-by-point replies and 13 new experiments**, such as:

*   **For Reviewer kdaW's concerns on domain distribution and embeddings:** We ran new experiments with **ground-truth labels** (`Exp 1-3`) and **varied SOTA embedding models** (`Exp 4-5`).
*   **For Reviewer kdaW & Ge2Q's concerns on generalization:** We tested on **more model sizes and architectures** (`Exp 6-7`).
*   **For Reviewer bjgP's question on information leakage:** We conducted ablations on a **setup without injection** (`Exp 8-10`).
*   **For Reviewer Ge2Q & Bbh9's questions on error analysis:** We provided a more **in-depth confidence analysis** (`Exp 11-13`).

Your feedback on these updates would be invaluable to us. Please let us know if our clarifications and new results have resolved your concerns, or if there are any remaining points you would like to discuss.

We look forward to hearing from you.

Best regards,

The Authors of Submission 27326

---

### Note · Authors · 2025-08-13

Dear Reviewers, AC, and SAC,

Thank you for the thoughtful reviews and constructive feedback. Below, we would like to provide a factual summary of the author–reviewer interactions, which may help provide a quick tour of the discussion status.

The discussion period was productive in clarifying our work's contributions. Here is the current status:

- `Ge2Q` (Initial: 5) expressed their strong support: “*I maintain that this is a strong paper and should be accepted.*”
- `bjgP` (Initial: 3) engaged constructively, satisfied with our initial rebuttal and considered *raising score* and *a further increase per the response to their new questions*. After our second-round response, they confirmed that it "*addressed my concerns and I will consider raising my score.*"
- `kdaW` (Initial: 4) highlighted our work’s *novelty* and *empirical rigor*; after our responses, no further questions were raised.
- `Bbh9` (Initial: 3) acknowledged our rebuttal as *thoughtful*, noting that “*This direction is promising.*” As requested, we provided a detailed analysis comparing our work with pre-training methods and outlined our concrete revision plans, after which we received no further follow-up questions.

We respectfully hope this summary aids in the final score consolidation, particularly if an intended score adjustment has not yet been reflected in the system.

We were also encouraged that key merits of our work were noted: novelty (`kdaW`, `bjgP`, `Ge2Q`), comprehensive experiments (`kdaW`, `bjgP`, `Ge2Q`), theoretical grounding (`bjgP`, `Ge2Q`, `Bbh9`), practical relevance (`bjgP`, `Ge2Q`, `Bbh9`), and computational efficiency (`bjgP`, `Ge2Q`).

Guided by the valuable discussion, we are committed to integrating the following enhancements into the final version:

1. **Method rationale & potential**: Ablations on domain distribution & embedding models (`kdaW`), leakage, stage-2 and marginal data (`bjgP`), statistical significance (`Ge2Q` & `Bbh9`), and generalization across additional model sizes and architectures (`kdaW` & `Ge2Q`).
2. **Deeper analysis & theory**: Description of theoretical basis and new analysis based on approximation error (`bjgP`).
3. **Presentation**: Clearer definitions, improved figures, new pseudo-code, and an explicit comparison with pre-training methods (`Bbh9`).

We are fully committed to incorporating all the feedback to deliver the highest quality manuscript.

Thank you again for your time and productive guidance!

Sincerely,
The Authors

---

### Decision · Program_Chairs · 2025-09-17

**Decision:**

Accept (poster)

**Comment:**

The paper provides a framework that uses diversity to fine-tune LLMs on domain-undetermined data. In terms of novelty, although there was no full consensus, three reviewers did find the approach and in particular the overall result novel: Ge2Q (commenting about the approach) “Unlike methods that require pre-existing domain labels, DAAR uniquely uses the LLM's own embedding ...” bjgP (commenting about the result) “The self-supervised framework DAAR is proposed to address domain-undetermined data fine-tuning by leveraging diversity as a reward signal, breaking free from reliance on pre-labeled data.”. In contrast one review mentioned novelty to be limited in that the individual components are not new, and the novelty lies in the overall pipeline. Given the other reviews, I see this as a minor issue, and believe that in terms of novelty and interest to the community, the result and methods meet the bar of NeurIPS.

Another property discussed is the empirical evidence providing the quality of the proposed technique. Here too there was no consensus but 3 out of 4 reviews did find the experiments to be sufficient. The initial reviews (Ge2Q, bjgP and kdaW) positively mentioned the breadth of the experiments: “The framework establishes a new state-of-the-art average performance across three model families and seven benchmarks, consistently outperforming nine different baseline methods”, “Experiments across 7 cross-domain benchmarks (e.g., mathematical reasoning, coding) and 3 model families (Qwen2, Llama3.1, etc.) show DAAR outperforms 9 baselines.”, “Demonstrates consistent SOTA results across diverse LLMs (e.g., Qwen, Llama) and challenging tasks (e.g., MATH, HumanEval)”. During the rebuttal, the reviewers raised the need of additional empirical studies evaluating sub-procedures, components, and comparing them against additional baselines. The authors provided experiments for most of these suggestions, mitigating the concerns of most reviewers. Some concerns remained (notably regarding additional baselines mentioned by Bbh9) but based on the discussion done among the reviewers and myself, I am convinced these are merely nice-to-have additions not key in proving the validity of the work.

To conclude, although the reviewers did not reach a consensus w.r.t the decision over the paper, I'm convinced that the remaining weaknesses of the paper are minor compared to its strengths and believe it will be a welcome addition to NeurIPS.